# The interpretation of computational model parameters depends on the context

**Maria Katharina Eckstein[1]\*, Sarah L Master[1,2], Liyu Xia[1,3], Ronald E Dahl[4†], Linda Wilbrecht[1,5], Anne GE Collins[1,5]**

[1]Department of Psychology, University of California, Berkeley, Berkeley, United States; [2]Department of Psychology, New York University, New York, United States; [3]Department of Mathematics, University of California, Berkeley, Berkeley, United States; [4]Institute of Human Development, University of California, Berkeley, Berkeley, United States; [5]Helen Wills Neuroscience Institute, University of California, Berkeley, Berkeley, United States

**\*For correspondence:**
maria.eckstein@berkeley.edu

**Present address:** [†]School of Public Health, UC Berkeley, Berkeley, United States

**Competing interest:** The authors declare that no competing interests exist.

**Abstract** Reinforcement Learning (RL) models have revolutionized the cognitive and brain sciences, promising to explain behavior from simple conditioning to complex problem solving, to shed light on developmental and individual differences, and to anchor cognitive processes in specific brain mechanisms. However, the RL literature increasingly reveals contradictory results, which might cast doubt on these claims. We hypothesized that many contradictions arise from two commonly-held assumptions about computational model parameters that are actually often invalid: That parameters *generalize* between contexts (e.g. tasks, models) and that they capture *interpretable* (i.e. unique, distinctive) neurocognitive processes. To test this, we asked 291 participants aged 8–30 years to complete three learning tasks in one experimental session, and fitted RL models to each. We found that some parameters (exploration / decision noise) showed significant generalization: they followed similar developmental trajectories, and were reciprocally predictive between tasks. Still, generalization was significantly below the methodological ceiling. Furthermore, other parameters (learning rates, forgetting) did not show evidence of generalization, and sometimes even opposite developmental trajectories. Interpretability was low for all parameters. We conclude that the systematic study of context factors (e.g. reward stochasticity; task volatility) will be necessary to enhance the generalizability and interpretability of computational cognitive models.

## Editor's evaluation

This study adopts a within-participant approach to address two important questions in the field of human reinforcement learning: to what extent do estimated computational model parameters generalize across different tasks and can their meaning be interpreted in the same way in different task contexts? The authors find that inferred parameters show moderate to little generalizability across tasks, and that their interpretation strongly depends on task context.

## Introduction

In recent decades, cognitive neuroscience has made breakthroughs in computational modeling, demonstrating that reinforcement learning (RL) models can explain foundational aspects of human thought and behavior. RL models can explain not only simple cognitive processes such as stimulus-outcome and stimulus-response learning (*Schultz et al., 1997*; *O'Doherty et al., 2004*; *Gläscher*

*et al., 2009*), but also highly complex processes, including goal-directed, temporally extended behavior (*Ribas-Fernandes et al., 2011*; *Daw et al., 2011*), meta-learning (*Wang et al., 2018*), and abstract problem solving requiring hierarchical thinking (*Eckstein and Collins, 2020*; *Botvinick, 2012*; *Collins and Koechlin, 2012*; *Werchan et al., 2016*). Underlining their centrality in the study of human cognition, RL models have been applied across the lifespan (*van den Bos et al., 2018*; *Bolenz et al., 2017*; *Nussenbaum and Hartley, 2019*), and in both healthy participants and those experiencing psychiatric illness (*Huys et al., 2016*; *Adams et al., 2016*; *Hauser et al., 2019*; *Ahn and Busemeyer, 2016*; *Deserno et al., 2013*). RL models are of particular interest because they also promise a close link to brain function: A specialized network of brain regions, including the basal ganglia and prefrontal cortex, implement computations that mirror specific components of RL algorithms, including action values and reward prediction errors (*Frank and Claus, 2006*; *Niv, 2009*; *Lee et al., 2012*; *O'Doherty et al., 2015*; *Glimcher, 2011*; *Garrison et al., 2013*; *Dayan and Niv, 2008*). In sum, RL, explaining behavior ranging from simple conditioning to complex problem solving, appropriate for diverse human (and nonhuman) populations, based on a compelling theoretical foundation (*Sutton and Barto, 2017*), and with strong ties to brain function, has seen a surge in published studies since its introduction (*Palminteri et al., 2017*), and emerged as a powerful and potentially unifying modeling framework for cognitive and neural processing.

Computational modeling enables researchers to condense rich behavioral datasets into simple, falsifiable models (e.g. RL) and fitted model parameters (e.g. learning rate, decision temperature) (*van den Bos et al., 2018*; *Palminteri et al., 2017*; *Daw, 2011*; *Wilson and Collins, 2019*; *Guest and Martin, 2021*; *Blohm et al., 2020*). These models and parameters are often interpreted as a reflection of (or '*window into*') cognitive and/or neural processes, with the ability to dissect these processes into specific, unique components, and to measure participants' inherent characteristics along these components. For example, RL models have been praised for their ability to separate the decision making process into value updating and choice selection stages, allowing for the separate investigation of each dimension. Hereby, RL models infer person-specific parameters for each dimension (e.g. learning rate and decision noise), seemingly providing a direct measure of individuals' inherent characteristics. Crucially, many current research practices are firmly based on these (often implicit) assumptions, which give rise to the expectation that parameters have a task- and model-independent *interpretation* and will seamlessly *generalize* between studies. However, there is growing—though indirect—evidence that these assumptions might not (or not always) be valid. The following section lays out existing evidence in favor and in opposition of model generalizability and interpretability. Building on our previous opinion piece, which—based on a review of published studies—argued that there is less evidence for model generalizability and interpretability than expected based on current research practices (*Eckstein et al., 2021*), this study seeks to directly address the matter empirically.

Many current research practices are implicitly based on the interpretability and generalizability of computational model parameters (despite the fact that many researchers explicitly distance themselves from them). For our purposes, we define a model variable (e.g. fitted parameter) as *generalizable* if it is consistent across uses, such that a person would be characterized with the same values independent of the specific model or task used to estimate the variable. Generalizability is a consequence of the assumption that parameters are intrinsic to participants rather than task dependent (e.g. a high learning rate is a personal characteristic that might reflect an individual's unique brain structure). One example of our implicit assumptions about generalizability is the fact that we often directly compare model parameters between studies—for example, comparing our findings related to learning rate parameters to a previous study's findings related to learning rate parameters. Note that such a comparison is only valid if parameters capture the same underlying constructs across studies, tasks, and model variations, that is, if parameters *generalize*. The literature has implicitly equated parameters in this way in review articles (*Huys et al., 2016*; *Adams et al., 2016*; *Hauser et al., 2019*; *Frank and Claus, 2006*; *Niv, 2009*; *Lee et al., 2012*; *O'Doherty et al., 2015*; *Glimcher, 2011*; *Dayan and Niv, 2008*), meta-analyses (*Garrison et al., 2013*; *Yaple and Yu, 2019*; *Liu et al., 2016*), and also most empirical papers, by relating parameter-specific findings across studies. We also implicitly evoke parameter generalizability when we study task-independent empirical parameter priors (*Gershman, 2016*), or task-independent parameter relationships (e.g. interplay between different kinds of learning rates [*Harada, 2020*]), because we presuppose that parameter settings are inherent to participants, rather than task specific.

We define a model variable as *interpretable* if it isolates specific and unique cognitive elements, and/or is implemented in separable and unique neural substrates. Interpretability follows from the assumption that the decomposition of behavior into model parameters 'carves cognition at its joints', and provides fundamental, meaningful, and factual components (e.g. separating value updating from decision making). We implicitly invoke interpretability when we tie model variables to neural substrates in a task-general way (e.g. reward prediction errors to dopamine function [*Schultz and Dickinson, 2000*]), or when we use parameters as markers of psychiatric conditions in a model-independent way (e.g. working-memory deficits in schizophrenia [*Collins et al., 2014*]). Interpretability is also required when we relate abstract parameters to aspects of real-world decision making (*Heinz et al., 2017*), and generally, when we assume that model variables are particularly 'theoretically meaningful' (*Huys et al., 2016*).

However, in the midst of the growing application of computational modeling of behavior, the focus has also shifted toward inconsistencies and apparent contradictions in the emerging literature, which are becoming apparent in cognitive (*Nassar and Frank, 2016*), developmental (*Nussenbaum and Hartley, 2019*; *Javadi et al., 2014*; *Blakemore and Robbins, 2012*; *DePasque and Galván, 2017*), clinical (*Adams et al., 2016*; *Hauser et al., 2019*; *Ahn and Busemeyer, 2016*; *Deserno et al., 2013*), and neuroscience studies (*Garrison et al., 2013*; *Yaple and Yu, 2019*; *Liu et al., 2016*; *Mohebi et al., 2019*), and have recently become the focus of targeted investigations (*Robinson and Chase, 2017*; *Weidinger et al., 2019*; *Brown et al., 2020*; *Pratt et al., 2021*). For example, some developmental studies have shown that learning rates increased with age (*Master et al., 2020*; *Davidow et al., 2016*), whereas others have shown that they decrease (*Decker et al., 2015*). Yet others have reported U-shaped trajectories with either peaks (*Rosenbaum et al., 2020*) or troughs (*Eckstein et al., 2022*) during adolescence, or stability within this age range (*Palminteri et al., 2016*) (for a comprehensive review, see *Nussenbaum and Hartley, 2019*; for specific examples, see *Nassar and Frank, 2016*). This is just one striking example of inconsistencies in the cognitive modeling literature, and many more exist (*Eckstein et al., 2022*). These inconsistencies could signify that computational modeling is fundamentally flawed or inappropriate to answer our research questions. Alternatively, inconsistencies could signify that the method is valid, but our current implementations are inappropriate (*Palminteri et al., 2017*; *Uttal, 1990*; *Webb, 2001*; *Navarro, 2019*; *Yarkoni, 2020*; *Wilson and Collins, 2019*). However, we hypothesize that inconsistencies can also arise for a third reason: Even if both method and implementation are appropriate, inconsistencies like the ones above are expected—and not a sign of failure—if implicit assumptions of generalizability and interpretability are not always valid. For example, model parameters might be more context-dependent and less person-specific than we often appreciate (*Nussenbaum and Hartley, 2019*; *Nassar and Frank, 2016*; *Yaple and Yu, 2019*; *Behrens et al., 2007*; *McGuire et al., 2014*).

To illustrate this point, the current project began as an investigation into the development of learning in adolescence, with the aim of combining the insights of three different learning tasks to gain a more complete understanding of the underlying mechanisms. However, even though each task individually showed strong and interesting developmental patterns in terms of model parameters (*Master et al., 2020*; *Eckstein et al., 2022*; *Xia et al., 2021*), these patterns were very different—and even contradictory—across tasks. This implied that specific model parameters (e.g. learning rate) did not necessarily isolate specific cognitive processes (e.g. value updating) and consistently measure individuals on these processes, but that they captured different processes depending on the learning context of the task (lack of *generalizability*). In addition, the processes identified by one parameter were not necessarily distinct from the cognitive processes (e.g. decision making) identified by other parameters (e.g. decision temperature), but could overlap between parameters (lack of *interpretability*). In a nutshell, the 'same' parameters seemed to measure something different in each task.

The goal of the current project was to assess these patterns formally: We determined the degree to which parameters generalized between three different RL tasks, investigated whether parameters were interpretable as unique and specific processes, and provide initial evidence for context factors that potentially modulate generalizability and interpretability of model parameters, including feedback stochasticity, task volatility, and memory demands. To this aim, we compared the same individuals' RL parameters, fit to different learning tasks in a single study, in a developmental dataset (291 participants, ages 8–30 years). Using a developmental dataset had several advantages: It provided large between-participant variance and hence better coverage of the parameter space, and allowed us

to specifically target outstanding discrepancies in the developmental psychology literature (***Nussenbaum and Hartley, 2019***). The three learning tasks we used varied on several common dimensions, including feedback stochasticity, task volatility, and memory demands (***Figure 1E***), and have been used previously to study RL processes (***Davidow et al., 2016***; ***Collins and Frank, 2012***; ***Javadi et al., 2014***; ***Master et al., 2020***; ***Eckstein et al., 2022***; ***Xia et al., 2021***). However, like many tasks in the literature, the tasks likely also engaged other cognitive processes besides RL, such as working memory and reasoning. The within-participant design of our study allowed us to test directly whether the same participants showed the same parameters across tasks (generalizability), and the combination of multiple tasks shed light on which cognitive processes the same parameters captured in each task (interpretability). We extensively compared and validated all RL models (***Palminteri et al., 2017***; ***Wilson and Collins, 2019***; ***Lee, 2011***) and have reported each task's unique developmental results separately (***Master et al., 2020***; ***Eckstein et al., 2022***; ***Xia et al., 2021***).

Our results show a striking lack of generalizability and interpretability for some tasks and parameters, but convincing generalizability for others. This reveals an urgent need for future research to address the role of context factors in computational modeling, and reveals the necessity of taking context factors into account when interpreting and generalizing results. It also suggests that some prior discrepancies are likely explained by differences in context.

## Results

This section gives a brief overview of the experimental tasks (***Figure 1B–D***) and computational models (***Figure 1F***; also see sections 'Task Design', 'Computational Models', and 'Appendix 2'; for details, refer to original publications [***Master et al., 2020***; ***Eckstein et al., 2022***; ***Xia et al., 2021***]). We then show our main findings on parameter generalizability (section 'Part I: parameter generalizability') and interpretability (section 'Part II: parameter interpretability'). All three tasks are learning tasks and have been previously well-captured by RL models, yet with differences in parameterization (***Javadi et al., 2014***; ***Davidow et al., 2016***; ***Collins and Frank, 2012***). In our study as well, the best-fitting RL models differed between tasks, containing some parameters that were the same across tasks, and some that were task-specific (***Figure 1F***). Thus, our setup provides a realistic reflection of the diversity of computational models in the literature.

Task A required participants to learn the correct associations between each of four stimuli (butterflies) and two responses (flowers) based on probabilistic feedback (***Figure 1B***). The best-fitting model contained three free parameters: learning rate from positive outcomes $\alpha_+$, inverse decision temperature $\beta$, and forgetting $F$. It also contained one fixed parameter: learning rate from negative outcomes $\alpha_- = 0$ (***Xia et al., 2021***). Task B required participants to adapt to unexpected switches in the action-outcome contingencies of a simple bandit task (only one of two boxes contained a gold coin at any time) based on semi-probabilistic feedback (***Figure 1C***). The best-fitting RL model contained four free parameters: $\alpha_+$, $\alpha_-$, $\beta$, and choice persistence $p$ (***Eckstein et al., 2022***). Task C required learning of stimulus-response associations like task A, but over several task blocks with varying numbers of stimuli, and using deterministic feedback (***Figure 1D***). The best model for this task combined RL and working memory mechanisms, containing RL parameters $\alpha_+$ and $\alpha_-$; working memory parameters capacity $K$, forgetting $F$, and noise $\epsilon$; and mixture parameter $\rho$, which determined the relative weights of RL and working memory (***Master et al., 2020***; ***Collins and Frank, 2012***). The Markov decision process (MDP) framework provides a common language to describe learning tasks like ours, by breaking them down into states, actions, and reward functions. Appendix 2 summarizes the tasks in this way and highlights major differences.

We employed rigorous model fitting, comparison, and validation to obtain the best-fitting models presented here (see Appendix 4 and ***Palminteri et al., 2017***; ***Daw, 2011***; ***Wilson and Collins, 2019***; ***Lee, 2011***): For each task, we compared a large number of competing models, based on different parameterizations and cognitive mechanisms, and selected the best one based on quantitative model comparison scores as well as the models' abilities to reproduce participants' behavior in simulation (***Appendix 4—figure 1***). We also used hierarchical Bayesian methods for model fitting and comparison where possible, to obtain the most accurate parameter estimates (***Lee, 2011***; ***Brown et al., 2020***). Individual publications provide further details on the set of models compared and validate the claim that the models presented here are the best-fitting ones for each task (***Master et al., 2020***; ***Eckstein et al., 2022***; ***Xia et al., 2021***), an important premise for the claim that individual parameters

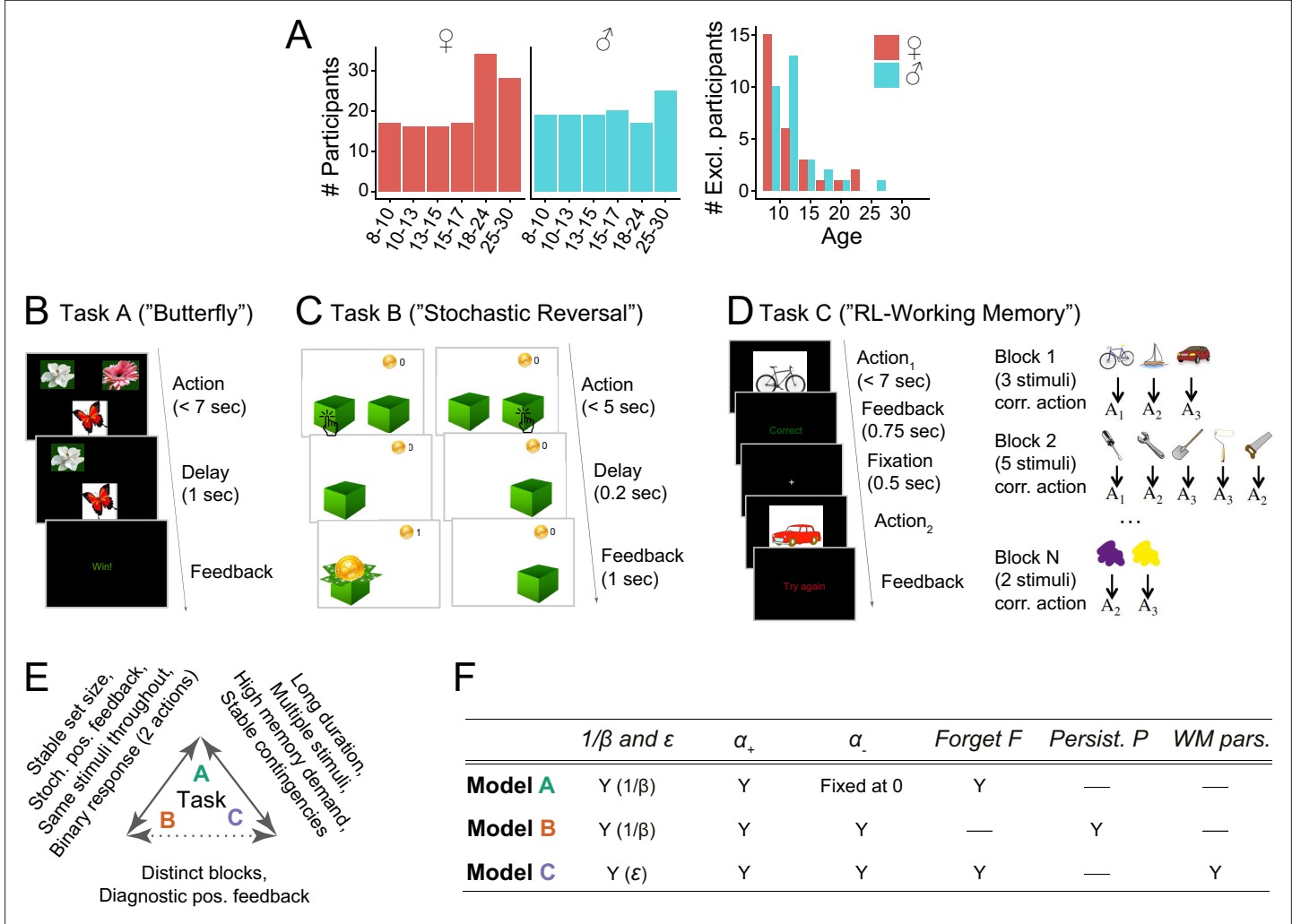

**Figure 1.** Overview of the experimental paradigm. (**A**) Participant sample. Left: Number of participants in each age group, broken up by sex (self-reported). Age groups were determined by within-sex age quartiles for participants between 8–17 years (see *Eckstein et al., 2022* for details) and 5 year bins for adults. Right: Number of participants whose data were excluded because they failed to reach performance criteria in at least one task. (**B**) Task A procedure of ('Butterfly task'). Participants saw one of four butterflies on each trial and selected one of two flowers in response, via button press on a game controller. Each butterfly had a stable preference for one flower throughout the task, but rewards were delivered stochastically (70% for correct responses, 30% for incorrect). For details, see section 'Task design' and the original publication (*Xia et al., 2021*). (**C**) Task B Procedure ('Stochastic Reversal'). Participants saw two boxes on each trial and selected one with the goal of finding gold coins. At each point in time, one box was correct and had a high (75%) probability of delivering a coin, whereas the other was incorrect (0%). At unpredictable intervals, the correct box switched sides. For details, see section 'Task design' and *Eckstein et al., 2022*. (**D**) Task C procedure ('Reinforcement learning-working memory'). Participants saw one stimulus on each trial and selected one of three buttons ($A_1 - A_3$) in response. All correct and no incorrect responses were rewarded. The task contained blocks of 2–5 stimuli, determining its 'set size'. The task was designed to disentangle set size-sensitive working memory processes from set size-insensitive RL processes. For details, see section 'Task design' and *Master et al., 2020*. (**E**) Pairwise similarities in terms of experimental design between tasks A (*Xia et al., 2021*), B (*Eckstein et al., 2022*), and C (*Master et al., 2020*). Similarities are shown on the arrows connecting two tasks; the lack of a feature implies a difference. E.g., a 'Stable set size' on tasks A and B implies an unstable set size in task C. Overall, task A shared more similarities with tasks B and C than these shared with each other. (**F**) Summary of the computational models for each task (for details, see section 'Computational models' and original publications). Each row shows one model, columns show model parameters. 'Y' (yes) indicates that a parameter is present in a given model, '—' indicates that a parameter is not present. '$\frac{1}{\beta}$ and $\epsilon$' refer to exploration / noise parameters; $\alpha_+$ ($\alpha_-$) to learning rate for positive (negative) outcomes; 'Persist. P' to persistence; 'WM pars'. to working memory parameters.

are well estimated. This qualitative validation step for each dataset ensures that potential parameter discrepancies between tasks are not due to a lack of modeling quality, and can indeed provide accurate information about parameter generalizability and interpretability. (Though we acknowledge that no model is ever right.)

## Part I: parameter generalizability

Crucially, the parameter inconsistencies observed in previous literature could be caused by non-specific differences between studies (e.g. participant samples, testing procedures, modeling approaches, research labs). Our within-participant design allows us to rule these out by testing whether the same participants show different parameter values when assessed using different tasks; this finding would be strong evidence for the hypothesized lack of parameter generalizability. To assess this, we first

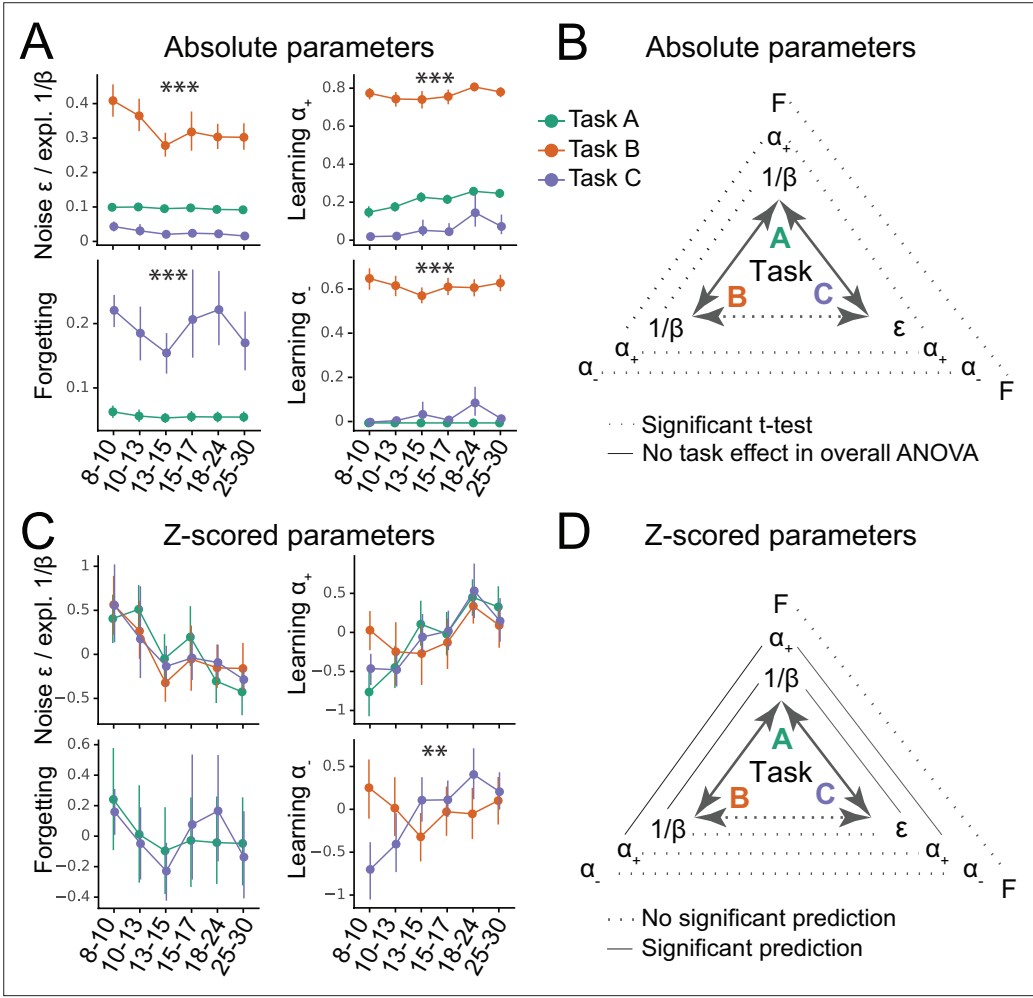

**Figure 2.** Generalizability of absolute parameter values (**A–B**) and of parameter age trajectories / z-scored parameters (**C–D**). (**A**) Fitted parameters over participant age (binned) for all three tasks (A: green; B: orange; C: blue). Parameter values differed significantly between tasks; significance stars show the p-values of the main effects of task on parameters (**Table 1**; * $p < .05$; ** $p < .01$; *** $p < .001$). Dots indicate means of the participants in each age group (for n's, see **Figure 1A**), error bars specify the confidence level (0–1) for interval estimation of the population mean. (**B**) Summary of the main results of part A. Double-sided arrows connecting tasks are replicated from **Figure 1E** and indicate task similarity (dotted arrow: small similarity; full arrow: large similarity). Lines connecting parameters between tasks show test statistics (**Table 1**). Dotted lines indicate significant task differences in Bonferroni-corrected pairwise t-tests (full lines would indicate the lack of difference). All t-tests were significant, indicating that absolute parameter values differed between each pair of tasks. (**C**) Parameter age trajectories, that is, within-task z-scored parameters over age. Age trajectories reveal similarities that are obscured by differences in means or variances in absolute values (part A). Significance stars show significant effects of task on age trajectories (**Table 2**). (**D**) Summary of the main results of part C. Lines connecting parameters between tasks show statistics of regression models predicting each parameter from the corresponding parameter in a different task (**Table 4**). Full lines indicate significant predictability and dotted lines indicate a lack thereof. In contrast to absolute parameter values, age trajectories were predictive in several cases, especially for tasks with more similarities (A and B; A and C), compared to tasks with fewer (B and C).

**Table 1.** Statistics of ANOVAs predicting raw parameter values from task (A, B, C). When an ANOVA showed a significant task effect, we followed up with post-hoc, Bonferroni-corrected t-tests. * $p < .05$; ** $p < .01$; *** $p < .001$.

| Parameter | Model | Tasks | F / t | df | p | sig. |
|---|---|---|---|---|---|---|
| $\frac{1}{\beta}$ | ANOVA | A, B | 830 | 1 | $p < 0.001$ | *** |
| | t-test | A vs B | 25 | 246 | $p < 0.001$ | *** |
| $\alpha_+$ | ANOVA | A, B, C | 2,018 | 2 | $p < 0.001$ | *** |
| | t-test | A vs B | 66 | 246 | $p < 0.001$ | *** |
| | t-test | A vs C | 12 | 246 | $p < 0.001$ | *** |
| | t-test | B vs C | 51 | 246 | $p < 0.001$ | *** |
| $\alpha_-$ | ANOVA | B, C | 2,357 | 1 | $p < 0.001$ | *** |
| | t-test | B vs C | 49 | 246 | $p < 0.001$ | *** |
| Forgetting | ANOVA | A, C | 161 | 1 | $p < 0.001$ | *** |
| | t-test | A vs C | 49 | 246 | $p < 0.001$ | *** |

determined whether participants showed similar parameter values across tasks, and then whether tasks showed similar parameter age trajectories.

## Differences in absolute parameter values

We first used repeated-measures analyses of variance (ANOVAs) to test for task effects on absolute parameter values (*Figure 2A*). When ANOVAs showed significant task effects, we followed up with repeated-measures t-tests to compare each pair of tasks, using the Bonferroni correction.

Learning rates $\alpha_+$ and $\alpha_-$ were so dissimilar across tasks that they occupied largely separate ranges: They were very low in task C ($\alpha_+$ mean: 0.07, sd: 0.18; $\alpha_-$ mean: 0.03, sd: 0.13), intermediate in task A ($\alpha_+$ mean: 0.22, sd: 0.09; $\alpha_-$ was fixed at 0), but fairly high in task B ($\alpha_+$ mean: 0.77, sd: 0.11; $\alpha_-$ mean: 0.62, sd: 0.14; for statistical comparison, see *Table 1*). Decision noise was high in task B ($\frac{1}{\beta}$ mean: 0.33, sd: 0.15), but low in tasks A ($\frac{1}{\beta}$ mean: 0.095, sd: 0.0087) and C ($\epsilon$ mean: 0.025, sd: 0.032; statistics in *Table 1* ignore $\epsilon$ because its absolute values were not comparable to $\frac{1}{\beta}$ due to the different parameterization; see section 'Computational models'). Forgetting was significantly higher in task C (mean: 0.19, sd: 0.17) than A (mean: 0.056, sd: 0.028). Task B was best fit without forgetting.

For all parameters, absolute parameter values hence differed substantially between tasks. This shows that the three tasks produced significantly different estimates of learning rates, decision noise/exploration, and forgetting for the same participants (*Figure 2B*). Interestingly, these parameter differences echoed specific task demands: Learning rates and noise/exploration were highest in task B, where frequent switches required quick updating and high levels of exploration. Similarly, forgetting was highest in task C, which posed the largest memory demands. Using regression models that controlled for age (instead of ANOVA) led to similar results (Table *Appendix 8—table 2*).

## Relative parameter differences

However, comparing parameters in terms of their absolute values has shortcomings because it minimizes the role of relative variance between participants, which reflects participants' mutual relationships to each other, and might be an important component of parameters. To test whether parameters generalized in relative, rather than absolute terms, we first correlated corresponding parameters between each pair of tasks, using Spearman correlation (*Appendix 8—figure 1*). Indeed, both $\alpha_+$ (*Appendix 8—figure 1A*) and noise/exploration parameters (*Appendix 8—figure 1B*) were significantly positively correlated between tasks A and B as well as between tasks A and C. Significant correlations were lacking between tasks B and C. This suggests that both $\alpha_+$ and noise/exploration generalized in terms of the relationships they captured between participants; however, this

**Table 2.** Assessing task effects on parameter age trajectories.

Model fits (AIC scores) of regression models predicting parameter age trajectories, comparing the added value of including ('AIC with task') versus excluding ('AIC without task') task as a predictor. Differences in AIC scores were tested statistically using F-tests. Better (smaller) model fits are highlighted in bold. The coefficients of the winning models (simpler model 'without task' unless adding task predictor leads to significantly better model fit) are shown in *Table 3*.

| Parameter | AIC without task | AIC with task | F(df) | p | sig. |
|---|---|---|---|---|---|
| $\frac{1}{\beta}/\epsilon$ | 2,044 | 2,054 | NA | NA | – |
| $\alpha_+$ | 2,044 | 2,042 | $F(4, 245) = 2.34$ | $p = 0.056$ | – |
| $\alpha_-$ | 1,395 | 1,373 | $F(2, 245) = 6.99$ | $p = 0.0011$ | ** |
| Forgetting | 1,406 | 1,411 | NA | NA | – |

generalization was only evident between tasks A and B or A and C, potentially due to the fact that task A was more similar to tasks B and C than these were to each other (*Figure 1E*; also see section 'Main axes of variation'). Fig. *Appendix 8—figure 3* shows the correlations between all pairs of features in the dataset (model parameters and behavioral measures). Note that noise parameters generalized between tasks A and C despite differences in parameterization ($\epsilon$ vs. $1/\beta$), showing robustness in the characterization of choice stochasticity (*Appendix 8—figure 1B*).

## Parameter age trajectories

This correlation analysis, however, is limited in its failure to account for age, an evident source of variance in our dataset. This means that apparent parameter generalization could be driven by a common dependence on age, rather than underlying age-independent similarities. To address this, we next focused on parameter age trajectories, aiming to remove differences between tasks that are potentially arbitrary (e.g. absolute mean and variance), while conserving patterns that are potentially

**Table 3.** Statistical tests on age trajectories: mixed-effects regression models predicting z-scored parameter values from task (A, B, C), age, and squared age (months).

When the task-less model fitted best, the coefficients of this ('grand') model are shown, reflecting shared age trajectories (*Table 2*; $\frac{1}{\beta}/\epsilon$, $\alpha_+$, forgetting). When the age-based model fitted better, pairwise follow-up models are shown ($\alpha_-$), reflecting task differences. p-Values of follow-up models were corrected for multiple comparison using the Bonferroni correction. * $p < .05$; ** $p < .01$, *** $p < .001$.

| Parameter | Tasks | Predictor | $\beta$ | p(Bonf.) | sig. |
|---|---|---|---|---|---|
| $\frac{1}{\beta}/\epsilon$ | A, B, C | Intercept | 1.86 | < 0.001 | *** |
| | | Age (linear) | –0.17 | 0.003 | ** |
| | | Age (quadratic) | 0.004 | < 0.001 | *** |
| $\alpha_+$ | A, B, C | Intercept | –2.10 | < 0.001 | *** |
| | | Age (linear) | 0.20 | < 0.001 | *** |
| | | Age (quadratic) | –0.004 | < 0.001 | *** |
| $\alpha_-$ | B, C | Task (main effect) | 4.15 | < 0.001 | *** |
| | | Task * linear age (interaction) | 0.43 | < 0.001 | *** |
| | | Task * quadratic age (interaction) | –0.010 | < 0.001 | *** |
| Forgetting | A, C | Intercept | 0.37 | 0.44 | |
| | | Age (linear) | –0.034 | 0.53 | |
| | | Age (quadratic) | 0.001 | 0.63 | |

more meaningful (e.g. shape of variance, i.e. participants' values relative to each other). Age trajectories were calculated by z-scoring each parameter within each task (*Figure 2C*). To test for differences, mixed-effects regression was used to predict parameters of all tasks from two age predictors (age and squared age) and task (A, B, or C). A better fit of this model compared to the corresponding one without task indicates that task characteristics affected age trajectories. In this case, we followed up with post-hoc models comparing individual pairs of tasks.

For $\alpha_-$, the task-based regression model showed a significantly better fit, revealing an effect of task on $\alpha_-$'s age trajectory (*Table 2*). Indeed, $\alpha_-$ showed fundamentally different trajectories in task B compared to C (in task A, $\alpha_-$ was fixed): In task B, $\alpha_-$ decreased linearly, modulated by a U-shaped curvature (linear effect of age: $\beta = -0.11$, $p < 0.001$; quadratic: $\beta = 0.003$, $p < 0.001$), but in task C, it increased linearly, modulated by an inverse-U curvature (linear: $\beta = 0.32$, $p < 0.001$; quadratic: $\beta = -0.07$, $p < 0.001$; *Figure 2C*). The fact that these patterns are opposites of each other was reflected in the significant interaction terms of the overall regression model (*Table 3*). Indeed, we previously reported a U-shaped trajectory of $\alpha_-$ in task B, showing a minimum around age 13–15 (*Eckstein et al., 2022*), but a consistent increase up to early adulthood in task C (*Xia et al., 2021*). This shows striking differences when estimating $\alpha_-$ using task B compared to C. These differences might reflect differences in task demands: Negative feedback was diagnostic in task C, requiring large learning rates from negative feedback $\alpha_-$ for optimal performance, whereas negative feedback was not diagnostic in task B, requiring small $\alpha_-$ for optimal performance.

For $\alpha_+$, adding task as a predictor did not improve model fit, suggesting that $\alpha_+$ showed similar age trajectories across tasks (*Table 2*). Indeed, $\alpha_+$ showed a linear increase that tapered off with age in all tasks (linear increase: task A: $\beta = 0.33$, $p < 0.001$; task B: $\beta = 0.052$, $p < 0.001$; task C: $\beta = 0.28$, $p < 0.001$; quadratic modulation: task A: $\beta = -0.007$, $p < 0.001$; task B: $\beta = -0.001$, $p < 0.001$; task C: $\beta = -0.006$, $p < 0.001$). For noise/exploration and forgetting parameters, adding task as a predictor also did not improve model fit (*Table 2*), suggesting similar age trajectories across tasks. For decision noise/exploration, the grand model revealed a linear decrease and tapering off with age (*Figure 2C*; *Table 3*), in accordance with previous findings (*Nussenbaum and Hartley, 2019*). For forgetting, the grand model did not reveal any age effects (*Figure 2C*; *Table 3*), suggesting inconsistent or lacking developmental changes.

In summary, $\alpha_-$ showed different age trajectories depending on the task. This suggests a lack of generalizability: The estimated developmental trajectories of learning rates for negative outcomes might not generalize between experimental paradigms. However, the age trajectories of noise/exploration parameters, $\alpha_+$, and forgetting did not differ between tasks. This lack of statistically-significant task differences might indicate parameter generalizability—but it could also reflect high levels of parameter estimation noise. Subsequent sections will disentangle these two possibilities.

## Predicting age trajectories

The previous analysis, focusing on parameter differences, revealed some *lack* of generalization (e.g. $\alpha_-$). The next analysis takes the inverse approach, assessing similarities in an effort to provide evidence *for* generalization: We used linear regression to predict participants' parameters in one task from the corresponding parameter on another task, controlling for age and squared age.

**Table 4.** Statistics of the regression models predicting each parameter from the corresponding parameter in a different task, while controlling for age.

Results were identical when predicting task A from B and task B from A, for all pairs of tasks. Therefore, only one set of results is shown, and predictor and outcome task are not differentiated. Stars indicate significance as before; '$ indicates $p < 0.1$.

| Parameter | Tasks | $\beta$ | p | sig. |
|---|---|---|---|---|
| $\frac{1}{\beta}, \epsilon$ | A & B | 0.28 | | *** |
| | A & C | 0.19 | 0.0022 | ** |
| | B & C | 0.039 | 0.54 | |
| $\alpha_+$ | A & B | 0.13 | 0.035 | * |
| | A & C | 0.23 | | *** |
| | B & C | –0.073 | 0.25 | |
| $\alpha_-$ | B & C | –0.12 | 0.058 | $ |
| Forgetting | A & C | 0.097 | 0.13 | |

For both $\alpha_+$ and noise/exploration parameters, task A predicted tasks B and C, and tasks B and C predicted task A, but tasks B and C did not predict each other (*Table 4*; *Figure 2D*), reminiscent of the correlation results (section 'Relative parameter differences'). For $\alpha_-$, tasks B and C showed a marginally significant *negative* relationship (*Table 4*), suggesting that predicting $\alpha_-$ between tasks can lead to systematically biased predictions, confirming the striking differences observed before (section 'Parameter age trajectories'). For forgetting, tasks A and C were not predictive of each other (*Table 4*), suggesting that the lack of significant differences we observed previously (*Table 3*) did not necessarily imply successful generalization, but might have been caused by other factors, for example, elevated noise.

## Statistical comparison to generalizability ceiling

Our analyses so far suggest that some parameters did not generalize between tasks: We observed differences in age trajectories (section 'Parameter age trajectories') and a lack of mutual prediction (section 'Predicting age trajectories'). However, the lack of correspondence could also arise due to other factors, including behavioral noise, noise in parameter fitting, and parameter trade-offs within tasks. To rule these out, we next established the ceiling of generalizability attainable using our method.

We established the ceiling in the following way: We first created a dataset with perfect generalizability, simulating behavior from agents that use the same parameters across all tasks (*Appendix 5—figure 1*). We then fitted this dataset in the same way as the human dataset (e.g. using the same models), and performed the same analyses on the fitted parameters, including an assessment of age trajectories (*Appendix 5—table 1*) and prediction between tasks (*Appendix 5—table 2*, *Appendix 5—table 3*, and *Appendix 5—table 4*). These results provide the practical ceiling of generalizability, given the limitations of our data and modeling approach. We then compared the human results to this ceiling to ensure that the apparent lack of generalization was a valid conclusion, rather than stemming from methodological constraints: If the empirical human dataset is significantly below ceiling, we can conclude a lack of generalization, but if it is not significantly different from the expected ceiling, our approach might lack validity.

The results of this analysis support our conclusions. Specifically, whereas humans had shown divergent trajectories for parameter $\alpha_-$ (*Figure 2B*; *Table 1*), the simulated agents (that used the same parameters for all tasks) did not show task differences for $\alpha_-$ or any other parameter (*Appendix 5—figure 1B*, *Appendix 5—table 1*), even when controlling for age (*Appendix 5—table 2*, *Appendix 5—table 3*). Furthermore, the same parameters were predictive between tasks in all cases (*Appendix 5—table 4*). These results show that our method reliably detected parameter generalization in a dataset that exhibited generalization.

Lastly, we established whether the degree of generalization in humans was significantly different from agents. To this aim, we calculated the Spearman correlations between each pair of tasks for each parameter, for both humans (section 'Relative parameter differences'; *Appendix 8—figure 1*) and agents, and then compared humans and agents using bootstrapped confidence intervals (Appendix 5). Human parameter correlations were significantly below the ceiling for most parameters (exceptions: $\alpha_+$ in A vs B; $\epsilon$ / $\frac{1}{\beta}$ in A vs C; *Appendix 5—figure 1C*). This suggests that the human sample showed less-than-perfect generalization for most task combinations and most parameters: Generalization was lower than in agents for parameters forgetting, $\alpha_-$, $\alpha_+$ (in two of three task combinations), and $\epsilon$ / $\frac{1}{\beta}$ (in two of three task combinations).

## Summary part I: Generalizability

So far, no parameter has shown generalization between tasks in terms of *absolute values* (*Figure 2A and B*), but noise/exploration and $\alpha_+$ showed similar *age trajectories* (*Figure 2C*), at least in tasks that were sufficiently similar (*Figure 2D*). To summarize, (1) all parameters differed significantly between tasks in terms of absolute values (*Figure 2A and B*). Intriguingly, absolute parameter values varied more between tasks than between participants within tasks, suggesting that task demands played a larger role in determining parameter values than participants' individual characteristics. This was the case for all four model parameters (Noise/Exploration, $\alpha_+$, $\alpha_-$, and Forgetting). (2) However, there was evidence that in some cases, parameter age trajectories generalized between tasks: Task identity did not affect the age trajectories of noise/exploration, forgetting, or learning rate $\alpha_+$ (*Figure 2C*), suggesting possible generalization. However, only noise/exploration

and $\alpha_+$ age trajectories were the same between tasks, hence revealing deeper similarities, and this was only possible when tasks were sufficiently similar (**Table 4**), highlighting the limits of generalization. No generalization was possible for $\alpha_-$, whose age trajectory differed both qualitatively and quantitatively between tasks, showing striking inverse patterns. Like for absolute parameter values, differences in parameter age trajectories were likely caused by differences in task demands. (3) Parameter $\alpha_+$ reached ceiling generalizability between tasks A and B, and parameter $\epsilon$ / $\frac{1}{\beta}$ between tasks A and C. Generalizability of all other task combinations and parameters was significantly lower than expected from a perfectly-generalizing population. (4) For the parameters whose age trajectories showed signs of generalization, our results replicated patterns in the literature, with noise/exploration decreasing and $\alpha_+$ increasing from childhood to early adulthood (**Nussenbaum and Hartley, 2019**).

## Part II: Parameter interpretability

To address the second assumption identified above, Part II focuses on parameter interpretability, testing whether parameters captured specific, unique, and meaningful cognitive processes. To this end, we first investigated the relations between different parameters to assess whether individual parameters were uniquely interpretable (i.e. specific and distinct from each other). We then determined how parameters were related to observed behavior, seeking evidence for external interpretability.

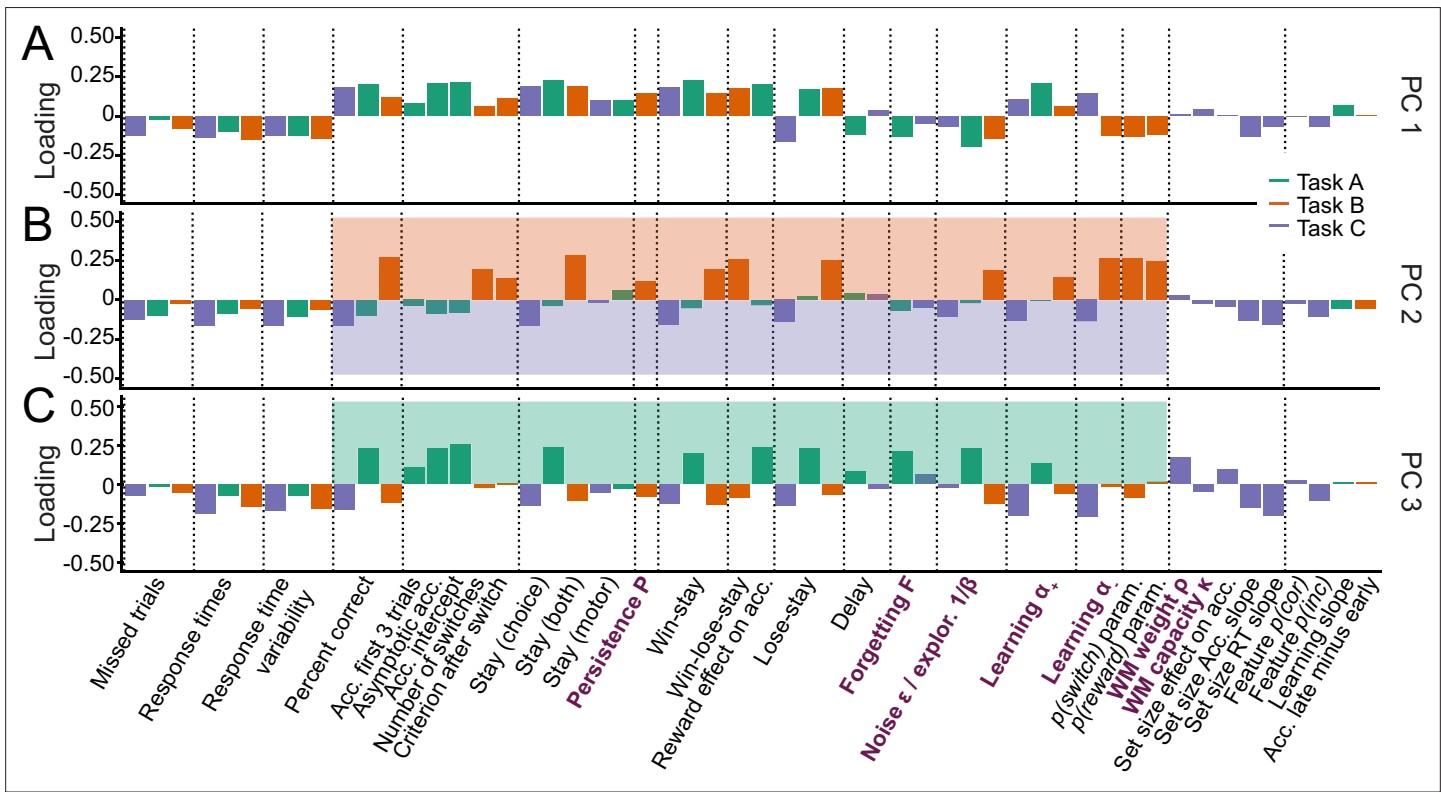

**Figure 3.** Identifying the major axes of variation in the dataset. A PCA was conducted on the entire dataset (39 behavioral features and 15 model parameters). The figure shows the factor loadings (y-axis) of of all dataset features (x-axis) for the first three PCs (panels A, B, and C). Features that are RL model parameters are bolded and in purple. Behavioral features are explained in detail in Appendix 1 and Appendix 3 (note that behavioral features differed between tasks). Dotted lines aid visual organization by grouping similar features across tasks (e.g. missed trials of all three tasks) or within tasks (e.g. working-memory-related features for task C). (**A**) PC1 captured broadly-defined task engagement, with negative loadings on features that were negatively associated with performance (e.g. number of missed trials) and positive loadings on features that were positively associated with performance (e.g. percent correct trials). (**B–C**) PC2 (**B**) and PC3 (**C**) captured task contrasts. PC2 loaded positively on features of task B (orange box) and negatively on features of task C (purple box). PC3 loaded positively on features of task A (green box) and negatively on features of tasks B and C. Loadings of features that are negative on PC1 are flipped in PC2 and PC3 to better visualize the task contrasts (section 'Principal component analysis (PCA)').

## Main axes of variation

To build a foundation for parameter interpretation, we first aimed to understand which cognitive processes and aspects of participant behavior were captured by each parameter. We opted for a data-driven approach, interpreting parameters based on the major axes of variance that emerged in our large dataset, identified without a priori hypotheses. Concretely, we used PCA to identify the principal components (PCs) of our joint dataset of behavioral features and model parameters (*Abdi and Williams, 2010*). We first gained a thorough understanding of these PCs, and then employed them to better understand what was captured by model parameters. Detailed information on our approach is provided in sections 'Principal component analysis (PCA)' (PCA methods), 'Appendix 6' (behavioral features), and *Appendix 8—figure 4* (additional PCA results).

We first analyzed PC1, the axis of largest variation and main source of individual differences in our dataset (25.1% of explained variance; *Appendix 8—figure 4A*). We found that behaviors that indicated good task participation (e.g. higher percentage of correct choices) loaded positively on PC1, whereas behaviors that indicated poor participation loaded negatively (e.g. more missed trials, longer response times; *Figure 3A*). This was the case for performance measures in the narrow sense of maximizing choice accuracy (e.g. percentage correct choices, trials to criterion, proportion of win-stay choices), but also in the wider sense of reflecting task engagement (e.g. number of missed trials, response times, response time variability). PC1 therefore captured a range of 'good', task-engaged behaviors, and is likely similar to the construct of 'decision acuity' (*Moutoussis et al., 2021*): Decision acuity was recently identified as the first component of a factor analysis (variant of PCA) conducted on 32 decision-making measures on 830 young people, and separated good and bad performance indices. Decision acuity reflected generic decision-making ability, predicted mental health factors, and was reflected in resting-state functional connectivity, but distinct from IQ (*Moutoussis et al., 2021*). Like decision acuity *Moutoussis et al., 2021*, our PC1 increased significantly with age, consistent with increasing performance (*Appendix 3—figure 1B*; age effects of subsequent PCs in *Appendix 8—figure 4*; *Appendix 8—table 1*).

How can this understanding of PC1 (decision acuity) help us interpret model parameters? In all three tasks, noise/exploration and forgetting parameters loaded negatively on PC1 (*Figure 3A*), showing that elevated decision stochasticity and the decay of learned information were associated with poorer performance in all tasks. $\alpha_+$ showed positive loadings throughout, suggesting that faster integration of positive feedback was associated with better performance in all tasks. Taken together, noise/exploration, forgetting, and $\alpha_+$ showed consistency across tasks in terms of their interpretation with respect to decision acuity. Contrary to this, $\alpha_-$ loaded positively in task C, but negatively in task B, suggesting that performance increased when participants integrated negative feedback faster in task C, but performance decreased when they did the same in task B. As mentioned before, contradictory patterns of $\alpha_-$ were likely related to task demands: The fact that negative feedback was diagnostic in task C likely favored fast integration of negative feedback, while the fact that negative feedback was not diagnostic in task B likely favored slower integration (*Figure 1E*). This interpretation is supported by behavioral findings: 'lose-stay' behavior (repeating choices that produce negative feedback) showed the same contrasting pattern as $\alpha_-$ on PC1, loading positively in task B, which shows that lose-stay behavior benefited performance, but negatively on task C, which shows that it hurt performance (*Figure 3A*). This supports the claim that lower $\alpha_-$ was beneficial in task B, while higher $\alpha_-$ was beneficial in task C, in accordance with participant behavior and developmental differences.

We next analyzed PC2 and PC3. For easier visualization, we flipped the loadings of all features with negative loadings on PC1 to remove the effects of task engagement (PC1) when interpreting subsequent PCs (for details, see section 'Principal component analysis (PCA)'). This revealed that PC2 and PC3 encoded task contrasts: PC2 contrasted task B to task C (loadings were positive / negative / near-zero for corresponding features of tasks B / C / A; *Figure 3B*). PC3 contrasted task A to both B and C (loadings were positive / negative for corresponding features on task A / tasks B and C; *Figure 3C*). (As opposed to most features of our dataset, missed trials and response times did not show these task contrasts, suggesting that these features did not differentiate between tasks). The ordering of PC2 before PC3 shows that participants' behavior differed more between task B compared to C (PC2: 8.9% explained variance) than between B and C compared to A (PC3: 6.2%; *Appendix 8—figure 4*), as expected based on task similarity (*Figure 1E*). PC2 and PC3 therefore

show that, after task engagement, the main variation in our dataset arose from behavioral differences between tasks.

How can this understanding of PC2-3 promote our understanding of model parameters? The task contrasts encoded by the main behavioral measures were also evident in several parameters, including noise/exploration parameters, $\alpha_+$, and $\alpha_-$: These parameters showed positive loadings for task B in PC2 (A in PC3), and negative loadings for task C (B and C; PC2: *Figure 3B*, PC3: 3 C). This indicates that noise/exploration parameters, $\alpha_+$, and $\alpha_-$ captured different behavioral patterns depending on the task: The variance present in these parameters allowed for the discrimination of all tasks from each other, with PC2 discriminating task B from C, and PC3 discriminating tasks B and C from A. In other words, these parameters were clearly distinguishable between tasks, showing that they did not capture the same processes. Had they captured the same processes across tasks, they would not be differentiable between tasks, similar to, for example, response times. What is more, each parameter captured sufficient task-specific variance to indicate in which task it was measured. In sum, these findings contradict the assumption that parameters are specific or interpretable in a task-independent way.

Taken together, the PCA revealed that the emerging major axes of variation in our large dataset, together capturing 40.2% of explained variance, were task engagement (PC1) and task differences (PC2-PC3). These dimensions can be employed to better understand model parameters: Task engagement / decision acuity (PC1) played a crucial role for all four parameters (*Figure 3A*), and this role was consistent across tasks for noise/exploration, forgetting, and $\alpha_+$. This consistency supports the claim that parameters captured specific, task-independent processes in terms of PC1. For $\alpha_-$, however, PC1 played inverse roles across tasks, showing a lack of task-independent specificity that was likely due to differences in task demands. Furthermore, PC2 and PC3 revealed that noise/exploration, $\alpha_+$, and $\alpha_-$ specifically encoded task contrasts, suggesting that the parameters captured different cognitive processes across tasks, lacking a task-independent core of meaning.

## Parameters and cognitive processes

Whereas the previous analysis revealed that parameter roles were not entirely consistent across tasks, it did not distinguish between parameter specificity (whether the same parameter captures the same cognitive processes across tasks) and distinctiveness (whether different parameters capture different cognitive processes).

To assess this, we probed how much parameter variance was explained by both corresponding and non-corresponding parameters across tasks: We predicted one parameter from all others to get a sense for which relationships were least and most explanatory, while accounting for all relationships of all parameters, using regression. We assumed that parameters reflected one or more cognitive processes, such that shared variance implies overlapping cognitive processes. If parameters are specific (i.e. reflect similar cognitive processes across tasks), then corresponding parameters should be predictive of each other (e.g. when predicting task B's $\frac{1}{\beta}$ from task A's parameters, task A's $\frac{1}{\beta}$ should show a significant regression coefficient). If parameters are also distinct, then non-corresponding parameters should furthermore not be predictive (e.g. no other parameters beside task A's $\frac{1}{\beta}$ should predict task B's $\frac{1}{\beta}$). We used repeated, k-fold cross-validated Ridge regression to avoid overfitting, obtaining unbiased out-of-sample estimates of the means and variances of explained variance $R^2$ and regression coefficients $w$ (for methods, see section 'Ridge regression').

Assessing general patterns that arose in this analysis, we found that all significant coefficients connected tasks A and B or tasks A and C but never tasks B and C, mirroring previous results (*Figure 2D*; section 'Relative parameter differences') with regard to task similarity (*Figure 1E*). This suggests that no parameter had a specific core that extended across all three tasks—the largest shared variance encompassed two tasks.

We first address parameter specificity. Focusing on noise/exploration parameters, coefficients were significant when predicting noise/exploration in task A from noise/exploration in tasks B or C, but the inverse was not true, such that coefficients were not significant when predicting tasks B or C from task A (*Figure 4A*; *Table 5*). The first result implies parameter specificity, showing that noise/exploration parameters captured variance (cognitive processes) in task A that they also captured in tasks B and C. The second result, however, implies a lack of specificity, showing that noise/exploration parameters captured additional cognitive processes in tasks B and C that they did not capture in task A. A further

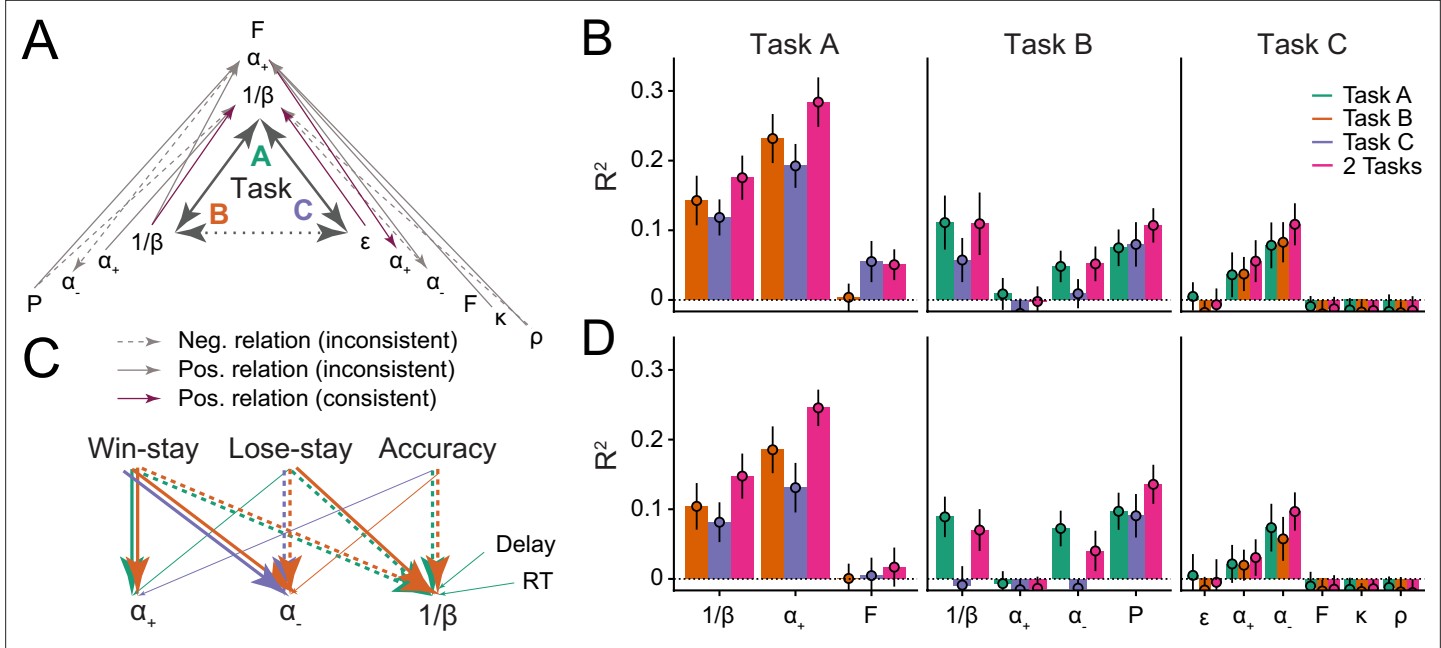

**Figure 4.** Assessing parameter interpretability by analyzing shared variance. (**A**) Parameter variance that is shared between tasks. Each arrow shows a significant regression coefficient when predicting a parameter in one task (e.g. $\alpha_+$ in task A) from all parameters of a different task (e.g. $P$, $\alpha_-$, $\alpha_+$, and $\frac{1}{\beta}$ in task B). The predicted parameter is shown at the arrow head, predictors at its tail. Full lines indicate positive regression coefficients, and are highlighted in purple when connecting two identical parameters; dotted lines indicate negative coefficients; non-significant coefficients are not shown. *Table 5* provides the full statistics of the models summarized in this figure. (**B**) Amount of variance of each parameter that was captured by parameters of other models. Each bar shows the percentage of explained variance ($R^2$) when predicting one parameter from all parameters of a different task/model, using Ridge regression. Part (**A**) of this figure shows the coefficients of these models. The x-axis shows the predicted parameter, and colors differentiate between predicting tasks. Three models were conducted to predict each parameter: One combined the parameters of both other tasks (pink), and two kept them separate (green, orange, blue). Larger amounts of explained variance (e.g., Task A $\frac{1}{\beta}$ and $\alpha_-$) suggest more shared processes between predicted and predicting parameters; the inability to predict variance (e.g. Task B $\alpha_+$; Task C working memory parameters) suggests that distinct processes were captured. Bars show mean $R^2$, averaged over $k$ data folds ($k$ was chosen for each model based on model fit, using repeated cross-validated Ridge regression; for details, see section 'Ridge regression'); error bars show standard errors of the mean across folds. (**C**) Relations between parameters and behavior. The arrows visualize Ridge regression models that predict parameters (bottom row) from behavioral features (top row) within tasks (full statistics in *Table 6*). Arrows indicate significant regression coefficients, colors denote tasks, and line types denote the sign of the coefficients, like before. All significant within-task coefficients are shown. Task-based consistency (similar relations between behaviors and parameters across tasks) occurs when arrows point from the same behavioral features to the same parameters in different tasks (i.e. parallel arrows). (**D**) Variance of each parameter that was explained by behavioral features; corresponds to the behavioral Ridge models shown in part (**C**).

lack of specificity was evident in that even the variance that both B and C captured in A was not the same: Prediction accuracy increased when combining tasks B and C to predict task A, showing that noise/exploration parameters in tasks B and C captured partly non-overlapping aspects of noise/exploration (*Figure 4B*, left-most set of bars, compare purple to orange and blue).

Focusing next on learning rates, specificity was evident in that learning rate $\alpha_+$ in task A showed a significant regression coefficient when predicting learning rates $\alpha_+$ and $\alpha_-$ in task C, and learning rate $\alpha_-$ in task C showed a significant coefficient when predicting learning rate $\alpha_+$ in task A (*Figure 4A*; *Table 5*). This suggests a shared core of cognitive processes between learning rates $\alpha_+$ and $\alpha_-$ in tasks A and C. However, a lack of specificity was evident in task B: When predicting $\alpha_+$ in task B, no parameter of any task showed a significant coefficient (including $\alpha_+$ in other tasks; *Table 5*), and it was impossible to predict variance in task B's $\alpha_+$ even when combining all parameters of the other tasks (*Figure 4B*, 'Task B' panel). This reveals that $\alpha_+$ captured fundamentally different cognitive processes in task B compared to the other tasks. The case was similar for parameter $\alpha_-$, which strikingly was inversely related between tasks A and B (*Table 5*), and impossible to predict in task B from all other parameters (*Figure 4B*). This reveals a fundamental lack of specificity, implying that learning rates in task B did not capture the same core of cognitive processes compared to other tasks.

**Table 5.** Selected coefficients of the repeated, k-fold cross-validated Ridge regression models predicting one parameter from all parameters of a different task.
The table includes all significant coefficients and selected non-significant coefficients.

| Predicted parameter (Task) | Predicting parameter (Task) | Coefficient | $p$ | sig. |
|---|---|---|---|---|
| Noise/exploration (A) | Exploration $\frac{1}{\beta}$ (B) | 0.14 | 0.031 | * |
| | $\alpha_-$ (B) | 0.14 | 0.032 | * |
| | Persistence (B) | –0.19 | 0.0029 | ** |
| | Noise $\epsilon$ (C) | 0.12 | 0.038 | * |
| | $\alpha_-$ (C) | –0.18 | 0.045 | * |
| | | –0.19 | 0.023 | * |
| Noise/exploration (B) | Noise/exploration (A) | 0.09 | 0.27 | – |
| Noise/exploration (C) | Noise/exploration (A) | 0.04 | 0.63 | – |
| $\alpha_+$ (A) | $\alpha_-$ (C) | 0.22 | 0.011 | * |
| | $\rho$ (C) | 0.16 | 0.050 | * |
| | $K$ (C) | 0.15 | 0.020 | * |
| | Exploration $\frac{1}{\beta}$ (B) | 0.19 | 0.0026 | ** |
| | $\alpha_-$ (B) | –0.21 | $< 0.001$ | *** |
| | $\alpha_+$ (B) | 0.0042 | 0.94 | – |
| | Persistence (B) | 0.23 | $< 0.001$ | *** |
| $\alpha_+$(B) | $\frac{1}{\beta}$ (A) | –0.077 | 0.37 | – |
| | $\alpha_+$ (A) | 0.058 | 0.48 | – |
| | $\alpha_+$ (C) | –0.00018 | 0.99 | – |
| | $\alpha_-$ (C) | –0.000055 | 1.00 | – |
| | Forgetting (A) | 0.015 | 0.82 | – |
| $\alpha_+$ (C) | $\alpha_+$ (A) | 0.20 | 0.013 | * |
| $\alpha_-$ (B) | $\alpha_+$ (A) | –0.25 | 0.0018 | ** |
| $\alpha_-$ (C)(C) | $\alpha_+$ (A) | 0.24 | 0.0022 | ** |

We next turned to distinctiveness, that is, whether different parameters capture different cognitive processes. Noise/exploration in task A was predicted by Persistence and $\alpha_-$ in task B, and by $\alpha_-$ and working memory weight $\rho$ in task C (*Figure 4A*; *Table 5*). This shows that processes that were captured by noise/exploration parameters in task A were captured by different parameters in other tasks, such that noise/exploration parameters did not capture distinct cognitive processes.

In the case of learning rates, $\alpha_+$ in task A was predicted nonspecifically by all parameters of task B (with the notable exception of $\alpha_+$ itself; *Figure 4A*; *Table 5*), suggesting that the cognitive processes that $\alpha_+$ captured in task A were captured by an interplay of several parameters in task B. Furthermore, task A's $\alpha_+$ was predicted by task C's working memory parameters $\rho$ and $K$ (*Figure 4A*; *Table 5*), suggesting that $\alpha_+$ captured a conglomerate of RL and working memory processes in task A that was isolated by different parameters in task C (*Collins and Frank, 2012*). In support of this interpretation, no variance in task C's working memory parameters could be explained by any other parameters (*Figure 4B*), suggesting that they captured unique working memory processes that were not captured by other parameters. Task C's RL parameters, on the other hand, could be explained by parameters in other tasks (*Figure 4B*), suggesting they captured overlapping RL processes. In tasks B and C, $\alpha_+$ and $\alpha_-$ were partly predicted by other learning rate parameters (specific and distinct), partly not

**Table 6.** Statistics of selected coefficients in the repeated, k-fold cross-validated Ridge regression models predicting each model parameter from all behavioral features of all three tasks.
The table includes all significant coefficients of within-task predictors, and a selected number of non-significant and between-task coefficients.

| Predicted parameter (Task) | Predicting parameter (Task) | coefficient | $p$ | sig. |
|---|---|---|---|---|
| Noise/exploration (A) | Win-stay (A) | –0.30 | <0.001 | *** |
| | Lose-stay (A) | –0.23 | <0.001 | *** |
| | Accuracy (A) | –0.19 | 0.0076 | ** |
| | Response times (A) | 0.092 | 0.029 | * |
| | Delay (A) | 0.25 | <0.001 | *** |
| Noise/exploration (B) | Win-stay (B) | –0.58 | <0.001 | *** |
| | Lose-stay (B) | 0.091 | 0.0034 | ** |
| | Accuracy (B) | –0.36 | <0.001 | *** |
| | Win-stay (A) | –0.12 | 0.032 | * |
| | Response times (A) | 0.059 | 0.051 | – |
| $\alpha_+$ (A) | Win-stay (A) | 0.74 | <0.001 | *** |
| $\alpha_+$ (B) | Win-stay (B) | 0.27 | <0.001 | *** |
| $\alpha_+$ (C) | Accuracy (C) | 0.24 | 0.033 | * |
| $\alpha_-$ (B) | Win-stay (B) | 0.29 | <0.001 | *** |
| | Lose-stay (B) | –0.71 | <0.001 | *** |
| | Accuracy (B) | –0.28 | <0.001 | *** |
| $\alpha_-$ (C) | Win-stay (C) | 0.16 | 0.009 | ** |
| | Lose-stay (C) | –0.41 | <0.001 | *** |

predicted at all (lack of specificity), and partly predicted by several parameters (lack of distinctiveness; *Figure 4A*).

In sum, in the case of noise/exploration, there was evidence for both specificity and a lack thereof (mutual prediction between some, but not all noise/exploration parameters). Noise/exploration parameters were also not perfectly distinct, being predicted by a small set of other parameters from different tasks. In the case of learning rates, some specificity was evident in the shared variance between tasks A and C, but that specificity was missing in task B. Distinctiveness was particularly low for learning rates, with variance shared widely between multiple different parameters. When conducting the same analyses in simulated agents using the same parameters across tasks (section 'Statistical comparison to generalizability ceiling'), we obtained much higher specificity and distinctiveness.

## Parameters and behavior

The previous sections suggested that parameters captured different cognitive processes across tasks (i.e. different internal characteristics of learning and choice). We lastly examined whether parameters also captured different behavioral features across tasks (e.g. tendency to stay after positive feedback), and whether behavioral features generalized better. To investigate this question, we assessed the relationships between model parameters and behavioral features across tasks, using regularized Ridge regression as before, and predicting each model parameter from each task's behavioral features (15 predictors, see 'Appendix 1' and 'Appendix '6; for regression methods, see section 'Ridge regression').

We found that noise/exploration parameters were predicted by the same behavioral features in tasks A and B, such that task A's accuracy, win-stay, and lose-stay behavior predicted task A's $\frac{1}{\beta}$; and task B's accuracy, win-stay, and lose-stay behavior predicted task B's $\frac{1}{\beta}$ (*Figure 4C*; *Table 6*). This shows consistency in terms of which (task-specific) behaviors were related to (task-specific) parameter $\frac{1}{\beta}$. Similarly for learning rates, $\alpha_+$ was predicted by the same behavior (win-stay) in tasks A and

B, and $\alpha_-$ was predicted by the same behaviors (lose-stay, win-stay) in tasks B and C (*Figure 4C*; *Table 6*). This consistency in $\alpha_-$ is especially noteworthy given the pronounced lack of consistency in the previous analyses.

In sum, noise/exploration parameters, $\alpha_+$, and $\alpha_-$ successfully generalized between tasks in terms of which behaviors they reflected (*Figure 4C*), despite the fact that many of the same parameters did not generalize in terms of how they characterized participants (sections 'Differences in absolute parameter values', 'Relative parameter differences', and 'Parameter age trajectories'), and which cognitive processes they captured (sections 'Main axes of variation and parameters and cognitive processes'). Notably, the behavioral and parameter differences we observed between tasks often seemed tuned to specific task characteristics (*Figure 1E*), both in the case of parameters (most notably $\alpha_-$; *Figures 2C and 3A*) and behavior (most notably lose-stay behavior; *Appendix 3—figure 1B*), suggesting that both behavioral responses and model parameters were shaped by task characteristics. This suggests a succinct explanation for why parameters did not generalize between tasks: Because different tasks elicited different behaviors (*Appendix 3—figure 1B*), and because each behavior was captured by the same parameter across tasks (*Figure 4C*), parameters necessarily differed between tasks.

## Discussion

Both generalizability (*Nassar and Frank, 2016*) and interpretability (i.e. the inherent 'meaningfulness' of parameters *Huys et al., 2016*) have been stated as advantages of computational modeling, and many current research practices (e.g. comparing parameter-specific findings between studies) endorse them (*Eckstein et al., 2021*). However, RL model generalizability and interpretability has so far eluded investigation, and growing inconsistencies in the literature potentially cast doubt on these assumptions. It is hence unclear whether, to what degree, and under which circumstances we should assume generalizability and interpretability. Our developmental, within-participant study revealed that these assumptions warrant both increased scepticism and continued investigation: Generalizability and interpretability were suprisingly low for most parameters and tasks, but reassuringly high for a few others:

Exploration/noise parameters showed considerable generalizability in the form of correlated variance and age trajectories. Furthermore, the decline in exploration/noise we observed between ages 8–17 was consistent with previous studies (*Nussenbaum and Hartley, 2019*; *Somerville et al., 2017*; *Gopnik, 2020*), revealing consistency across tasks, models, and research groups that supports the generalizability of exploration/noise parameters. Still, for 2/3 pairs of tasks, the degree of generalization was significantly below the level of generalization expected by agents with perfect generalization.

Interpretability of exploration/noise parameters was mixed: Despite evidence for specificity in some cases (overlap in parameter variance between tasks), it was missing in others (lack of overlap), and crucially, parameters lacked distinctiveness (substantial overlap in variance with other parameters). Thus, while exploration/noise parameters were generalizable across tasks, they were not neurocognitively "interpretable" (as defined above).

Learning rate from negative feedback showed a substantial lack of generalizability: parameters were less consistent within participants than within tasks, and age trajectories differed both quantitatively and qualitatively. Learning rates from positive feedback, however, showed some convincing patterns of generalization. These results are consistent with the previous literature, which shows mixed results for learning rate parameters (*Nussenbaum and Hartley, 2019*). In terms of interpretability, learning rates from positive and negative feedback combined were somewhat specific (overlap in variance between some tasks). However, a lack of specificity (lack of shared core variance) and distinctiveness (fundamental entangling with several other parameters, most notably working memory parameters) overshadowed this result.

Taken together, our study confirms the patterns of generalizable exploration/noise parameters and task-specific learning rate parameters that are emerging from the literature (*Nussenbaum and Hartley, 2019*). Furthermore, we show that this is not a result of between-participant comparisons, but that the same participants will show different parameters when tested using different tasks. The inconsistency of learning rate parameters leads to the important conclusion that we cannot measure an individual's 'intrinsic learning rate', and that we should not draw general conclusions about 'the development of learning rates' with the implication that they apply to all contexts.

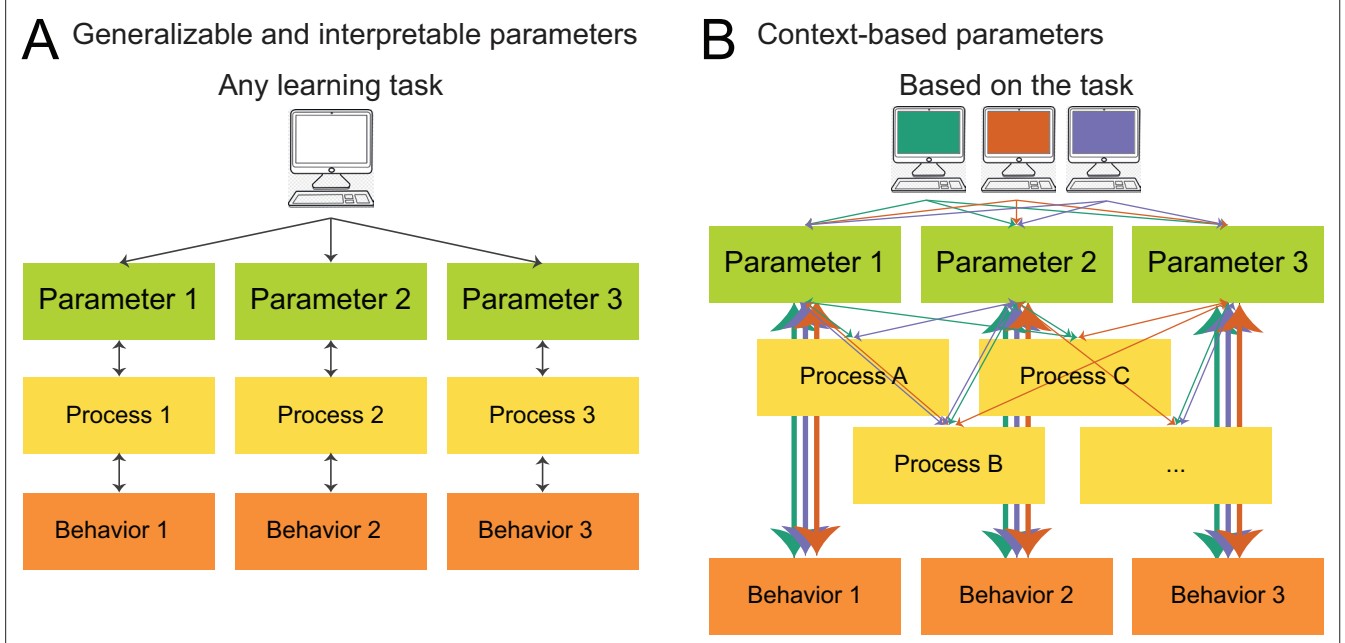

**Figure 5.** What do model parameters measure? (**A**) View based on generalizability and interpretability. In this view, which is implicitly endorsed by much current computational modeling research, models are fitted in order to reveal individuals' intrinsic model parameters, which reflect clearly delineated, separable, and meaningful (neuro)cognitive processes, a concept we call *interpretability*. Interpretability is the assumption that every model parameter captures a specific cognitive process (bidirectional arrows between each parameter and process), and that cognitive processes are separable from each other (no connections between processes). Task characteristics are treated as irrelevant, a concept we call *generalizability*, such that parameters of any learning task (within reason) are expected to capture similar cognitive processes. (**B**) Updated view, based on our results, that acknowledges the role of context (e.g. task characteristics, model parameterization, participant sample) in computational modeling. Which cognitive processes are captured by each model parameter is influenced by context (green, orange, blue), as shown by distinct connections between parameters and cognitive processes. Different parameters within the same task can capture overlapping cognitive processes (not interpretable), and the same parameters can capture different processes depending on the task (not generalizable). However, parameters likely capture consistent behavioral features across tasks (thick vertical arrows).

These findings help clarify the source of parameter inconsistencies in previous literature (besides replication problems and technical issues, such as model misspecification [*Nussenbaum and Hartley, 2019*], lack of model comparison and validation [*Palminteri et al., 2017*; *Wilson and Collins, 2019*], lack of model critique [*Nassar and Frank, 2016*], inappropriate fitting methods [*Daw, 2011*; *Lee, 2011*], and lack of parameter reliability [*Brown et al., 2020*]): Our results show that discrepancies are expected even with a consistent methodological pipeline and up-to-date modeling techniques, because they are an expected consequence of variations in *context* (e.g. features of the experimental task [section Parameters and behavior] and the computational model). The results also suggest that the mapping between cognitive processes and exhibited behavior is many-to-many, such that different cognitive mechanisms (e.g. reasoning, value learning, episodic memory) can give rise to the same behaviors (e.g. lose-stay behavior) and parameters (e.g. $\alpha_-$), while the same cognitive mechanism (e.g. value learning) can give rise to different behaviors (e.g. win-stay, lose-shift) and influence several parameters (e.g. $\alpha_+$, $\alpha_-$), depending on context factors. Under this view, analysis of model parameters alone does not permit unequivocal conclusions about cognitive processes if the context varies (*Figure 5B*), and the interpretation of model parameter results requires careful contextualization.

Future research needs to investigate context factors to characterize these issues in more detail. For example, which task characteristics determine which parameters will generalize and which will not, and to what extent? Does context impact whether parameters capture overlapping versus distinct variance? Here, the systematic investigation of task space (i.e., testing the same participants on a large battery of learning tasks created as the full factorial of all task features) could elucidate the relationships between parameter generalizability and task-based context factors (e.g., stochasticity, volatility,

reward probability). To determine the distance between tasks, the MDP framework might be especially useful because it decomposes tasks along theoretically meaningful features. Future research will also need to determine the relative contributions of different sources of inconsistency, differentiating those caused by technical issues from those caused by context differences.

In sum, our results suggest that relating model parameters to cognitive constructs and real-world behavior might require us to carefully account for task variables and environmental variability in general. This ties into a broader open question of how neurocognitive processes are shared between tasks (*Eisenberg et al., 2019*; *Moutoussis et al., 2021*), and reflects a larger pattern of thought in psychology that we cannot objectively assess an individual's cognitive processing while ignoring context. We have shown that in lab studies, different task contexts recruit different system settings within an individual; similarly, our real-life surroundings, the way they change during development, and our past environments (*Lin et al., 2020*; *Lin et al., 2022*) may also modulate which cognitive processes we recruit.

## Limitations

Our study faces several potential limitations, both due to the technical aspects of model creation and selection, and to the broader issue of parameter reliability. One potential technical limitation is the existence of within-model parameter correlations. These correlations may mean the values of parameters of the same model trade off during fitting, potentially leading to lower parameter correlations between models, and decreased estimates of parameter generalizability. However, this limitation is unlikely to affect our overall conclusion: Our simulation analysis showed that generalization was detectable despite this issue (section 'Statistical comparison to generalizability ceiling'), suggesting that we would have been able to detect more generalization in humans if it had been present. Furthermore, the majority of previous work using computational models to study human behavior is subject to the same within-model parameter tradeoffs (e.g. common negative correlation between $\alpha$ and $\beta$ in RL models), meaning that the results of our study likely give a realistic estimate of expected parameter generalization in the current literature.

Another limitation relates to the potential effects of model misspecification on our results. An example of model misspecification is the failure to include a variable in the model that was relevant in the data-generating process (e.g. outcome-independent choice persistence); such misspecification can lead to the inaccurate estimation of other parameters in the model (e.g. learning rate [*Katahira, 2018*]). In our study, model misspecification—if present—could account for some of the lack of generalization we observed. As for the previous limitation, however, the fact that model misspecification is likely a ubiquitous feature of the current modeling literature (and potentially fundamentally unattainable when fitting complex data-generating processes such as human decision makers) means that our results likely provide a realistic picture of the generalizability of current models.

Another potential limitation is the difference between the models for each task, despite shared or overlapping cognitive processes. It is possible, for example, that parameters would generalize better if the same model had been used across tasks. The current dataset, however, is not suitable to answer this question: It would be impossible to fit the same model to each task due to issues of model misspecification (when using a model that is too simple) or violation of the principle of simplicity (when using a model that is too complex; for details, see 'Appendix 7'). Future research will be required to address this issue, and to potentially dissociate the effects of model differences and task differences we here jointly call 'context'.

Lastly, model parameter reliability might play a crucial role for our results: If parameters lack consistency between two instantiations of the same task (reliability), generalization between different tasks would necessarily be low as well. A recent wave of research, however, has convincingly demonstrated that good reliability is possible for several common RL models (*Brown et al., 2020*; *Shahar et al., 2019*; *Pratt et al., 2021*; *Waltmann et al., 2022*), and we employ the recommended methods here (*Xia et al., 2021*; *Eckstein et al., 2022*). In addition, our simulation analysis shows that our approach can detect generalization.

In conclusion, a variety of methodological issues could explain (part of) the lack of generalization we find for most parameters in the human sample. However, these issues cannot explain all of our results because the same approach successfully detects generalization in a simulated dataset. Furthermore, none of these issues are unique to our approach, but likely ubiquitous in the current modeling

literature. This means that our results likely provide a realistic estimate of parameter generalization based on current methods. A more detailed discussion of each limitation is provided in 'Appendix 7'.

## Moving forward

With this research, we do not intend to undermine RL modeling as a practice, or challenge pre-existing findings drawn from it, but to improve its quality. Computational model parameters potentially provide highly valuable insights into (neuro)cognitive processing—we just need to refrain from assuming that the identified processes are always—through their mere nature as model parameters—specific, distinct, and 'theoretically meaningful' (*Huys et al., 2016*). Some parameters with the same names do not tend to transfer between different tasks, making them non-interchangeable, while others seem to transfer well. And in all cases, the behavioral features captured by parameters seem to generalize well. In the long term, we need to understand why RL parameters differ between tasks. We suggest three potential, not mutually exclusive answers:

1. **Adaptation and Optimality.** Variance in RL parameters may reflect how participants adapt their behavior to task demands, an explanation proposed by *Nussenbaum and Hartley, 2019*. Whereas it is commonly assumed that parameters reflect participants' intrinsic cognitive 'settings' (e.g. 10-year-olds have a learning rate of 20%; 16-year-olds of 40%), the optimality-based view suggests that participants instead adaptively tune parameters to task characteristics (e.g. adopting lower learning rates in stable than volatile contexts [*Behrens et al., 2007*; *Nassar et al., 2016*]). Hence, different tasks lead to different parameter estimates because different values are required for optimal behavior; an 'optimal' participant—achieving optimal behavior in each task—would therefore naturally show different values across tasks. Similar optimality-based views are held by others (*McGuire et al., 2014*). If adaptation to achieve optimality exists, then we would also predict, for example, that learning rates differ between deterministic and stochastic tasks because each task requires different amounts of behavioral change in response to feedback to reach optimal performance. We indeed observed this pattern in the current study. Age differences in parameters can be explained as differences in adaptation flexibility and/or differences in optimal settings due to interaction with different environments. Participants might require differing levels of change detection or adaptation abilities, depending on their developmental stage (e.g. adolescent cognition may be better adapted to changing environments). More research is needed, however, to determine whether parameter optimality and the capacity to optimize behavior can explain all inconsistencies in the literature. For example, our finding that participants showed the most optimal learning rates in the intermediate age range in task B (*Eckstein et al., 2022*), whereas optimality increased monotonously with age in tasks A and C (*Master et al., 2020*; *Xia et al., 2021*), suggests that other factors besides optimization might play a role as well.

2. **Modulatory processes.** RL parameters may vary as a function of modulatory processes that are not well-captured in current RL models. Modulatory processes have been described in cognition and neurobiology and likely serve to shift functional outputs (e.g. hunger increasing motivation [*Berridge, 2007*; *Yu and Dayan, 2005*; *Bouret and Sara, 2005*]). Some modulatory processes reflect external contextual information (e.g. uncertainty affects dopamine neuron firing [*Gershman, 2017*; *Starkweather et al., 2018*; *Gershman and Uchida, 2019*]), and RL processes might depend on these same modulatory processes (e.g. RL reward-prediction errors and dopamine [*Schultz et al., 1997*]). Indeed, environments with different degrees of uncertainty have been shown to elicit different learning rates (*Behrens et al., 2007*; *Lin et al., 2022*), and EEG markers of neuromodulator release predicted learning rates (*Jepma et al., 2016*). It is thus possible that neuromodulation by task uncertainty modulates RL processes, reflected in RL parameters. In our data, feedback stochasticity and task volatility likely contribute to uncertainty-related modulation. However, other factors like task similarity, task volatility (*Behrens et al., 2007*; *Eckstein et al., 2022*; *Nassar et al., 2016*), feedback stochasticity, memory load (*Master et al., 2020*; *Collins and Frank, 2012*), feedback valence and conditioning type (*Garrison et al., 2013*), and choice of model parameters (e.g. forgetting [*Master et al., 2020*; *Xia et al., 2021*]), counter-factual learning (*Eckstein et al., 2022*), negative and positive learning rates (*Harada, 2020*; *Katahira, 2018*; *Sugawara and Katahira, 2021*), have also been shown to affect RL parameters, but are independent of uncertainty. More research is therefore needed to investigate the extent of the contribution of modulatory processes, and its impact on cognition and computation.

3. **RL processes are multifaceted.** RL parameters capture a multitude of cognitive processes, whose composition likely differs across tasks (*Figure 5B*; *Eckstein et al., 2021*). RL algorithms

are framed in the most general way to allow application to a wide range of contexts, including AI, neuroscience, and psychology (*Sutton and Barto, 2017*; *Eckstein et al., 2021*; *Lake et al., 2017*; *Collins, 2019*). As behavioral models, their use has spanned a variety of behaviors, meaning that the same parameters capture cognitive processes that vary considerably in type and complexity: For example, the same RL parameters have been said to capture the slow acquisition of implicit preferences (*Schultz et al., 1997*), long-term memory for preferences (*Collins, 2018*), quick recognition of contingency switches (*Eckstein et al., 2022*; *Tai et al., 2012*), selection of abstract high-level strategies (*Eckstein and Collins, 2020*; *Collins and Koechlin, 2012*; *Donoso et al., 2014*), meta-learning (*Wang et al., 2018*), habitual and goal-directed decision making (*Daw et al., 2011*), working memory or episodic memory-guided choice (*Collins and Frank, 2012*; *Bornstein and Norman, 2017*; *Vikbladh et al., 2019*), and many others. This list of cognitive processes outnumbers the list of typical RL model parameters, suggesting that RL parameters necessarily capture different (combinations of) cognitive processes depending on the paradigm. Indeed, adaptive learning does not seem to be a unitary phenomenon, but seems to be composed of several distinct neuro-cognitive factors (*McGuire et al., 2014*).

## Conclusion

Our research has important implications for computational modeling in general, and specifically for fields that focus on individual differences, including developmental and clinical computational research: We show that contextual factors critically impact computational modeling results. Larger, targeted studies will be necessary to identify the most important contextual factors and their precise roles, and to quantify their effects. Other areas of modeling besides RL might face similar issues, given that generalizability and interpretability are also commonly assumed in models of sequential sampling (*Sendhilnathan et al., 2020*; *McDougle and Collins, 2021*), Bayesian inference (*Eckstein et al., 2022*; *Radulescu et al., 2019*; *Konovalov and Krajbich, 2018*), model-based versus model-free RL (*Brown et al., 2020*; *Kool et al., 2016*; *Hare, 2020*), and others.

If a model parameter lacks generalizability and/or interpretability, it does not measure task-independent, person-specific characteristics, as we often assume. This parameter is more closely tied to the specific, contextual factors of experimental paradigms, and should be interpreted within the context of that task, and only compared between studies with the clear understanding of this task-dependence. We hope that acknowledging this will help the field of computational modeling to accurately interpret computational models (in direct relation to the experimental task), to combine insights of different studies (by taking into account differences in parameter optimality, modulatory factors, and processes captured by each parameter), and to achieve improved generalizability and interpretability of findings in the future. This work aims not to discourage the use of RL models to model behavior, but to improve the application of these models, in particular the robustness of the conclusions we draw from their fits.

## Materials and methods
### Study design

Our sample of 291 participants was balanced between females and males, and all ages (8–30 years) were represented equally (*Figure 1A*, left). Participants completed four computerized tasks, questionnaires, a saliva sample during the 1–2 hr lab visit, and another take-home sample (see section 'Testing procedure'). To reduce noise, we excluded participants based on task-specific performance criteria (see section 'Participant sample'). Due to worse performance, more younger than older participants were excluded, which is a caveat for the interpretation of age effects (note, however, that these exclusions cannot account for the observed age effects but act against them; *Figure 1A*). Our tasks—A ('Butterfly task' *Xia et al., 2021*; *Davidow et al., 2016*), B ('Stochastic Reversal' *Tai et al., 2012*; *Eckstein et al., 2022*), and C ('Reinforcement learning-Working memory' *Master et al., 2020*; *Collins and Frank, 2012*)—were all classic reinforcement learning tasks: on each trial, participants chose between several actions in an effort to earn rewards, which were presented as binary feedback (win/point or lose/no point) after each choice.

The tasks varied on several common dimensions (*Figure 1E*), which have been related to discrepancies in behavioral and neurocognitive results in the literature (*Garrison et al., 2013*; *Yaple and Yu, 2019*; *Liu et al., 2016*). For example, in one task (task C), positive feedback was deterministic, such

that every correct action led to a positive outcome, whereas in the two other tasks (tasks A and B), positive feedback was stochastic, such that some correct actions led to positive and others to negative outcomes. A different set of two tasks (B and C) provided diagnostic positive feedback, such that every positive outcome indicated a correct action, whereas in the third (A), positive feedback was non-diagnostic, such that positive outcomes could indicate both correct and incorrect actions. Two tasks (A and C) presented several different stimuli/states for which correct actions had to be learned, whereas the third (B) only presented a single one. Overall, task A shared more similarities with both tasks B and C than either of these shared with each other, allowing us to ask the exploratory question whether task similarity played a role in parameter generalizability and interpretability. A comprehensive list of task differences is shown in *Figure 1E*, and each task is described in more detail in section 'Task design'. Section 'Appendix 3' explains the most prominent findings of each task individually, and shows several behavioral measures over age.

## Participant sample

### Sample overview

All procedures were approved by the Committee for the Protection of Human Subjects at the University of California, Berkeley, with reference number 2016-06-8925. We tested 312 participants: 191 children and adolescents (ages 8–17) and 55 adults (ages 25–30) were recruited from the community and completed a battery of computerized tasks, questionnaires, and saliva samples; 66 university undergraduate students (aged 18–50) completed the four tasks as well, but not the questionnaires or saliva sample. Community participants of all ages were pre-screened for the absence of present or past psychological and neurological disorders; the undergraduate sample indicated the absence of these. Compensation for community participants consisted of $25 for the 1–2 hr in-lab portion of the experiment and $25 for completing optional take-home saliva samples; undergraduate students received course credit for participation in the 1-hr study.

### Participant exclusion

Two participants from the undergraduate sample were excluded because they were older than 30, and 7 were excluded because they failed to indicate their age. This led to a sample of 191 community participants under 18, 57 undergraduate participants between the ages of 18–28, and 55 community participants between the ages of 25–30. Of the 191 participants under 18, 184 completed task B, and 187 completed tasks A and C. Reasons for not completing a task included getting tired, running out of time, and technical issues. All 57 undergraduate participants completed tasks B and C and 55 completed task A. All 55 community adults completed tasks B and A, and 45 completed task C. Appropriate exclusion criteria were implemented separately for each task to exclude participants who failed to pay attention and who performed critically worse than the remaining sample (for task A, see *Xia et al., 2021*; task B *Eckstein et al., 2022*; task C *Master et al., 2020*). Based on these criteria, 5 participants under the age of 18 were excluded from task B, 10 from task A, and none from task C. One community adult participant was excluded from task A, but no adult undergraduates or community participants were excluded from tasks B or C.

Because this study related the results from all three tasks, we only included participants who were not excluded in any task, leading to a final sample of 143 participants under the age of 18 (male: 77; female: 66), 51 undergraduate participants (male: 17; female: 34), and 53 adults from the community (male: 25; female: 28), for a total of 247 participants (male: 119; female: 128). We excluded the fourth task of our study from the current analysis, which was modeled after a rodent task and used in humans for the first time (*Johnson and Wilbrecht, 2011*), because the applied performance criterion led to the exclusion of the majority of participants under 18. We split participants into quantiles based on age, which were calculated separately within each sex (for details, see *Eckstein et al., 2022*).

### Testing procedure

After entering the testing room, participants under 18 years and their guardians provided informed assent and permission; participants over 18 provided informed consent. Guardians and participants over 18 filled out a demographic form. Participants were led into a quiet testing room in view of their guardians, where they used a video game controller to complete four computerized tasks, in the following order: The first task was called '4-choice' and assessed reversal learning in an environment

with 4 different choice options, with a duration of approximately 5 min (designed after *Johnson and Wilbrecht, 2011*). This task was excluded from the current analysis (see section 'Participant exclusion'). The second task was C ('Reinforcement learning-Working memory') and took about 25 min to complete (*Collins and Frank, 2012*; *Master et al., 2020*). After the second task, participants between the ages of 8–17 provided a saliva sample (for details, see *Master et al., 2020*) and took a snack break (5–10 min). After that, participants completed task A ('Butterfly task'), which took about 15 min (*Davidow et al., 2016*; *Xia et al., 2021*), and task B ('Stochastic Reversal'), which took about 10 min to complete (*Eckstein et al., 2022*). At the conclusion of the tasks, participants between 11 and 18 completed the Pubertal Development Scale (PDS *Petersen et al., 1988*) and were measured in height and weight. Participants were then compensated with $25 Amazon gift cards. For subjects under 11, their guardians completed the PDS on their behalf. The PDS questionnaire and saliva samples were administered to investigate the role of pubertal maturation on learning and decision making. Pubertal analyses are not the focus of the current study and will be or have been reported elsewhere (*Master et al., 2020*; *Eckstein et al., 2022*; *Xia et al., 2021*). For methodological details, refer to *Master et al., 2020*. The entire lab visit took 60–120 min, depending on the participant, and the order of procedures was the same for all subjects.

## Task design

### Task A ('butterfly task')

The goal of task A was to collect as many points as possible, by guessing correctly which of two flowers was associated with each of four butterflies. Participants were instructed to guess which flower each butterfly liked more, having been told that butterflies would sometimes also choose the less-liked flower (i.e. act probabilistically). Correct guesses were rewarded with 70% probability, and incorrect guesses with 30%. The task contained 120 trials (30 for each butterfly) that were split into 4 equal-sized blocks, and took between 10 and 20 min to complete. More detailed information about methods and results can be found in *Xia et al., 2021*.

### Task B ('stochastic reversal')

The goal of task B was to collect golden coins, which were hidden in two green boxes. Participants completed a child-friendly tutorial, in which they were instructed to help a leprechaun find his treasure by collecting individual coins from two boxes. Task volatility (i.e. boxes switching sides) and stochasticity (i.e. correct box not rewarded each time) were introduced one-by-one (for details, see *Eckstein et al., 2022*). The task could be in one of two states: 'Left box is correct' or 'Right box is correct'. In the former, selecting the left box led to reward in 75% of trials, while selecting the right box never led to a reward (0%). Several times throughout the task, task contingencies changed unpredictably and without notice (after participants had reached a performance criterion indicating they had learned the current state), and the task switched states. Participants completed 120 trials of this task (2–9 reversals), which took approximately 5–15 min. For more information and additional task details, refer to *Eckstein et al., 2022*.

### Task C ('reinforcement learning-working memory')

The goal of task C was to collect as many points as possible by pressing the correct key for each stimulus. Participants were instructed to learn an 'alien language' of key presses by associating individual pictures with specific key presses. Pressing the correct key for a specific stimulus deterministically led to reward, and the correct key for a stimulus never changed. Different blocks required subjects to learn about different numbers of stimuli, with set sizes ranging from 2 to 5 images. In each block, each stimulus was presented 12–14 times, for a total of 13 * set size trials per block. Three blocks had set sizes of 2–3, and 2 blocks had set sizes of 4–5, for a total of 10 blocks. The task took between 15 and 25 minutes to complete. For more details, as well as a full analysis of this dataset, refer to *Master et al., 2020*.

## Computational models

For all tasks, we used RL theory to model how participants adapted their behavior in order to maximize reward. RL models assume that agents learn a policy $\pi(a|s)$ that determines (probabilistically) which action $a$ to take in each state $s$ of the world (*Sutton and Barto, 2017*). Here and in most

cognitive RL models, this policy is based on action values $Q(a|s)$, that is, the values of each action $a$ in each state $s$. Agents learn action values by observing the reward outcomes, $r_t$, of their actions at each time step $t$. Learning consists of updating existing action values $Q_t(a|s)$ using the 'reward prediction error', the difference between the expected reward $Q_t(a|s)$ and the actual reward $r_t$:

$$Q_{t+1}(a|s) = Q_t(a|s) + \alpha(r_t - Q_t(a|s))$$

How much a learner weighs past action value estimates relative to new outcomes is determined by parameter $\alpha$, the learning rate. Small learning rates favor past experience and lead to stable learning over long time horizons, while large learning rates favor new outcomes and allow for faster and more flexible changes according to shorter time horizons. With enough time and in a stable environment, the RL updating scheme guarantees that value estimates will reflect the environment's true reward probabilities, and thereby allow for optimal long-term choices (*Sutton and Barto, 2017*).

In order to choose actions, most cognitive RL models use a (noisy) 'softmax' function to translate action values $Q(a|s)$ into policies $p(a|s)$:

$$p(a_i|s) = \frac{exp(\beta\ Q(a_i|s))}{\sum_{a_j \in A} exp(\beta\ Q(a_j|s))}$$

$A$ refers to the set of all available actions (tasks A and B have 2 actions, task C has 3), and $a_i$ and $a_j$ to individual actions within the set. How deterministically versus noisily this translation is executed is determined by exploration parameter $\beta$, also called inverse decision temperature, and/or $\epsilon$, the decision noise (see below). Small decision temperatures $\frac{1}{\beta}$ favor the selection of the highest-valued actions, biasing an agent towards exploitation, whereas large decision temperatures select actions of low and high values more evenly, enabling exploration. Parameter $\epsilon$ adds undirected noise to action selection, selecting random actions with a small probability $\epsilon$ on each trial.

Besides $\alpha$, $\beta$, and noise, cognitive RL models often include additional parameters to better fit empirical behavior in humans or animals. Common choices include Forgetting—a consistent decay of action values back to baseline—, and Persistence—the tendency to repeat the same action independent of outcomes, a parameter also known as sticky choice or perseverance (*Sugawara and Katahira, 2021*). In addition, cognitive models often differentiate learning from positive versus negative rewards, splitting learning rate $\alpha$ into two separate parameters $\alpha_+$ and $\alpha_-$, which are applied to only positive and only negative outcomes, respectively (*Harada, 2020*; *Javadi et al., 2014*; *Christakou et al., 2013*; *van den Bos et al., 2012*; *Frank et al., 2004*; *Cazé and van der Meer, 2013*; *Palminteri et al., 2016*; *Lefebvre et al., 2017*; *Dabney et al., 2020*). The next paragraphs introduce these parameters in detail.

In task A, the best fitting model included a forgetting mechanism, which was implemented as a decay in Q-values applied to all action values of the three stimuli (butterflies) that were not shown on the current trial:

$$Q_{t+1}(a|s) = (1 - f) * Q_{t+1}(a|s) + f * 0.5.$$

The free parameter $0 < 1$ reflects individuals' tendencies to forget.

In task B, free parameter $P$ captured choice persistence, which biased choices on the subsequent trial toward staying ($P > 0$) or switching ($P < 0$). $P$ modifies action values $Q(a|s)$ into $Q'(a|s)$, as follows:

$$Q'_t(a|s) = Q_t(a|s) + P \ \ if \ \ a_t = a_{t-1}$$

$$Q'_t(a|s) = Q_t(a|s) \ \ if \ \ a_t \neq a_{t-1}$$

In addition, the model of task B included counter-factual learning parameters $\alpha_{C+}$ and $\alpha_{C-}$, which added counter-factual updates based on the inverse outcome and affected the non-chosen action. For example, after receiving a positive outcome ($r = 1$) for choosing left ($a$), counter-factual updating would lead to an 'imaginary' negative outcome ($\bar{r} = 0$) for choosing right ($\bar{a}$).

$$Q_{t+1}(\bar{a}|s) = Q_t(\bar{a}|s) + \alpha_{C-}(\bar{r} - Q_t(\bar{a}|s)) \ \ if \ \ r_t = 0$$

$$Q_{t+1}(\bar{a}|s) = Q_t(\bar{a}|s) + \alpha_{C+}(\bar{r} - Q_t(\bar{a}|s)) \ \ if \ \ r_t = 1$$

$\bar{a}$ indicates the non-chosen action, and $\bar{r}$ indicates the inverse of the received outcome, $\bar{r} = 1 - r$. The best model fits were achieved with $\alpha_{C+} = \alpha_+$ and $\alpha_{C-} = \alpha_-$, so counter-factual learning rates are not reported in this paper.

In tasks A and B, positive and negative learning rates are differentiated in the following way:

$$Q_{t+1}(a|s) = Q_t(a|s) + \alpha_+(r_t - Q_t(a|s)) \ \ if \ \ r_t = 1$$
$$Q_{t+1}(a|s) = Q_t(a|s) + \alpha_-(r_t - Q_t(a|s)) \ \ if \ \ r_t = 0$$

In the best model for task A, only $\alpha_+$ was a free parameter, while $\alpha_-$ was fixed to 0. In task C, $\alpha_-$ was a function of $\alpha_+$, such that $\alpha_- = b * \alpha_+$, where $b$ is the neglect bias parameter that determines how much negative feedback is neglected compared to positive feedback. Throughout the paper, we report $\alpha_- = b * \alpha_+$ for task C.

In addition to an RL module, the model of task C included a working memory module with perfect recall of recent outcomes, but fast forgetting and strict capacity limitations. Perfect recall was modeled as an RL process with learning rate $\alpha_{WM+} = 1$ that operated on working-memory weights $W(a|s)$ rather than action values. On trials with positive outcomes ($r = 1$), the model reduces to:

$$W_{t+1}(a|s) = r_t$$

On trials with negative outcomes ($r = 0$), multiplying $\alpha_{WM+} = 1$ with the neglect bias $b$ leads to potentially less-than perfect memory:

$$W_{t+1}(a|s) = W_t(a|s) + b * (r_t - W_t(a|s))$$

Working-memory weights $W(a|s)$ were transformed into action policies $p_{WM}(a|s)$ in a similar way as RL weights $Q(a|s)$ were transformed into action probabilities $p_{RL}(a|s)$, using a softmax transform combined with undirected noise:

$$p(a_i|s) = (1 - \epsilon) * \frac{exp(\beta \ Q(a_i|s))}{\sum_{a_j \in a} exp(\beta \ Q(a_j|s))} + \epsilon * \frac{1}{|a|}$$

$|a| = 3$ is the number of available actions and $\frac{1}{|a|}$ is the uniform policy over these actions; $\epsilon$ is the undirected noise parameter.

Forgetting was implemented as a decay in working-memory weights $W(a|s)$ (but not RL Q-values):

$$W_{t+1}(a|s)_{t+1} = (1 - f) * W_t(a|s)_t + f * \frac{1}{3}$$

Capacity limitations on working memory were modeled as an adjustment in the weight $w$ of $p_{WM}(a|s)$ compared to $p_{RL}(a|s)$ in the final calculation of action probabilities $p(a|s)$:

$$w = \rho * (min(1, \frac{K}{ns}))$$
$$p(a|s) = w * p_{WM}(a|s) + (1 - w) * p_{RL}(a|s)$$

The free parameter $\rho$ is the probability of using values stored in working memory to choose an action (relative to RL), $ns$ indicates a block's stimulus set size, and $K$ captures individual differences in working memory capacity.

We fitted a separate RL model to each task, using state-of-the-art methods for model construction, fitting, and validation (**Wilson and Collins, 2019**; **Palminteri et al., 2017**). Models for tasks A and B were fitted using hierarchical Bayesian methods with Markov-Chain Monte-Carlo sampling, which is an improved method compared to maximum likelihood that leads to better parameter recovery, amongst other advantages (**Gelman et al., 2013**; **Katahira, 2016**; **Watanabe, 2013**). The model for task C was fitted using classic non-hierarchical maximum-likelihood because model parameter $K$ is discrete, which renders hierarchical sampling less tractable. In all cases, we verified that the model parameters were recoverable by the selected model-fitting procedure, and that the models were identifiable. Details of model-fitting procedures are provided in the original publications (**Master et al., 2020**; **Eckstein et al., 2022**; **Xia et al., 2021**).

For additional details on any of these models, as well as detailed model comparison and validation, the reader is referred to the original publications (task A: **Xia et al., 2021**; task B: **Eckstein et al., 2022**; task C: **Master et al., 2020**).

## Principal component analysis (PCA)

The PCA in section Main axes of variation included 15 model parameters ($\alpha_+$ and noise/exploration in each task; Forgetting and $\alpha_-$ in two tasks; Persistence in task B; four working memory parameters in task C; see section 'Computational models') and 39 model-free features, including simple behavioral features (e.g. overall performance, reaction times, tendency to switch), results of behavioral regression models (e.g. effect of delay between presentations of the same stimulus on accuracy), and the model parameters of an alternative Bayesian inference model in task B. All behavioral features, including their development over age, are described in detail in Appendix 6 and *Appendix 3—figure 1B*. For simplicity, section Main axes of variation focused on the first three PCs only; the weights, explained variance, and age trajectories of remaining PCs are shown in *Appendix 8—figure 4*.

PCA is a statistical tool that decomposes the variance of a dataset into so-called 'principal components' (PCs; *Abdi and Williams, 2010*). PCs are linear combinations of a dataset's original features (e.g. response times, accuracy, learning rates), and explain the same variance in the dataset as the original features. The advantage of PCs is that they are orthogonal to each other and therefore capture independent aspects of the data. In addition, subsequent PCs explain subsequently less variance, such that selecting just the top PCs of a dataset retains the bulk of the variance and the ability to reconstruct the dataset up to some ceiling determined by random noise. When using this approach, it is important to understand which concept each PC captures. So-called factor loadings, the original features' weights on each PC, can provide this information.

PCA performs a *change of basis*: Instead of describing the dataset using the original features (in our case, 54 behaviors and model parameters), it creates new features, PCs, that are linear combinations of the original features and capture the same variance, but are orthogonal to each other. PCs are created by eigendecomposition of the covariance matrix of the dataset: the eigenvector with the largest eigenvalue shows the direction in the dataset in which most variance occurs, and represents the first PC. Eigenvectors with subsequently smaller eigenvalues form subsequent PCs. PCA is related to Factor analysis, and results are very consistent between both methods in our dataset. PCA and FA are often used for dimensionality reduction. In this case, only a small number of PCs / Factors is retained, whereas the majority is discarded, in an effort to retain most variance with a reduced number of features.

We highlight the most central behavioral features here; more detail is provided in 'Appendix 1' and 'Appendix 6'. Response to feedback was assessed using features 'Win-stay' (percentage of trials in which a rewarded choice was repeated), and 'Lose-stay' (percentage of trials in which a non-rewarded choice was repeated). For task B, we additionally included 'Win-lose-stay' tendencies, which is the proportion of trials in which participants stay after a winning trial that is followed by a losing trial. This is an important measure for this task because the optimal strategy required staying after single losses.

We also included behavioral persistence measures in all tasks. In tasks A and C, these included a measure of action repetition (percentage of trials in which the previous key was pressed again, irrespective of the stimulus and feedback) and choice repetition (percentage of trials in which the action was repeated that was previously selected for the same stimulus, irrespective of feedback). In task B, both measures were identical because every trial presents the same stimulus.

We further included task-specific measures of performance. In task A, these were: the average accuracy for the first three presentations of each stimulus, reflecting early learning speed; and the asymptote, intercept, and slope of the learning progress in a regression model predicting performance (for details about these measures, see *Xia et al., 2021*). In task B, task-specific measures of performance included the number of reversals (because reversals were performance-based); and the average number of trials to reach criterion after a switch. In tasks A and C, we also included a model-independent measure of forgetting. In task A, this was the effect of delay on performance in the regression model mentioned above. In task C, this was the effect of delay in a similar regression model, which also included set size, the number of previous correct choices, and the number of previous incorrect choices, whose effects were also included. Lastly for task C, we included the slope of accuracy and response times over set sizes, as measures of the effect of set size on performance. For task B, we also included the difference between early (first third of trials) and late (last third) performance as a measure of learning. To avoid biases in the PCA toward any specific task, we included equal numbers of behavioral features for each task. Before performing the PCA, we

individually standardized each feature, such that each feature was centered with a mean of 0 and a standard deviation of 1.

To facilitate the interpretation of PC2 and PC3, we normalized the loadings (PCA weights) of each feature (behavioral and model parameter) with respect to PC1, flipping the loadings of all features in PC2 and PC3 that loaded negatively on PC1. This step ensured that the directions of factor loadings on PC2 and PC3 were interpretable in the same way for all features, irrespective of their role for task performance, and revealed the encoding of task contrasts.

### Ridge regression

In sections Parameters and cognitive processes and parameters and behavior, we use regularized, cross-validated Ridge regression to determine whether parameters captured overlapping variance, which would point to an overlap in cognitive processes. We used Ridge regression to avoid problems that would be caused by overfitting when using regular regression models. Ridge regression regularizes regression weight parameters $w$ based on their L2-norm. Regular regression identifies a vector of regression weights $w$ that minimize the linear least squares $\|y - wX\|_2^2$. Here, $\|a\|_2^2 = \sqrt{\sum_{a_i \in x} a_i^2}$ is the L2-norm of a vector $a$, vector $y$ represents the outcome variable (in our case, a vector of parameters, one fitted to each participant), matrix $X$ represents the predictor variables (in our case, either several behavioral features for each participant [section 'Parameters and cognitive processes'], or several parameters fitted to each participant [section 'Parameters and behavior']), and vector $w$ represents the weights assigned to each feature in $X$ (in our case, the weight assigned to each predicting behavioral pattern or each predicting parameter).

When datasets are small compared to the number of predictors in a regression model, *exploding* regression weights $w$ can lead to overfitting. Ridge regression avoids this issue by not only minimizing the linear least squares like regular regression, but also the L2 norm of weights $w$, that is, by minimizing $\|y - wX\|_2^2 + \alpha * \|w\|_2^2$. Parameter $\alpha$ is a hyper-parameter of Ridge regression, which needs to be chosen by the experimenter. To avoid bias in the selection of $\alpha$, we employed repeated cross-validated grid search. At each iteration of this procedure, we split the dataset into a predetermined number $s \in [2, 3, \ldots, 8]$ of equal-sized folds, and then fitted a Ridge regression to each fold, using values of $\alpha \in [0, 10, 30, 50, 100, 300, \ldots, 10,000, 100,000, 1,000,000]$. For each $s$, we determined the best value of $\alpha$ based on cross-validation between folds, using the amount of explained variance, $R^2$, as the selection criterion. To avoid biases based on the random assignment of participants into folds, we repeated this procedure $n = 100$ times for each value of $\alpha$. To avoid biases due to the number of folds, the entire process was repeated for each $s$, and the final value of $s$ was selected based on $R^2$. We used the python package 'scikit learn' (*Pedregosa et al., 2011*) to implement the procedure.

We conducted three models per parameter to determine the relations between parameters: predicting each parameter from all the parameters of each of the other two tasks (2 models); and predicting each parameter from all parameters of both other tasks combined (1 model; *Figure 4A*). We conducted the same three models per parameter to determine the relations between parameters and behaviors, predicting each parameter from behavioral features of the other tasks (*Figure 4A*). In addition, we conducted a fourth model for behaviors, predicting each parameter from the behaviors of all three tasks combined, to assess the contributions of all behaviors to each parameter (*Figure 4C*). Meta-parameters $s$ and $\alpha$ were allowed to differ (and differed) between models. The final values of $R^2$ (*Figure 4B and D*) and the final regression weights $w$ (*Figure 4A and C*; *Table 6*) were determined by refitting the winning model.

### Data Availability

The data collected for this study are openly available at osf.io/h4qr6/. The analysis code is available at https://github.com/MariaEckstein/SLCN/blob/master/models/MetaSLCN-01ReadInData.ipynb. (copy archived at swh:1:rev:4fb5955c1142fcbd8ec80d7fccdf6b35dbfd1616, *Eckstein, 2022*).

## Acknowledgements

We thank Ian Ballard, Mayank Agrawal, Gautam Agarwal, and Bas van Opheusden for helpful comments on this manuscript, and Catherine Hartley and other members of her lab for fruitful discussion. We also thank Angela Radulescu and one anonymous reviewer for their helpful comments and suggestions. Numerous people contributed to this research: Amy Zou, Lance Kriegsfeld, Celia Ford,

Jennifer Pfeifer, Megan Johnson, Vy Pham, Rachel Arsenault, Josephine Christon, Shoshana Edelman, Lucy Eletel, Neta Gotlieb, Haley Keglovits, Julie Liu, Justin Morillo, Nithya Rajakumar, Nick Spence, Tanya Smith, Benjamin Tang, Talia Welte, and Lucy Whitmore. We are also grateful to our participants and their families. The work was funded by National Science Foundation SL-CN grant 1640885 to RD, AGEC, and LW.

# Additional information

## Funding

| Funder | Grant reference number | Author |
|---|---|---|
| National Science Foundation | sl-cn 1640885 | Ronald E Dahl<br>Anne GE Collins<br>Linda Wilbrecht |

The funders had no role in study design, data collection and interpretation, or the decision to submit the work for publication.

## Author contributions

Maria Katharina Eckstein, Conceptualization, Data curation, Software, Formal analysis, Investigation, Visualization, Methodology, Writing – original draft, Project administration, Writing – review and editing; Sarah L Master, Project administration, Writing – review and editing; Liyu Xia, Resources, Writing – review and editing; Ronald E Dahl, Funding acquisition, Writing – review and editing; Linda Wilbrecht, Resources, Supervision, Funding acquisition, Writing – original draft, Writing – review and editing; Anne GE Collins, Conceptualization, Resources, Supervision, Funding acquisition, Investigation, Methodology, Writing – original draft, Writing – review and editing

## Author ORCIDs

Maria Katharina Eckstein (i) http://orcid.org/0000-0002-0330-9367
Sarah L Master (i) http://orcid.org/0000-0003-2726-4586
Ronald E Dahl (i) http://orcid.org/0000-0002-1794-7132
Linda Wilbrecht (i) http://orcid.org/0000-0003-3492-8141
Anne GE Collins (i) http://orcid.org/0000-0003-3751-3662

## Ethics

Human subjects: We obtained informed consent from all participants at least 18 years of age, and informed assent and parents' informed consent from younger participants. We obtained ethical approval for this research from the Institutional Review Board of UC Berkeley.

## Decision letter and Author response

Decision letter https://doi.org/10.7554/eLife.75474.sa1
Author response https://doi.org/10.7554/eLife.75474.sa2

# Additional files

## Supplementary files

• MDAR checklist

## Data availability

The data collected for this project have been made available online on the OSF data servers.

The following datasets were generated:

| Author(s) | Year | Dataset title | Dataset URL | Database and Identifier |
|---|---|---|---|---|
| Eckstein M | 2021 | The Unique Advantage of Adolescents in Probabilistic Reversal | https://osf.io/jm2c8/ | Open Science Framework, jm2c8 |
| Xia L, Collins A | 2021 | Modeling Changes in Probabilistic Reinforcement Learning during Adolescence | https://osf.io/wq4te/ | Open Science Framework, wq4te |
| Eckstein M, Master SL, Zou AR, Collins A | 2021 | Data for "The interpretation of computational model parameters depends on the context" | https://osf.io/h4qr6/ | Open Science Framework, h4qr6 |

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

# Appendix 1

## Behavioral measures

In addition to RL model parameters, we also extracted behavioral features from each of the three tasks, conducted regression analyses, and in one case fitted a non-RL computational model. The resulting features of these analyses were used in the PCA (sections Main axes of variation and principal component analysis (PCA); *Figure 3*; *Appendix 8—figure 4*), behavioral Ridge regression (sections Parameters and behavior and ridge regression), and comprehensive correlation matrix (*Appendix 8—figure 3*). The following list details these features, using the labels in *Figure 3*, *Appendix 8—figure 4*, and *Appendix 8—figure 3* to denote each feature:

- Basic performance measures
  - 'Missed trials' (percentage of missed trials)
  - 'Response times' (average response times on correct trials)
  - 'Response time variability' (standard deviation of response times on correct trials)
  - 'Percent correct' (accuracy; overall percentage of correct trials)
- Advanced performance measures
  - 'Acc. first 3 trials' (only for task A; average accuracy on the first three trials of each stimulus; a measure of initial learning)
  - 'Asymptotic acc.' (only for task A; an exponential curve was fit to the learning curve of each subject. The functional form was: $0.5 + a - a * exp(-c * (t - 1))$, where $t$ is trial number. $a$ is bounded $0 < 0.5$, and $c > 0$. This feature refers to the asymptote of the curve: $0.5 + a - a * exp(-c * (T - 1))$, where $T$ is total number of trials completed by the subject)
  - 'Learning slope' (only for task A; this feature refers to the slope $log(c)$ of the model above)
  - 'Acc. intercept' (only for task A; hierarchical regression was fit to predict the probability of a correct choice on each trial. The regression formula was $p(correct) 1 + prew + delay + (1 + prew + delay|id)$, where $(...|id)$ indicates random effects by subject id. This feature refers to the random intercept for each subject)
  - 'Reward effect on acc.' (in the same regression model, this feature refers to the random slope for the reward history predictor, which is the z-scored number of correct trials for each butterfly)
  - 'Delay' (only tasks A and C; in the same [or an equivalent] regression model, this feature refers to the random slope for the delay predictor, which is the z-scored number of trials since last time the same stimulus was seen and the participant chose the correct action)
  - 'Number of switches' (only for task B; the number of task switches experienced by each participant; because the occurrence of switches was based on performance, the number of switches is an additional criterion of task performance)
  - 'Criterion after switch' (only for task B; number of trials after a task switch until participants reached a performance criterion of 2 correct choices after the task switch)
  - 'Acc. late minus early' (only for task B; difference in accuracy between the first and last third of trials in the task, as a measure of 'slow' learning)
- Action repetition
  - 'Stay (choice)' (for tasks A and C; percentage of choices that were repeated with respect to each stimulus, averaged across stimuli)
  - 'Stay (motor)' (for tasks A and C; percentage of choices that were repeated between two subsequent trials, irrespective of the shown stimulus)
  - 'Stay (both)' (for task B; because the task only provided a single stimulus, stay-choice and stay-motor were identical)
  - 'Win-stay' (Fraction of trials in which participants repeated a choice for a given stimulus that was rewarded on the previous trial for that stimulus, normalized by number of win trials, and averaged over stimuli: $\frac{win\ stay}{win\ stay+win\ shift}$)
  - 'Win-lose-stay' (for task B only; fraction of trials in which participants repeated a choice that was rewarded two trials back, but not rewarded on the previous trial, normalized by number of win-lose trials)
  - 'Lose-stay' (Fraction of trials in which participants repeated a choice for a given stimulus that was not rewarded on the previous trial for that stimulus, normalized by number of lose trials, and averaged over stimuli: $\frac{lose\ stay}{lose\ stay+lose\ shift}$)
- Bayesian model parameters (task B only)
  - '$p(switch)$ param.' (a Bayesian inference model was fit in addition to the RL model. The Bayesian model employed a mental model of the task, which was based on two hidden

states 'Left is correct' and 'Right is correct', and used Bayesian inference to infer the current hidden state based on recent outcomes. The free parameters of this model were the task parameters of the mental model, switch probability on each trial $p_{switch}$, and probability of reward for a correct choice $p_{reward}$, choice parameters Persistence $P$ and inverse decision temperature $\beta$. Detailed information about this model is provided in *Eckstein et al., 2022*. This feature refers to model parameter $p_{switch}$.)

- ○ '$p(reward)$ param.' (parameter $p(switch)$ in the Bayesian model)
- Behavioral measures of working memory (task C only)
  - ○ 'Set size effect on acc.' (a logistic regression was fitted to choice data, assessing the effects of set size, delay, number of previous correct trials, and number of previous incorrect trials on choice accuracy; this feature refers to the effect of set size)
  - ○ 'Feature $p(cor)$' (this features refers to the effect of the number of previous correct trials)
  - ○ 'Feature $p(inc)$' (this features refers to the effect of the number of previous incorrect trials)
  - ○ 'Set size Acc. slope' (performance was averaged for blocks of each set size, and the slope in performance over set sizes was determined)
  - ○ 'Set size Acc. slope' (similar to the previous feature, but replacing performance with response times)

# Appendix 2

## Task descriptions in the Markov decision process (MDP) framework

In order to enhance theoretical consistency between RL studies in the psychological literature, one step is the use of a common framework to describe and design laboratory tasks. One such framework is the MDP. Table *Appendix 2—table 1* shows our three tasks in terms of this frameworks, and complements *Figure 1E*.

**Appendix 2—table 1.** Task descriptions in the Markov decision process framework.
POMDP: Partially-observable Markov Decision Process. 'Pos. stochastic': positive outcomes are delivered stochastically. 'Neg. deterministic': negative outcomes are delivered deterministically.

| | Number of States | Number of Actions | Reward function |
|---|---|---|---|
| Task A | 4 (1 per trial) | 2 (visible on screen) | Stable, stochastic |
| Task B | Stateless / POMDP | 2 (visible on screen) | Volatile, pos. stochastic, neg. deterministic |
| Task C | 2/3/4/5 (1 per trial) | 3 (not visible) | Stable, deterministic |

## Appendix 3

### Previous results

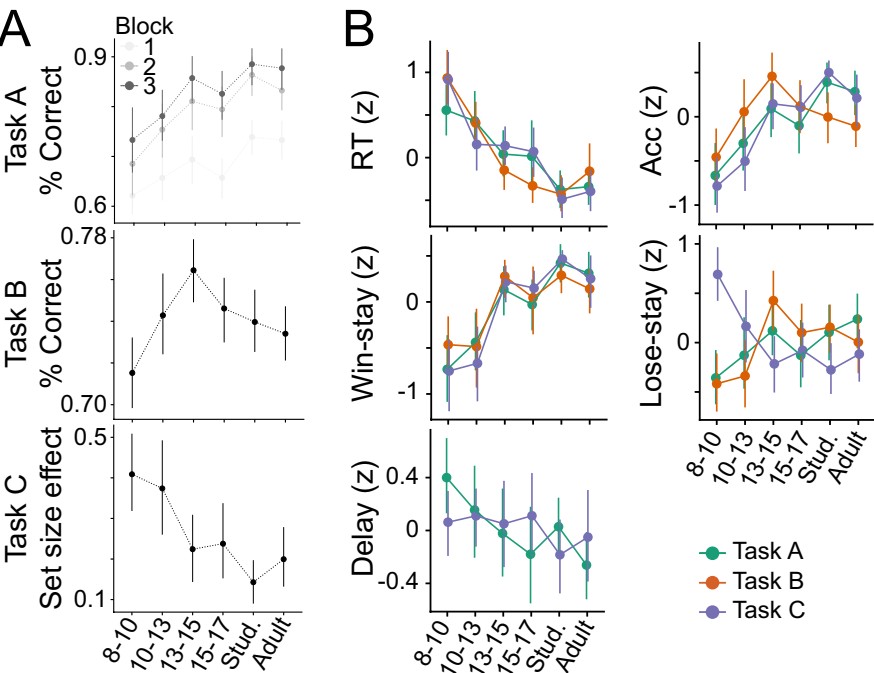

**Appendix 3—figure 1.** Main results of tasks A, B, and C. (**A**) Top: In task A, performance increased with age and plateaued in early adulthood, as captured in decreases in decision temperature $\frac{1}{\beta}$ and increases in learning rate $\alpha$ (**Xia et al., 2021**). Performance also increased over task time (blocks). Middle: In task B, performance showed a remarkable inverse U-shaped age trajectory: Performance increased markedly from early childhood (8–10 years) to mid-adolescence (13-15), but decreased in late adolescence (15-17) and adulthood (18-30) (**Rosenbaum et al., 2020**). Bottom: Task C showed that the effect of set size on performance (regression coefficient) decreased with age, which was captured by increases in RL learning rate, but stable WM limitations (**Master et al., 2020**). (**B**) Main behavioral features over age; colors denote task; all features are z-scored. Some measures (e.g. response times [RT], win-stay choices) were consistent across tasks, while others (e.g. accuracy [Acc.], lose-stay choices) showed significant differences (see Table **Appendix 6—table 1**).

Each task was first analyzed independently, and detailed results have been presented elsewhere (**Master et al., 2020**; **Eckstein et al., 2022**; **Xia et al., 2021**). We summarize the key results here. In task A, participants saw one of four butterflies on each trial, and aimed to pick the one of two flowers that was preferred by this butterfly. Each butterfly had a stable preference for one flower, and participants received a stochastic reward (80% probability) when they chose this flower. Nevertheless, sometimes the butterfly liked the opposite flower, and participants got a reward with 20% when they chose the opposite flower (**Figure 1B**; section Testing procedure). Task A has been used previously to investigate the role of reward sensitivity and its interplay with episodic memory, shedding light on the neural substrate of these processes, notably the striatum and hippocampus, and revealing a unique role of adolescence in stochastic learning (**Davidow et al., 2016**). In our sample, performance on task A increased with age through the early-twenties and then stabilized (**Xia et al., 2021**; **Appendix 3—figure 1A**). Using hierarchical Bayesian methods to fit RL models, we showed that this performance increase was driven by increasing positive learning rate $\alpha_+$ and decreasing decision noise $\frac{1}{\beta}$. Forgetting rates decreased very slightly with age, and negative learning rate $\alpha_-$ was 0, suggesting that participants ignored negative outcomes (**Figure 2A and C**).

In task B, participants saw two boxes and selected one on each trial, with the goal of collecting gold coins. For some period of time, one box was correct and led to a stochastic reward (75% probability), while the other was unrewarded (0% probability). Then, the contingencies switched unpredictably and unsignaled, and the opposite box became the correct one. A 120-trials session contained 2–7 switches (**Figure 1C**; section Testing procedure). Task B was adapted from the rodent literature,

where it has been used to show a causal link between stimulation of striatal spiny projection neurons and subsequent choices (*Tai et al., 2012*). Stochastic reversal tasks are also common in the human literature (*Peterson et al., 2009*; *Swainson et al., 2000*; *van der Schaaf et al., 2011*; *Waltz and Gold, 2007*; *Dickstein et al., 2010*; *Cools et al., 2002*; *Cools et al., 2009*; *Lourenco and Casey, 2013*; *Izquierdo et al., 2017*). A notable feature of our task compared to others is the deterministic feedback for incorrect choices. In our study, we found that human youth age 13–15 years markedly outperformed younger youth (8-12), older youth (16-17), and even young adults (18-30), suggesting that adolescent brains might be specifically adapted to perform well in the stochastic and volatile environment of task B. Computational modeling, using hierarchical Bayesian fitting, revealed that some model parameters (e.g. decision temperature $\frac{1}{\beta}$, Persistence) increased monotonically from childhood to adulthood, whereas others (e.g. learning rate for negative feedback $\alpha_-$, Bayesian inference parameters $p_{switch}$ and $p_{reward}$) showed pronounced U-shapes with peaks in 13-to-15-year-olds, similar to performance. Blending RL and a Bayesian inference models using PCA revealed that adolescents operated at a sweet spot that combined mature levels of task performance with child-like, short time scales of learning, and provided an explanation for adolescents' superior performance (*Eckstein et al., 2022*).

Task C was designed to dissociate the effects of RL and working memory, and has been used in diverse samples of adult participants (*Collins and Frank, 2012*; *Collins et al., 2017b*; *Heinz et al., 2017*; *Collins et al., 2017a*; *Collins and Frank, 2018*; *Collins, 2018*; *van den Bos et al., 2012*), but this study was the first to test it in a developmental samples (*Master et al., 2020*). In this task, participants saw one stimulus at a time (e.g. bee) and chose one of three actions in response (left, up, right; *Figure 1D*, right). Feedback was deterministic, that is, reliably identified each action as correct or incorrect. The goal of task C was to learn the correct response for each stimulus. The key feature of the task is that stimuli appear in independent blocks of different sizes, ranging from 2 to 5 stimuli (e.g. the bee could be presented in a block containing just 1 other animal, or up to 4 other animals). As set sizes increase, participants have been shown to shift the balance between using their capacity-limited, but reliable working memory system, to using their unlimited, but slower RL system (*Collins and Frank, 2012*). Task C estimates both memory systems, RL as well as working memory. We found that participants aged 8–12 learned slower than participants aged 13–17, and were more sensitive to set size (*Appendix 3—figure 1A*). Computational modeling revealed that developmental changes in RL were more protracted than changes in working memory: RL learning rate $\alpha_+$ increased until age 18, whereas WM parameters showed weaker and more subtle changes early in adolescence (*Master et al., 2020*).

# Appendix 4

## Computational model validation

This section provides more information on the set of computational models compared in each task and validates the claim that the models employed in the current study are the best-fitting models, and provide comparably good fit to each respective task.

*Appendix 4—figure 1* shows human and simulated model behavior side-by-side for each of the three tasks. Specifically, the figure shows the human behavior we observed in the current study, as well as the simulated behavior of the winning model of each task for artificial agents, each of which used the fitted parameters of a particular human participant. The fact that in each case, human behavior is approximated closely by model simulations validates the claim that the models we have chosen as the winning models explain human behavior adequately, and other models are not expected to perform qualitatively better.

To arrive at this conclusion, several competing models were compared in each study: In task A, the following six models were compared: Classic RL ($\alpha, \beta$); RL with asymmetric learning rates ($\alpha_+, \alpha_-, \beta$); Asymmetric RL with $\alpha_- = 0$ ($\alpha_+, 0, \beta$); RL with forgetting ($\alpha, \beta f$), Asymmetric RL with forgetting ($\alpha_+, \alpha_-, \beta, f$); and Asymmetric RL with $\alpha_- = 0$ and forgetting ($\alpha_+, 0, \beta, f$).

In task B, final comparison involved seven models with increasing complexity (the order of adding free parameters was determined in pre-analyses): Classic RL ($\alpha, \beta$); RL with counterfactual updating ($\alpha, \beta$, counterfactual $\alpha$); RL with counterfactual updating and perseverance ($\alpha, \beta$, counterfactual $\alpha$, perseverance); RL with perseverance, separate learning from positive versus negative outcomes, and counterfactual updating for positive outcomes ($\alpha_+, \beta$, counterfactual $\alpha_+$, perseverance, $\alpha_-$); RL with perseverance, separate learning from positive versus negative outcomes, and counterfactual updating for positive and negative outcomes ($\alpha_+, \beta$, counterfactual $\alpha_+$, perseverance, $\alpha_-$, counterfactual $\alpha_-$); winning, simplified 4-parameter RL model with perseverance and separate learning rates for positive versus negative outcomes, which are identical to the respective counterfactual updating rates ($\alpha_+$ = counterfactual $\alpha_+$, $\alpha_-$ = counterfactual $\alpha_-$, $\beta$, perseverance).

In task C, model comparison involved six competing models: Classic RL ($\alpha, \beta$), RL with undirected noise, RL with positive learning bias, RL with forgetting, RL with 4 learning rates, and the winning RL model with working memory ("RLWM").

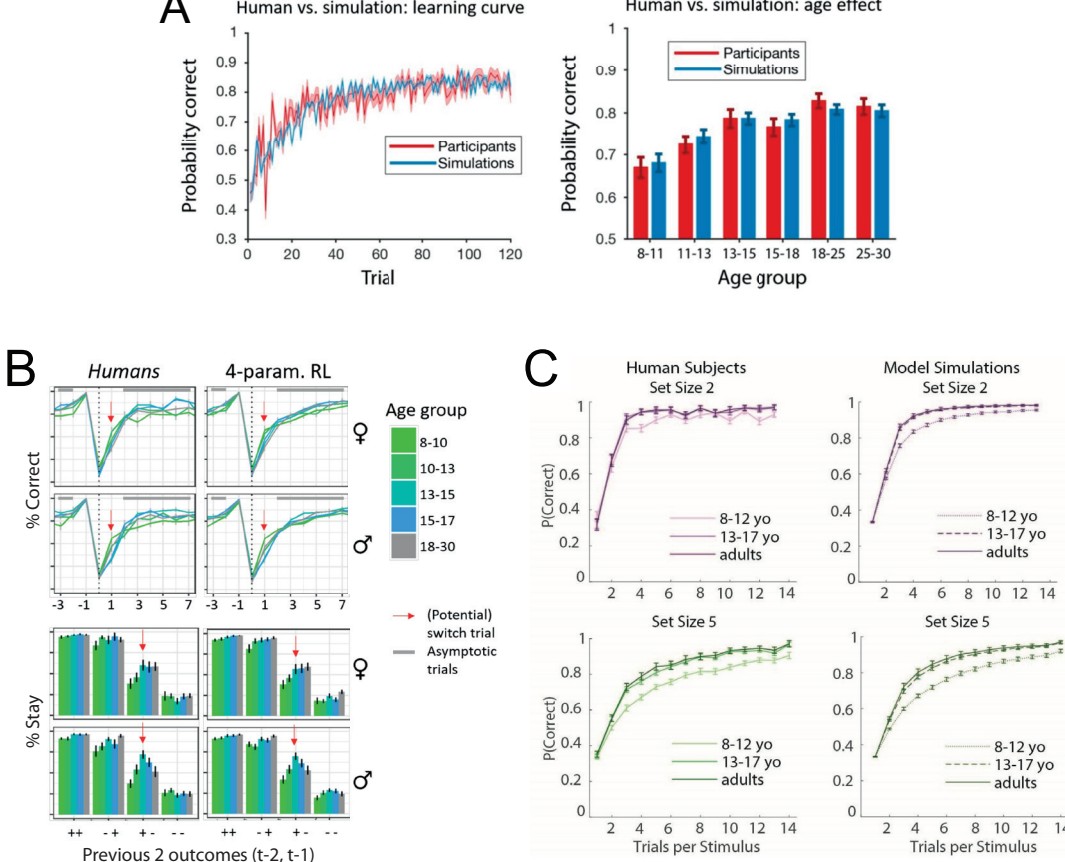

**Appendix 4—figure 1.** Behavioral validation of the winning model for each task. (**A**) Task A. The left figure shows performance (y-axis; probability of correct choice) over time on the task (x-axis; trial number). The right figure shows the average performance for each age group (in years). Red indicates human data, and blue indicates simulations from the winning model, based on best-fitting parameters. The close match between the red and blue datapoints indicates good model fit. (**A**) is reproduced from Figure 2 from *Xia et al., 2021*. (**B**) Task B. The top figure shows performance (y-axis; percentage of correct choices) aligned to switch trials (x-axis; i.e., trial on which the correct box switches sides), separately for male and female participants. The bottom figure shows another behavioral measures, the probability of repeating the same choice (y-axis; '% stay') based on the previous outcome history (x-axis; '+ +': two rewards in a row; '- +': no reward followed by reward; etc.), separately for male and female participants. Colors indicate participant age. The columnwise panels compare human behavior (left) to simulated behavior of the winning RL model (right). The close correspondence between human and simulated model behavior indicates good model fit. (**B**) is reproduced from Figure 4 from *Eckstein et al., 2022*. (**C**) Task C. Each figure shows human performance (y-axis; percentage of correct trials) over time (x-axis; number of trials for each stimulus), with colors differentiating age groups. The two rows show blocks of different set sizes (top: set size of two stimuli per block; bottom: set size of five). The left two figures show human behavior, the right two show model simulations. (**C**) is reproduced from Figure 3C from *Master et al., 2020*.

## Appendix 5

### Comparison of human parameter generalization to ceiling

Section Statistical comparison to generalizability ceiling describes the results of our analysis comparing humans to simulated agents with perfect generalization. This section provides additional methodological detail.

We each to create an agent population that was maximally similar to the human population, in order to obtain a ceiling of generalizability that was as realistic as possible for the current study. To this aim, we created simulated agents in the following way: We first obtained age trajectories for each parameter ($\alpha_+$, $\alpha_-$, $\epsilon$ / $\frac{1}{\beta}$, forget) by averaging human z-scored parameter values across tasks (e.g. for $\alpha_-$, averaging z-scored values of $\alpha_-$ across tasks B and C). We then obtained task-specific parameter values by 'un-z-scoring' these age trajectories: $param_{task} = param_z * stdev_{task} + mean_{task}$. This operation projects the shared parameter age trajectory of a parameter into the appropriate scale for each task. We chose this approach instead of averaging raw parameter values across tasks because the scale of each parameter differed so much between tasks (*Figure 2A*) that simulated behavior would hardly be interpretable outside this range.

For analyses, the same exact methods were used as for humans when fitting parameters (section Computational models), visualizing age trajectories (Fig. *Appendix 5—figure 1A* and *Appendix 5—figure 1B*), and performing statistical tests (ANOVA: *Appendix 5—table 1*; regression models: Tables *Appendix 5—table 2*, *Appendix 5—table 3*, and *Appendix 5—table 4*).

To statistically compare human results to the ceiling obtained from the simulated sample, we performed a bootstrapped correlation analysis: We first calculated the raw Spearman correlation scores between each pair of tasks for each parameter (e.g. $\alpha_+$ task A $\leftrightarrow$ task B; $\alpha_+$ task A $\leftrightarrow$ task C; $\alpha_+$ task B $\leftrightarrow$ task C), for both the human and simulated sample (dots in Fig. *Appendix 5—figure 1C*). We then calculated 95% confidence intervals for each correlation coefficient using bootstrapping, using the 'BCa' (reverse of the bias-corrected and accelerated bootstrap confidence interval) method of python's scipy package, with 1000 bootstrap samples per correlation coefficient. To determine whether there was a significant difference between humans and simulations, we determined whether the human correlation coefficient was within the 95% confidence interval of the simulated sample. This corresponds to a two-sided rejection criterion of $p = 0.05$.

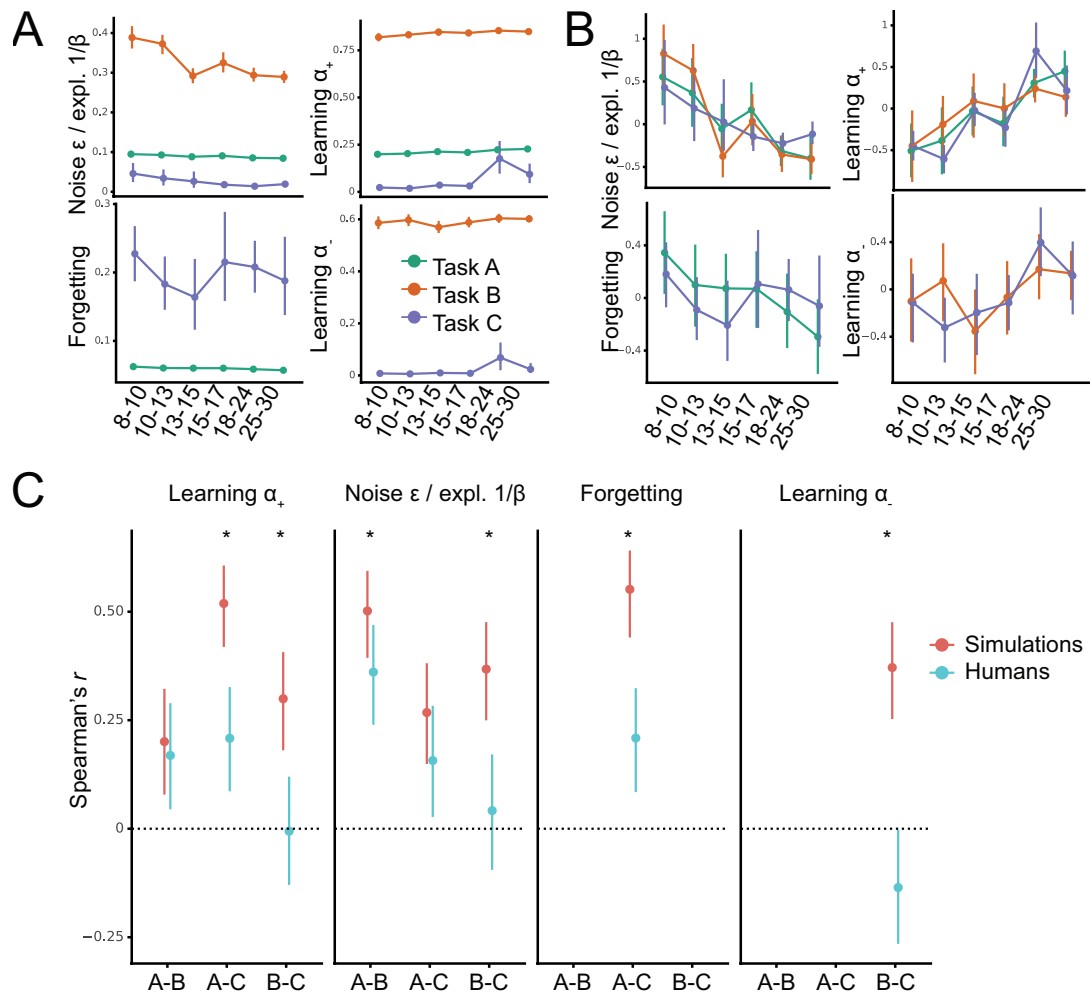

**Appendix 5—figure 1.** Comparison of human parameter correlations to generalization ceiling. (**A–B**) Same as *Figure 2A and B*, but for simulated agents with perfect generalization, rather than humans. (**C**) Parameter correlations (dots) for each pair of tasks (x-axis), with bootstrapped 95% confidence intervals (error bars). Stars indicate significance at the level of $p = 0.05$, that is, the human correlation coefficient is not contained within the confidence interval of the corresponding simulated correlation coefficient.

**Appendix 5—table 1.** Same as Table 1, but for simulated agents with perfect generalization, rather than humans.

| Parameter | Model | Tasks | F / t | df | p | sig. |
|---|---|---|---|---|---|---|
| $\frac{1}{\beta}$ | ANOVA | A, B | 2629 | 1 | $p < 0.001$ | *** |
| | t-test | A vs B | 49 | 246 | $p < 0.001$ | *** |
| $\alpha_+$ | ANOVA | A, B, C | 3753 | 2 | $p < 0.001$ | *** |
| | t-test | A vs B | 189 | 246 | $p < 0.001$ | *** |
| | t-test | A vs C | 13 | 246 | $p < 0.001$ | *** |
| | t-test | B vs C | 67 | 246 | $p < 0.001$ | *** |
| $\alpha_-$ | ANOVA | B, C | 6608 | 1 | $p < 0.001$ | *** |
| | t-test | B vs C | 81 | 246 | $p < 0.001$ | *** |

*Appendix 5—table 1 Continued on next page*

*Appendix 5—table 1 Continued*

| Parameter | Model | Tasks | F / t | df | $p$ | sig. |
|---|---|---|---|---|---|---|
| Forgetting | ANOVA | A, C | 185 | 1 | $p < 0.001$ | *** |
| | t-test | A vs C | 14 | 246 | $p < 0.001$ | *** |

**Appendix 5—table 2.** Same as Table 2, but for simulated agents with perfect generalization, rather than humans.

| Parameter | AIC without task | AIC with task | F(df) | p | sig. |
|---|---|---|---|---|---|
| $\frac{1}{\beta}/\epsilon$ | 1,938 | 1,937 | NA | NA | – |
| $\alpha_+$ | 1,992 | 1,996 | NA | NA | – |
| $\alpha_-$ | 1,350 | 1,355 | NA | NA | – |
| Forgetting | 1,345 | 1,344 | $F(2, 245) = 4.42$ | $p = 0.013$ | * |

**Appendix 5—table 3.** Same as Table 3, but for simulated agents with perfect generalization, rather than humans.

| Parameter | Tasks | Predictor | $\beta$ | $p$(Bonf.) | sig. |
|---|---|---|---|---|---|
| $\frac{1}{\beta}/\epsilon$ | A, B, C | Intercept | 2.55 | $< 0.001$ | *** |
| | | Age (linear) | –0.25 | $< 0.001$ | *** |
| | | Age (quadratic) | 0.005 | $< 0.001$ | *** |
| $\alpha_+$ | A, B, C | Intercept | –2.27 | $< 0.001$ | *** |
| | | Age (linear) | 0.22 | $< 0.001$ | *** |
| | | Age (quadratic) | –0.004 | $< 0.001$ | *** |
| $\alpha_-$ | A, B, C | Intercept | –1.03 | 0.055 | |
| | | Age (linear) | 0.06 | 0.12 | |
| | | Age (quadratic) | –0.002 | 0.25 | |
| Forgetting | A, C | Intercept | 0.57 | 0.29 | |
| | | Age (linear) | –0.045 | 0.47 | |
| | | Age (quadratic) | 0.001 | 0.70 | |

**Appendix 5—table 4.** Same as Table 4, but for simulated agents with perfect generalization, rather than humans.

| Parameter | Tasks | $\beta$ | p | sig. |
|---|---|---|---|---|
| $\frac{1}{\beta}, \epsilon$ | A & B | 3.04 | $< 0.001$ | *** |
| | A & C | 0.72 | $< 0.001$ | *** |
| | B & C | 0.44 | $< 0.001$ | *** |
| $\alpha_+$ | A & B | 0.34 | 0.002 | ** |
| | A & C | 0.01 | $< 0.001$ | *** |
| | B & C | 0.008 | 0.009 | ** |
| $\alpha_-$ | B & C | 0.018 | $< 0.001$ | *** |
| Forgetting | A & C | 0.023 | $< 0.001$ | *** |

# Appendix 6

## Age trajectories of behavioral measures

This section provides additional information about selected behavioral features (Appendix 1) and assesses their development with age. For statistical testing, we assessed age trajectories using similar regression models as before (section Parameter age trajectories).

Response times, reflecting choice fluidity and task engagement, sped up with age in all tasks, whereby age trajectories differed significantly between tasks A and B in pairwise follow-up models (grand model $AIC_{with\ task} = 1.868$, $AIC_{no\ task} = 1.871$; for detailed statistics, see Table *Appendix 6— table 1*; *Appendix 3—figure 1B*). Accuracy, reflecting subjective ease and task engagement, showed a significant increase with age, and no significant pairwise differences in age trajectories after correcting for multiple comparisons, despite the better fit of the model including task compared to the model without task ($AIC_{with\ task} = 2.015$, $AIC_{no\ task} = 2.024$; *Appendix 3—figure 1B*; Table *Appendix 6—table 1*).

**Appendix 6—table 1.** Statistics of mixed-effects regression models predicting z-scored behavioral features from task (task A, task B, task C), age, and squared age (months).

The task-less grand model is reported when it had the best model fit (win-stay, Delay). Otherwise, pairwise follow-up models are shown (RT, ACC, lose-stay), with p-values corrected for multiple comparison using the Bonferroni correction. * $p < .05$; ** $p < .01$, *** $p < .001$.

| Parameter | Tasks | Predictor | $\beta$ | $p$(Bonf.) | sig. |
|---|---|---|---|---|---|
| RT | task B & task A | Task (main effect) | 2.15 | 0.006 | ** |
| | | Linear age | −0.23 | 0.003 | ** |
| | | Task * linear age (interaction) | −0.25 | 0.003 | ** |
| | | Task * quadratic age (interaction) | 0.007 | 0.003 | ** |
| | task B & task C | Task (main effect) | −0.76 | 0.69 | |
| | | Linear age | −0.48 | < 0.001 | *** |
| | | Task * linear age (interaction) | 0.10 | 0.45 | |
| | | Task * quadratic age (interaction) | −0.003 | 0.288 | |
| | task A & task C | Task (main effect) | 1.40 | 0.63 | |
| | | Linear age | −0.23 | 0.003 | ** |
| | | Task * linear age (interaction) | −0.15 | 0.084 | |
| | | Task * quadratic age (interaction) | 0.004 | 0.129 | |
| ACC | task B & task A | Task (main effect) | 1.27 | 0.36 | |
| | | Linear age | 0.27 | < 0.001 | *** |
| | | Task * linear age (interaction) | −0.10 | 0.87 | |
| | | Task * quadratic age (interaction) | 0.001 | 1 | |
| | task B & task C | Task (main effect) | −2.15 | 0.033 | * |
| | | Linear age | 0.17 | 0.036 | * |
| | | Task * linear age (interaction) | 0.20 | 0.118 | |
| | | Task * quadratic age (interaction) | −0.004 | 0.33 | |
| | task A & task C | Task (main effect) | −0.88 | 0.60 | |
| | | Linear age | 0.27 | < 0.001 | *** |
| | | Task * linear age (interaction) | 0.10 | 0.57 | |
| | | Task * quadratic age (interaction) | −0.003 | 0.57 | |

*Appendix 6—table 1 Continued on next page*

*Appendix 6—table 1 Continued*

| Parameter | Tasks | Predictor | $\beta$ | $p$(Bonf.) | sig. |
|---|---|---|---|---|---|
| WS | —— | Intercept | −3.05 | < 0.001 | *** |
| | | Age (linear) | 0.31 | < 0.001 | *** |
| | | Age (quadratic) | −0.007 | < 0.001 | *** |
| LS | task B & task A | Task (main effect) | −0.90 | 0.42 | |
| | | Linear age | 0.075 | 0.87 | |
| | | Task * linear age (interaction) | 0.12 | 0.42 | |
| | | Task * quadratic age (interaction) | −0.004 | 0.29 | |
| | task B & task C | Task (main effect) | 4.84 | < 0.001 | *** |
| | | Linear age | 0.20 | 0.015 | * |
| | | Task * linear age (interaction) | −0.51 | < 0.001 | *** |
| | | Task * quadratic age (interaction) | 0.012 | < 0.001 | *** |
| | task A & task C | Task (main effect) | 3.94 | < 0.001 | *** |
| | | Linear age | 0.075 | 0.54 | |
| | | Task * linear age (interaction) | −0.39 | < 0.001 | *** |
| | | Task * quadratic age (interaction) | 0.008 | 0.003 | ** |
| Delay | —— | Intercept | 0.95 | 0.035 | * |
| | | Age (linear) | −0.09 | 0.07 | |
| | | Age (quadratic) | 0.002 | 0.14 | |

Win-stay (WS) behavior reflects participants' tendency to repeat rewarded actions, while lose-stay (LS) behavior reflects participants' tendency to repeat non-rewarded actions. Win-stay behavior increased with age, without task differences ($AIC_{\text{with task}} = 1.961$, $AIC_{\text{no task}} = 1.959$; ***Appendix 3—figure 1B***; Table ***Appendix 6—table 1***). Lose-stay behavior showed marked task differences ($AIC_{\text{with task}} = 2.075$, $AIC_{\text{no task}} = 2.109$), with inverse trajectories in task C compared to the other tasks: In task C, lose-stay behavior decreased monotonically until mid-adolescence (linear effect of age: $w = −0.31$, $p < 0.001$; quadratic effect: $w = 0.007$, $p < 0.001$), whereas in task A, it increased slightly (linear effect of age: $w = 0.075$, $p < 0.001$; quadratic effect: $w = −0.001$, $p < 0.001$). In task B, lose-stay behavior showed an inverse-U trajectory (linear effect: $w = 0.20$, $p < 0.001$; quadratic: $w = −0.005$, $p < 0.001$; ***Appendix 3—figure 1B***). These differences mirrored differences in optimal task strategies: Lose-stay is always a bad strategy in task C because negative feedback is diagnostic and actions with negative outcomes should never be repeated. In tasks A and B, on the other hand, some proportion of lose-stay choices is necessary because individual negative feedback is not diagnostic, and several pieces of evidence need to be integrated over time.

Lastly, the Delay pattern measured the decrease in accuracy with increasing delay between two presentations of the same stimulus (Appendix 1). Delay did not show significant age changes (Table ***Appendix 6—table 1***), and did not differ between tasks ($AIC_{\text{with task}} = 1.405$, $AIC_{\text{no task}} = 1.402$).

## Appendix 7

### Limitations of this research

Within-task parameter correlations

One limitation of our results is that regression analyses might be contaminated by parameter cross-correlations (sections Relative parameter differences, Parameter age trajectories, Predicting age trajectories), which would reflect modeling limitations (non-orthogonal parameters), and not necessarily shared cognitive processes. Concretely, parameters could be correlated between tasks for two reasons: (1) Because parameters are generalizable and consistent (which would be a great outcome). (2) However, a parameter like learning rate could also be correlated between two tasks because exploration parameters are correlated (e.g. due to generalization), simply because learning rates are negatively correlated with exploration parameters (which actually is the case in our study, as shown in *Appendix 8—figure 2*). So the 'actual' correlation of exploration parameters between tasks would lead to a 'spurious' correlation of learning rate parameters between tasks, just because exploration and learning rates are correlated within tasks. In other words, the significant correlation in *Appendix 8—figure 2* could indicate an 'actual' or a 'spurious' correlation, and we cannot know with certainty which one we are observing. However, this issue likely applies to most—if not all—computational modeling studies, and we hope that future research will provide more clarity on the issue. In other words, parameters $\alpha$ and $\beta$ are mathematically related in the regular RL modeling framework (*Sutton and Barto, 2017*; *Daw, 2011*), and we observed significant within-task correlations between these parameters for two of our three tasks (*Appendix 8—figure 2*, *Appendix 8—figure 3*). This indicates that caution is required when interpreting correlation results. However, correlations were also present between tasks (*Appendix 8—figure 1*, *Appendix 8—figure 3*), suggesting that within-model trade-offs were not the only explanation for shared variance, and that shared cognitive processes likely also played a role.

Another issue might arise if such parameter cross-correlations differ between models, due to the differences in model parameterizations across tasks. For example, memory-related parameters (e.g. $F$, $K$ in models A and C) might interact with learning- and choice-related parameters (e.g. $\alpha_+$, $\alpha_-$, noise/exploration), but such an interaction is missing in models that do not contain memory-related parameters (e.g. task B). If this indeed the case, that is, parameters trade off with each other in different ways across tasks, then a lack of correlation between tasks might not reflect a lack of generalization, but just the differences in model parameterizations. *Appendix 8—figure 2* indeed shows significant, medium-sized, positive and negative correlations between several pairs of Forgetting, memory-related, learning-related, and exploration parameters (though with relatively small effect sizes; Spearman correlation: $0.17 < |r| < 0.22$).

The existence of these correlations (and differences in correlations between tasks) suggest that memory parameters likely traded off with each other, as well as with other parameters, which potentially affected generalizability across tasks. However, some of the observed correlations might be due to shared causes, such as a common reliance on age, and the regression analyses in the main paper control for these additional sources of variance, and might provide a cleaner picture of how much variance is actually shared between parameters.

Furthermore, correlations between parameters within models are frequent in the existing literature, and do not prevent researchers from interpreting parameters—in this sense, the existence of similar correlations in our study allows us to address the question of generalizability and interpretability in similar circumstances as in the existing literature. And lastly, we confirmed that our method is able to detect generalizability using the simulation approach described in section Statistical Comparison to Generalizability Ceiling. In other words, even though within-task parameter cross-correlations likely induced some noise, the sensitivity with which we are still able to detect generalization was enough to show successful generalization in the simulated sample (and a significant reduction in humans).

Test-retest reliability

Furthermore, parameter generalizability is naturally bounded by parameter reliability, that is, the stability of parameter estimates when participants perform the same task twice (test-retest reliability) or when estimating parameters from different subsets of the same dataset (split-half reliability). The reliability of RL models has recently become the focus of several parallel investigations (*Weidinger et al., 2019*; *Brown et al., 2020*; *Pratt et al., 2021*; *Shahar et al., 2019*), some employing very similar tasks to ours (*Waltmann et al., 2022*). The investigations collectively suggest that excellent reliability can often be

achieved with the right methods, most notably by using hierarchical model fitting. Reliability might still differ between tasks or models, potentially being lower for learning rates than other RL parameters (*Waltmann et al., 2022*), and differing between tasks (e.g. compare *Weidinger et al., 2019* to *Brown et al., 2020*). In this study, we used hierarchical fitting for tasks A and B and assessed a range of qualitative and quantitative measures of model fit for each task (*Eckstein et al., 2022*; *Master et al., 2020*; *Xia et al., 2021*), boosting our confidence in high reliability of our parameter estimates, and the conclusion that the lack of between-task parameter correlations was not due to a lack of parameter reliability, but a lack of generalizability. This conclusion is further supported by the fact that larger between-task parameter correlations ($r > 0.5$) than those observed in humans were attainable—using the same methods—in a simulated dataset with perfect generalization.

## Model misspecification

Another concern relates to potential model misspecification and its effects on model parameter estimates: If components of the true data-generating process are not included in a model (i.e. a model is misspecified), estimates of existing model parameters may be biased. For example, if choices have an outcome-independent history dependence that is not modeled properly, learning rate parameters have shown to be biased (*Katahira, 2018*). Indeed, we found that learning rate parameters were inconsistent across the tasks in our study, and two of our models (A and C) did not model history dependence in choice, while the third (model B) only included the effect of one previous choice (persistence parameter), but no multi-trial dependencies. It is hence possible that the differences in learning rate parameters between tasks were caused by differences in the bias induced by misspecification of history dependence, rather than a lack of generalization. Though pressing, however, this issue is difficult to resolve in practicality, because it is impossible to include all combinations of possible parameters in all computational models, that is, to exhaustively search the space of possible models ('Every model is wrong, but to varying degrees'). Furthermore, even though our models were likely affected by some degree of misspecification, the research community is currently using models of this kind. Our study therefore sheds light on generalizability and interpretability in a realistic setting, which likely includes models with varying degrees of misspecification. Lastly, our models were fitted to high standards and achieved good behavioral recovery (Fig. *Appendix 4—figure 1*), which also reduces the likelihood of model misspecification.

## Difference in models between tasks

Another pressing issue is to what degree the claims of this study are dependent on the precise specification of the model for each task. For example, if all models included the same common set of parameters, would the same claims hold? This question could theoretically be addressed by using the same exact model (i.e. including the exact same equations and parameters) on all tasks. However, this approach is in practice unfeasible:

1) If we chose the 'smallest common denominator' model, that is, the simplest model that could produce behavior on all tasks (e.g. simple $\alpha$-$\beta$ RL), we would induce significant model misspecification as described above, and render fitted parameters—and claims about their generalizability and interpretability—uninterpretable.

2) However, choosing a 'mega' model including all current models as special cases is likewise impossible, for two reasons: First, even our relatively large dataset would not allow fitting such a big model due to the number of free parameters (i.e. the mega model would lose in model comparison to simpler models due to Occam's razor). And second, each individual task is too restrictive to fit such a model (e.g. task B does not tax memory for states, and would not allow fitting the range of memory parameters present in the other two models). Taken together, from a theoretical perspective, comparing parameters of the same model between different tasks would provide a good test of parameter generalizability. However, this is in practice infeasible given current methods and standards (e.g. simplistic, highly-controlled tasks; current modeling practices, including model fitting; data limitations). Advances in any of these areas might lead to an increase in the generalizability and interpretability of computational modeling parameters in the future.

Taking a step back, current practices ostensibly force us to choose between model misspecification on one hand and model generality on the other (*Navarro, 2019*): If we use the same, general model for different tasks, we induce model misspecification as described above, leading to biased and uninterpretable parameters. But if we use task-specific models that reproduce human behavior more closely, we induce differences in parameterization that likely create differences in interpretation and generalizability.

## Appendix 8

## Other supplemental figures

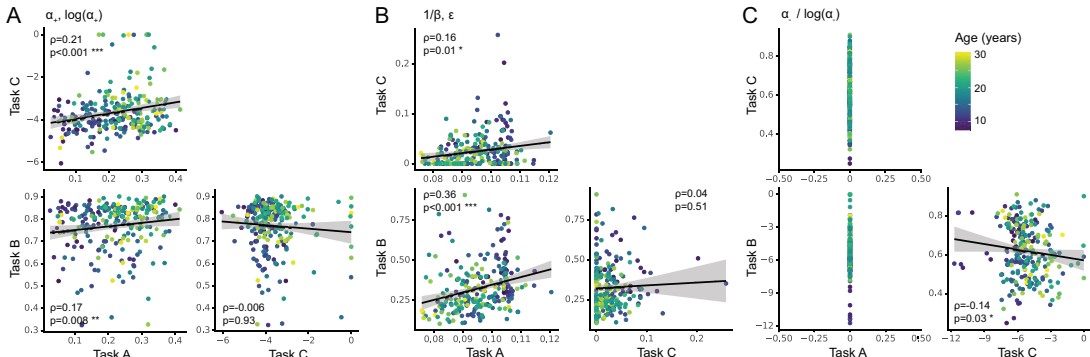

**Appendix 8—figure 1.** Between-task parameter correlations. (**A**) Parameter $\alpha_+$ across tasks ($log(\alpha_+)$ in task C). (**B**) Parameters $\frac{1}{\beta}$ (task A, B) and $\epsilon$ (task C). (**C**) Parameter $\alpha_-$ across tasks ($log(\alpha_-)$ in task C). Same conventions as in Fig. **Appendix 8—figure 2**.

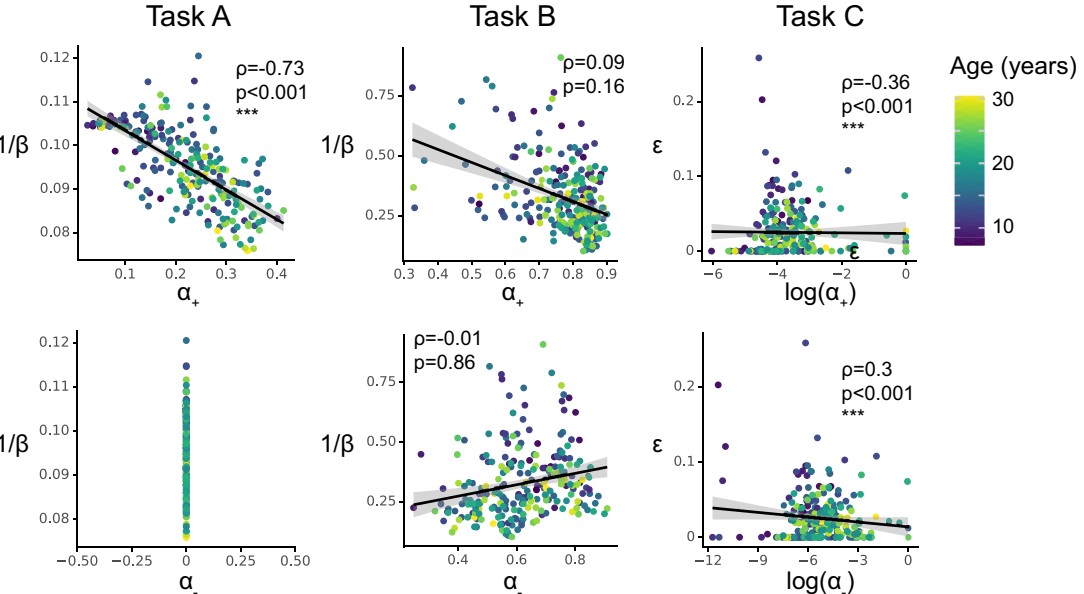

**Appendix 8—figure 2.** Within-task parameter correlations, focusing on learning rates (x-axes) and exploration / noise parameters (y-axes). Each column shows one task. Each dot in the scatter plots refers to a participant, colors indicates age. Inserted are Spearman correlation statistics.

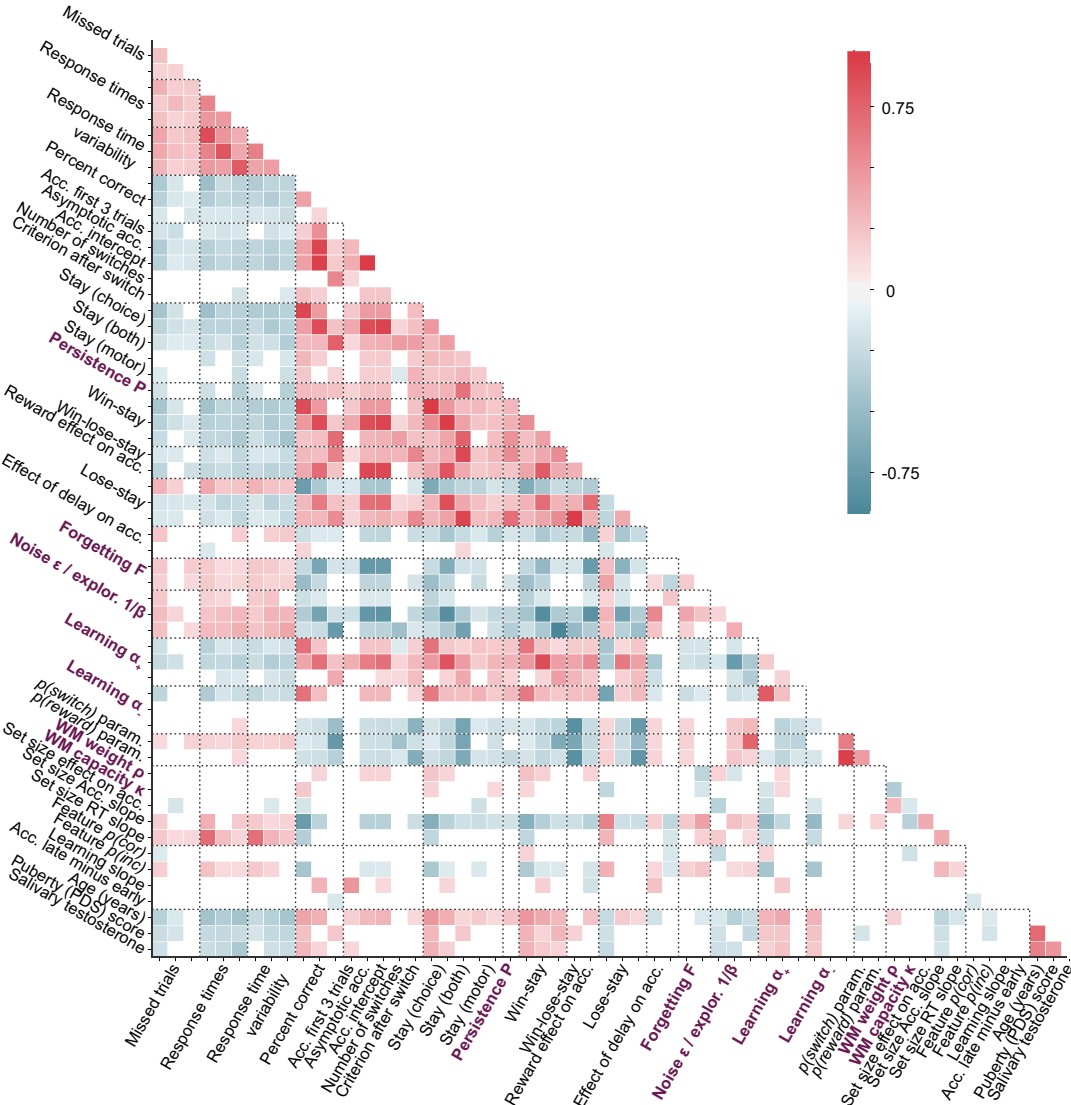

**Appendix 8—figure 3.** Full Spearman correlation matrix of all features in the dataset. Feature order is the same as in *Figure 3*. Deeper red (blue) colors indicate stronger positive (negative) correlations in terms of Spearman's $\rho$ (see color legend). Only correlations with p-values are shown; remaining squares are left blank.

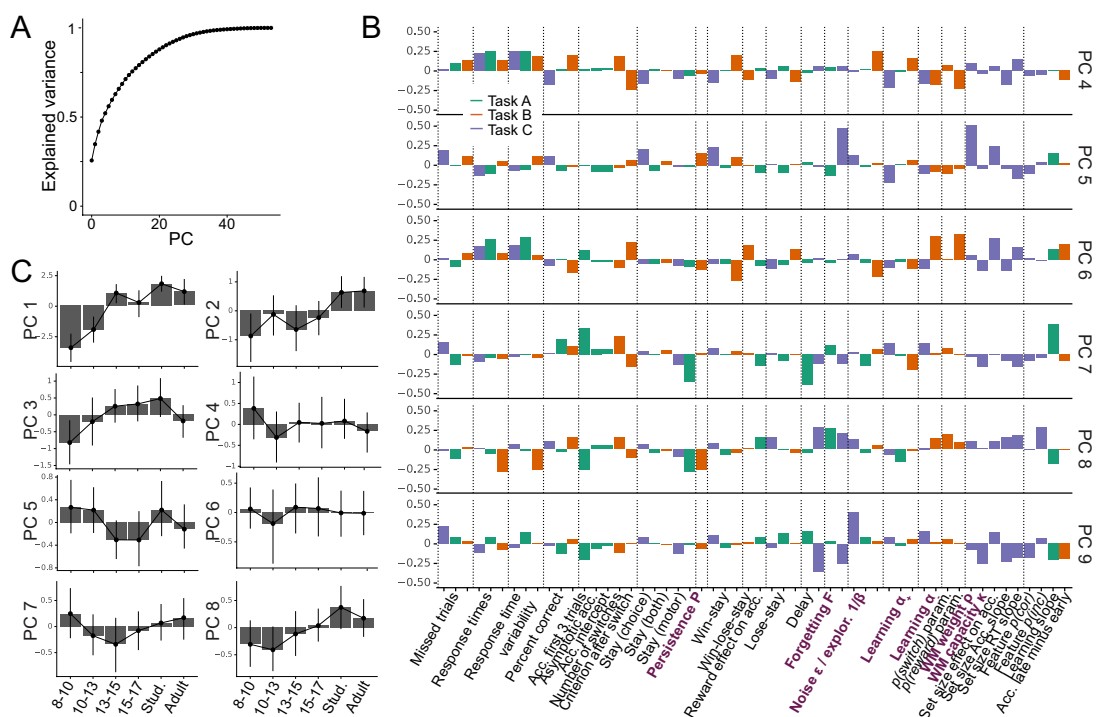

**Appendix 8—figure 4.** Additional PCA results. (**A**) Cumulative variance explained by all PCs of the PCA (*Figure 3*; 2.2.1). The smooth, non-stepped function does not provide evidence for lower-dimensional structure within the dataset. (**B**) Feature loadings (weights) of PC4-PC9. Loadings are flipped based on their relation to task performance, like for PC2-PC3 in *Figure 3*. (**C**) Age trajectories of the top 8 PCs, by age group. Corresponding statistics in *Appendix 8—table 1*.

**Appendix 8—table 1.** Statistics of regular regression models predicting each PC from two age predictors (linear and quadratic).

| PC | Effect | $\beta$ | t | p | sig. |
|---|---|---|---|---|---|
| 1 | age (linear) | 1.56 | 6.56 | | *** |
| | age (quadratic) | 0.035 | 5.61 | | *** |
| 2 | age (linear) | 0.34 | 2.17 | 0.031 | * |
| | age (quadratic) | 0.007 | 1.64 | 0.10 | — |
| 3 | age (linear) | 0.46 | 3.27 | 0.001 | ** |
| | age (quadratic) | –0.011 | –3.13 | 0.002 | ** |

**Appendix 8—table 2.** Statistics of mixed-effects regression models predicting parameter values from task (A, B, and C), age, and squared age (months).
Only effects including task are reported. * $p < .05$; ** $p < .01$, *** $p < .001$.

| Parameter | Tasks | Predictor | | p | sig. |
|---|---|---|---|---|---|
| $\alpha_+$ | task B & task A | Task (main effect) | 0.79 | < 0.001 | *** |
| | | Task * linear age (interaction) | –0.025 | 0.009 | ** |
| | | Task * quadratic age (interaction) | 0.001 | 0.021 | * |
| | task B & task C | Task (main effect) | 0.84 | < 0.001 | *** |
| | | Task * linear age (interaction) | –0.012 | 0.41 | |
| | | Task * quadratic age (interaction) | < 0.001 | 0.55 | |

*Appendix 8—table 2 Continued on next page*

*Appendix 8—table 2 Continued*

| Parameter | Tasks | Predictor | | p | sig. |
|---|---|---|---|---|---|
| | task A & task C | Task (main effect) | 0.048 | 0.70 | |
| | | Task * linear age (interaction) | −0.12 | 0.37 | |
| | | Task * quadratic age (interaction) | < 0.001 | 0.36 | |
| $\frac{1}{\beta}$ | task B & task A | Task (main effect) | 0.49 | < 0.001 | *** |
| | | Task * linear age (interaction) | −0.026 | 0.046 | * |
| | | Task * quadratic age (interaction) | 0.001 | < 0.001 | *** |
| $\alpha_-$ | task B & task C | Task (main effect) | 11.70 | < 0.001 | *** |
| | | Task * linear age (interaction) | 0.58 | < 0.001 | *** |
| | | Task * quadratic age (interaction) | −0.013 | < 0.001 | *** |
| Forgetting | task B & task C | Task (main effect) | 0.10 | 0.36 | |
| | | Task * linear age (interaction) | 0.005 | 0.70 | |
| | | Task * quadratic age (interaction) | < 0.001 | 0.67 | |

