## [Editor Report]

This study adopts a within-participant approach to address two important questions in the field of human reinforcement learning: to what extent do estimated computational model parameters generalize across different tasks and can their meaning be interpreted in the same way in different task contexts? The authors find that inferred parameters show moderate to little generalizability across tasks, and that their interpretation strongly depends on task context.

---

## [Decision Letter]

**Decision letter after peer review:**

Thank you for submitting your article "Learning Rates Are Not All the Same: The Interpretation of Computational Model Parameters Depends on the Context" for consideration by *eLife*. Your article has been reviewed by 2 peer reviewers, and the evaluation has been overseen by a Reviewing Editor and Timothy Behrens as the Senior Editor. The following individual involved in review of your submission has agreed to reveal their identity: Angela Radulescu (Reviewer #2).

Essential revisions:

1) The authors should carefully consider whether model misspecification may play a role in the observed results. More information should be provided on the set of models compared to validate the claim that the models presented are the best-fitting models and to establish whether each best-fitting model provides a comparably good fit to its respective task. For example, might models that incorporate various different forms of memory processes provide a better fit to the data for Tasks A and B? Moreover, to what degree are the claims in the current paper dependent on the precise specification of the model for each task? If a common set of parameters are included in each model specification, do the same claims hold?

2) While it may be beyond the scope of the current data, it would be helpful to include some discussion of test-retest reliability of the current tasks, and of RL tasks more generally, and how these measures relate to expectations about generalizability.

3) Was there a reason that the authors used PCA rather than factor analysis to extract common components across task parameters/behavioral indices?

4) The finding that both the positive learning rate and inverse temperature generalized across tasks is an important result that is not given the same weight in the exposition of the results as those parameters that exhibit generalization failures. The abstract and title could be edited to provide a more balanced reflection of these findings and to eliminate the somewhat "straw man"-ish implication that the field expects that the absolute values of learning rates should be the same across tasks.

5) To determine what might be a reasonably expected ceiling on the degree of generalizability, as suggested by the reviewer, the authors could simulate data from agents using the same generative parameters across the three tasks and report the correlations between the resulting parameters estimates.

6) Descriptions of the tasks that capture their distinguishing features through the lens of Markov Decision Process theory might be valuable to contextualize the current results within a broader reinforcement learning literature.

The reviewers also made a number of concrete suggestions for ways in which the manuscript might be improved that the authors should consider carefully.

*Reviewer #1 (Recommendations for the authors):*

I hope that the above three points will be discussed in the Discussion section or in the relevant part of the Method section. In the following, I will discuss some minor points.

p 4, Line 87

The citations for clinical research areas [15-18] seem to be mostly review articles in computational psychiatry, and I am not sure if the literature explicitly addressing the inconsistencies discussed here is properly cited. For example, a review of the lack of agreement between studies on whether the association with depression appears in learning rates or temperature parameters (Robinson and Chase, 2017, Computational Psychiatry, 1, 208-233) and the literature cited there may also be included.

p.14

The symbol '$' in Table 4 needs to be explained.

p.25

Figure 5: Is 'B' the correct label for 'Panel C'?

*Reviewer #2 (Recommendations for the authors):*

With regard to obtaining a ceiling on correlations that indicate relative generalizability, it might be possible to get an answer to this w/o collecting new data by simulating behavior of the *same N agents* (with uniformly sampled params, or params sampled from prior group distribution used in the hierarchical fit, or params sampled from distributions centered around ideal observer values). If we then re-run model fitting on these simulated datasets, what is the max correlation between LRs estimated from different tasks? And can we bootstrap to obtain some CIs around this ceiling? In general, the authors consider the tasks to be fairly similar, but I'm not sure that is actually the case -- even A and C, which share the most similarities, differ markedly in the reward function (probabilistic vs. deterministic).

With regard to clarifying the interaction between memory parameters and LR/exploration, have authors specifically determined the degree to which memory-related parameters F and/or K interact with α_+/β, e.g. by plotting likelihood function surfaces as a function of pairs of parameters? I also couldn't find the pairwise correlation between F (Task A) and F( Task C) in this section, were these reported? I'm guessing these were not significant based on the results in 2.1.4 showing F in A does not predict F in B, but it would be helpful to include them for completeness.

The current study was not designed for systematically varying different aspects of the MDP to see how this affects inferred parameter generalizability and interpretability, which is okay. But perhaps one step the paper can make in this direction of theoretical consistency is to explicitly name the features of each task from an MDP theory standpoint, or at least additionally include this as a table or additional panel in Figure 1. e.g. size of state and action space (+ mapping to "set size" jargon), reward function, etc. Authors do this at several points in the paper (including the appendix), and this can only help as a practice that allows the RL community to think about tasks in a common way. In the present study, this convention can illuminate why the tasks presented here are actually quite different from one another. Task A: 4 states (1 per trial), 2 actions (visible on screen), stable stochastic reward function; Task B: stateless, 2 actions, reversal in stochastic reward function; Task C: RL-WM task: 2-5 states (1 per trial), 3 actions (not visible on screen), stable deterministic reward function.

With regard to model selection:

– For the section describing the computational models for the first time, it would be helpful to summarize info about parameters in a table, emphasizing those parameters common across best-fitting models. For example, just from a rhetorical standpoint, it seemed strange that Tasks A and C are not both best fit by a model that allows LR- to vary freely, even though both tasks are taken as similar in the paper's taxonomy.

– It would be helpful to show upfront in a figure that each best-fitting model explains data in its respective task as well as models in the other tasks. i.e. can we say that each model explains about the same number of choices in every task? This is important because incomplete models are the reason we might see "contamination" whereby variance explained by one parameter in a task is explained by a different parameter in a second task.

– I did wonder here to what extent some of the inconsistencies can be explained by the fact that authors only instantiated a model that formalizes explicit memory demands only for Task C. We know from past work (e.g. by Bornstein and colleagues) that even in stateless bandit tasks such as Task B, explicit memory of outcomes stretching back tens of trials ago impact decisions. Task A is also potentially solvable by a WM module that keeps track of explicit state-action associations, bearing similarity to Task C. Is the forgetting parameter F enough to capture all the memory processes that might *also* be present in Task A? And would a RL-WM model with a fixed set size do better at explaining behavior on Task A? The lack of specificity in the noise/exploration parameter for Task A (evidenced by the fact that it was correlated with different parameters in other tasks, including working memory weight from C), strongly supports this possibility, as this parameter is known to absorb unexplained variance.

Finally, I think it is important to provide a bit more detail upfront about the training and instructions protocol for each task, and, to the extent that it is possible, provide evidence that instruction comprehension is not related to parameter differences. One simple analysis here would be to examine the "lose-stay" distribution in Tasks B and C and look for any evidence of bimodality that can be traced to how instructions were presented. I would also be curious if this distribution varies with age, potentially explaining the striking divergence in age trajectories for LR- in Tasks B and C.

---

## [Author Response]

Essential revisions:[Essential revisions]: “More information should be provided on the set of models compared to validate the claim that the models presented are the best-fitting models and to establish whether each best-fitting model provides a comparably good fit to its respective task.”

We thank the reviewer for this helpful comment, as we indeed did not provide much information about our models. We have addressed this issue in several ways, including a new supplementary section with information about the set of models compared, modifications to the introduction that provide more information about the winning models, a new panel that specifies the parameters of each winning model in Figure 1, clearer references to the original publications (which contain in-depth information about the specific models for each task), and a supplemental figure that compares the behavior of humans and simulations of each winning model. For convenience, we copy these modifications below.

New supplementary section with information about the set of models compared:

“In task A, the following six models were compared: Classic RL (α, β); RL with asymmetric learning rates (α+, α-, β); Asymmetric RL with α- = 0 (α+, 0, β); RL with forgetting (α, β, f), Asymmetric RL with forgetting (α+, α-, β, f); and Asymmetric RL with α- = 0 and forgetting (α+, 0, β, f).

In task B, final comparison involved seven models with increasing complexity (the order of adding free parameters was determined in pre-analyses): Classic RL (α, β); RL with counterfactual updating (α, β, counterfactual α); RL with counterfactual updating and perseverance (α, β, counterfactual α, perseverance); RL with perseverance, separate learning from positive versus negative outcomes, and counterfactual updating for positive outcomes (α+, β, counterfactual α+, perseverance, α-); RL with perseverance, separate learning from positive versus negative outcomes, and counterfactual updating for positive and negative outcomes (α+, β, counterfactual α+, perseverance, α-, counterfactual α-); winning, simplified 4-parameter RL model with perseverance and separate learning rates for positive versus negative outcomes, which are identical to the respective counterfactual updating rates (α+ = counterfactual α+, α = counterfactual α-, β, perseverance).

In task C, model comparison involved six competing models: Classic RL (α, β), RL with undirected noise, RL with positive learning bias, RL with forgetting, RL with 4 learning rates, and the winning RL model with working memory ("RLWM").”

In order to establish that each best-fitting model provides a good fit to its respective task data, we have Appendix 4-figure 1 that shows simulated behavior from each model, which in each case closely resembles human behavior.

[Reviewer #3]: “For the section describing the computational models for the first time, it would be helpful to summarize info about parameters in a table, emphasizing those parameters common across best-fitting models. For example, just from a rhetorical standpoint, it seemed strange that Tasks A and C are not both best fit by a model that allows LR- to vary freely, even though both tasks are taken as similar in the paper's taxonomy.”

We thank the reviewer for this comment. We have added the suggested table to Figure 1, shown in the comment above.

As for the models of tasks A and C, despite many similarities, both tasks also differ on several dimensions, including the nature of feedback (stochastic vs deterministic), the number of stimuli to learn about, and the number and visibility of available actions. These differences likely explain the differences in computational models, including the difference in LR-.

[Reviewer #3]: “It would be helpful to show upfront in a figure that each best-fitting model explains data in its respective task as well as models in the other tasks. i.e. can we say that each model explains about the same number of choices in every task? This is important because incomplete models are a reason we might see "contamination" whereby variance explained by one parameter in a task is explained by a different parameter in a second task.”

We agree with the sentiment that all models should fit their respective task equally well. However, there is no good quantitative measure of model fit that is comparable across tasks and models – for example, because of the difference in difficulty between the tasks, the number of choices explained would not be a valid measure to compare how well the models are doing across tasks. To address this issue, we have added the new supplemental section (Appendix C) mentioned above that includes information about the set of models compared, and explains why we have reason to believe that all models fit (equally) well. We also created the new supplemental Figure D.7, which directly compares human and simulated model behavior in each task, and shows a close correspondence for all tasks. Because the quality of all our models was a major concern for us in this research, we also refer the reviewer and other readers to the three original publications that describe all our modeling efforts in much more detail, and hopefully convince the reviewer that our model fitting was performed according to high standards.

[Essential revisions]: “For example, might models that incorporate various different forms of memory processes provide a better fit to the data for Tasks A and B?”[Reviewer #3]: “I did wonder here to what extent some of the inconsistencies can be explained by the fact that authors only instantiated a model that formalizes explicit memory demands for Task C. We know from past work (e.g. by Bornstein and colleagues) that even in stateless bandit tasks such as Task B, explicit memory of outcomes stretching back tens of trials ago impact decisions. Task A is also potentially solvable by a WM module that keeps track of explicit state-action associations, bearing similarity to Task C. Is the forgetting parameter F enough to capture all the memory processes that might *also* be present in Task A? And would a RL-WM model with a fixed set size do better at explaining behavior on Task A? The lack of specificity in the noise/exploration parameter for Task A (evidenced by the fact that it was correlated with different parameters in other tasks, including working memory weight from C), strongly supports this possibility, as this parameter is known to absorb unexplained variance.”

We appreciate this very thoughtful question, which raises several important issues. (1) As the reviewer said, the models for task A and task C are relatively different even though the underlying tasks are relatively similar (minus the differences the reviewer already mentioned, in terms of visibility of actions, number of actions, and feedback stochasticity). (2) We also agree that the model for task C did not include episodic memory processes even though episodic memory likely played a role in this task, and agree that neither the forgetting parameters in tasks A and C, nor the noise/exploration parameters in tasks A, B, and C are likely specific enough to capture all the memory / exploration processes participants exhibited in these tasks.

However, this problem is difficult to solve: We cannot fit an episodic-memory model to task B because the task lacks an episodic-memory manipulation (such as, e.g., in Bornstein et al., 2017), and we cannot fit a WM model to task A because it lacks the critical set-size manipulation enabling identification of the WM component (modifying set size allows the model to identify individual participants’ WM capacities, so the issue cannot be avoided in tasks with only one set size). Similarly, we cannot model more specific forgetting or exploration processes in our tasks because they were not designed to dissociate these processes. If we tried fitting more complex models that include these processes to these tasks, they would most likely lose in model comparison because the increased complexity would not lead to additional explained behavioral variance, given that the tasks do not elicit the relevant behavioral patterns. Because the models therefore do not specify all the cognitive processes that participants likely employ, the situation described by the reviewer arises, namely that different parameters sometimes capture the same cognitive processes across tasks and models, while the same parameters sometimes capture different processes.

And while the reviewer focussed largely on memory-related processes, the issue of course extends much further: Besides WM, episodic memory, and more specific aspects of forgetting and exploration, our models also did not take into account a range of other processes that participants likely engaged in when performing the tasks, including attention (selectivity, lapses), reasoning / inference, mental models (creation and use), prediction / planning, hypothesis testing, etc., etc. In full agreement with the reviewer’s sentiment, we recently argued that this situation is ubiquitous to computational modeling, and should be considered very carefully by all modelers because it can have a large impact on model interpretation (Eckstein et al., 2021).

If we assume that many more cognitive processes are likely engaged in each task than are modeled, and consider that every computational model includes just a small number of free parameters, parameters then necessarily reflect a multitude of cognitive processes. The situation is additionally exacerbated by the fact that more complex models become increasingly difficult to fit from a methodological perspective, and that current laboratory tasks are designed in a highly controlled and consequently relatively simplistic way that does not lend itself to simultaneously test a variety of cognitive processes.

The best way to deal with this situation, we think, is to *recognize* that in different contexts (e.g., different tasks, different computational models, different subject populations), the same parameters can capture different behaviors, and different parameters can capture the same behaviors, for the reasons the reviewer lays out. Recognizing this helps to avoid misinterpreting modeling results, for example by focusing our interpretation of model parameters to our specific task and model, rather than aiming to generalize across multiple tasks. We think that recognizing this fact also helps us understand the factors that determine whether parameters will capture the same or different processes across contexts and whether they will generalize. This is why we estimated here whether different parameters generalize to different degrees, which other factors affect generalizability, etc. Knowing the practical consequences of using the kinds of models we currently use will therefore hopefully provide a first step in resolving the issues the reviewer laid out.

[Reviewer #3]: “With regard to clarifying the interaction between memory parameters and LR/exploration, have authors specifically determined the degree to which memory-related parameters F and/or K interact with α_+/β, e.g. by plotting likelihood function surfaces as a function of pairs of parameters? I also couldn't find the pairwise correlation between F(Task A) and F(Task C) in this section, were these reported? I'm guessing these were not significant based on the results in 2.1.4 showing F in A does not predict F in B, but it would be helpful to include them for completeness.”

We thank the reviewer for this comment, which raises an important issue. We are adding the specific pairwise correlations and scatter plots for the pairs of parameters the reviewer asked about in Author response image 1 (“bf_α” = LR task A; “bf_forget” = F task A; “rl_forget” = F task C; “rl_log_α” = LR task C; “rl_K” = WM capacity task C):

Within tasks:

**Author response image 1. sa2fig1:** 

Between tasks:

To answer the question in more detail, we have expanded our section about limitations stemming from parameter tradeoffs in the following way:“One limitation of our results is that regression analyses might be contaminated by parameter cross-correlations (sections 2.1.2, 2.1.3, 2.1.4), which would reflect modeling limitations (non-orthogonal parameters), and not necessarily shared cognitive processes. For example, parameters α and β are mathematically related in the regular RL modeling framework, and we observed significant within-task correlations between these parameters for two of our three tasks (suppl. Figure H.10, H.11). This indicates that caution is required when interpreting correlation results. However, correlations were also present between tasks (suppl. Figure H.9, H.11), suggesting that within-model trade-offs were not the only explanation for shared variance, and that shared cognitive processes likely also played a role.

Another issue might arise if such parameter cross-correlations differ between models, due to the differences in model parameterizations across tasks. For example, memory-related parameters (e.g., F, K in models A and C) might interact with learning- and choice-related parameters (e.g., α+, α-, noise/exploration), but such an interaction is missing in models that do not contain memory-related parameters (e.g., task B). If this indeed the case, i.e., parameters trade off with each other in different ways across tasks, then a lack of correlation between tasks might not reflect a lack of generalization, but just the differences in model parameterizations. Appendix 5-figure 1 indeed shows significant, medium-sized, positive and negative correlations between several pairs of Forgetting, memory-related, learning-related, and exploration parameters (though with relatively small effect sizes; Spearman correlation: 0.17 < |r| < 0.22).

The existence of these correlations (and differences in correlations between tasks) suggest that memory parameters likely traded off with each other, as well as with other parameters, which potentially affected generalizability across tasks. However, some of the observed correlations might be due to shared causes, such as a common reliance on age, and the regression analyses in the main paper control for these additional sources of variance, and might provide a cleaner picture of how much variance is actually shared between parameters.

Furthermore, correlations between parameters within models are frequent in the existing literature, and do not prevent researchers from interpreting parameters---in this sense, the existence of similar correlations in our study allows us to address the question of generalizability and interpretability in similar circumstances as in the existing literature.”

[Essential revisions]: “Moreover, to what degree are the claims in the current paper dependent on the precise specification of the model for each task? If a common set of parameters are included in each model specification, do the same claims hold?”

We thank the reviewers for this excellent question, and added the following paragraph to our Limitations sections to discuss the issue:

“Another pressing issue is to what degree the claims of this study are dependent on the precise specification of the model for each task. For example, if all models included the same common set of parameters, would the same claims hold? This question could theoretically be addressed by using the same exact model (i.e., including the exact same equations and parameters) on all tasks. However, this approach is in practice unfeasible:

1) If we chose the "smallest common denominator" model, i.e., the simplest model that could produce behavior on all tasks (e.g., simple α-β RL), we would induce significant model misspecification as described above, and render fitted parameters---and claims about their generalizability and interpretability---uninterpretable.

2) However, choosing a "mega" model including all current models as special cases is likewise impossible, for two reasons: First, even our relatively large dataset would not allow fitting such a big model due to the number of free parameters (i.e., the mega model would lose in model comparison to simpler models, since model comparison penalizes models with high high numbers of free parameters). And second, each individual task is too restrictive to fit such a model (e.g., task B does not tax memory for states, and would not allow fitting the range of memory parameters present in the other two models).

Taken together, from a theoretical perspective, comparing parameters of the same model between different tasks would provide a good test of parameter generalizability. However, this is in practice infeasible given current methods and standards (e.g., simplistic, highly-controlled tasks; current modeling practices, including model fitting; data limitations). Advances in any of these areas might lead to an increase in the generalizability and interpretability of computational modeling parameters in the future.

Taking a step back, current practices ostensibly force us to choose between model misspecification on one hand and model generality on the other […]: If we use the same, general model for different tasks, we induce model misspecification as described above, leading to biased and uninterpretable parameters. But if we use task-specific models that reproduce human behavior more closely, we induce differences in parameterization that likely create differences in interpretation and generalizability.“

[Essential revisions]: “While it may be beyond the scope of the current data, it would be helpful to include some discussion of test-retest reliability of the current tasks, and of RL tasks more generally, and how these measures relate to expectations about generalizability.”

We thank the reviewer for this useful comment, and have added the following paragraph to the Discussion section to address it:

“Furthermore, parameter generalizability is naturally bounded by parameter reliability, i.e., the stability of parameter estimates when participants perform the same task twice (test-retest reliability) or when estimating parameters from different subsets of the same dataset (split-half reliability). The reliability of RL models has recently become the focus of several parallel investigations […], some employing very similar tasks to ours […]. The investigations collectively suggest that excellent reliability can often be achieved with the right methods, most notably by using hierarchical model fitting. Reliability might still differ between tasks or models, potentially being lower for learning rates than other RL parameters […], and differing between tasks (e.g., compare […] to […]). In this study, we used hierarchical fitting for tasks A and B and assessed a range of qualitative and quantitative measures of model fit for each task […], boosting our confidence in high reliability of our parameter estimates, and the conclusion that the lack of between-task parameter correlations was not due to a lack of parameter reliability, but a lack of generalizability. This conclusion is further supported by the fact that larger between-task parameter correlations (r>0.5) than those observed in humans were attainable---using the same methods---in a simulated dataset with perfect generalization.”

[Essential revisions]: “Was there a reason that the authors used PCA rather than factor analysis to extract common components across task parameters/behavioral indices?”

To answer the reviewer's first question: We indeed standardized all features before performing the PCA. Apologies for missing to include this information – we have now added a corresponding sentence to the methods sections.

We also thank the reviewer for the mentioned reference, which is very relevant to our findings and can help explain the roles of different PCs. Like in our study, Moutoussis et al. found a first PC that captured variability in task performance, and subsequent PCs that captured task contrasts. We added the following paragraph to our manuscript:

“PC1 therefore captured a range of "good", task-engaged behaviors, likely related to the construct of "decision acuity" […]. Like our PC1, decision acuity was the first component of a factor analysis (variant of PCA) conducted on 32 decision-making measures on 830 young people, and separated good and bad performance indices. Decision acuity reflects generic decision-making ability, and predicted mental health factors, was reflected in resting-state functional connectivity, but was distinct from IQ […].”

To answer the reviewer's question about PCA versus FA, both approaches are relatively similar conceptually, and oftentimes share the majority of the analysis pipeline in practice. The main difference is that PCA breaks up the existing variance in a dataset in a new way (based on PCs rather than the original data features), whereas FA aims to identify an underlying model of latent factors that explain the observable features. This means that PCs are linear combinations of the original data features, whereas Factors are latent factors that give rise to the observable features of the dataset with some noise, i.e., including an additional error term.

However, in practice, both methods share the majority of computation in the way they are implemented in most standard statistical packages: FA is usually performed by conducting a PCA and then rotating the resulting solution, most commonly using the Varimax rotation, which maximizes the variance between features loadings on each factor in order to make the result more interpretable, and thereby foregoing the optimal solution that has been achieved by the PCA (which lack the error term). Maximum variance in feature loadings means that as many features as possible will have loadings close to 0 and 1 on each factor, reducing the number of features that need to be taken into account when interpreting this factor. Most relevant in our situation is that PCA is usually a special case of FA, with the only difference that the solution is not rotated for maximum interpretability. (Note that this rotation can be minor if feature loadings already show large variance in the PCA solution.)

To determine how much our results would change in practice if we used FA instead of PCA, we repeated the analysis using FA. See Figure 3 and Author response image 1, the results are quite similar:

FA

**Author response image 3. sa2fig3:** 

We therefore conclude that our specific results are robust to the choice of method used, and that there is reason to believe that our PC1 is related to Moutoussis et al.’s F1 despite the differences in method.

[Essential revisions]: “The finding that both the positive learning rate and inverse temperature generalized across tasks is an important result that is not given the same weight in the exposition of the results as those parameters that exhibit generalization failures. The abstract and title could be edited to provide a more balanced reflection of these findings and to eliminate the somewhat "straw man"-ish implication that the field expects that the absolute values of learning rates should be the same across tasks.”

We thank the reviewers for this recommendation and have reworked the paper substantially to address the issue. We have modified the highlights, abstract, introduction, discussion, conclusion, and relevant parts of the Results section to provide equal weight to the successes and failures of generalization.

Highlights:

Abstract:

The introduction now introduces different potential outcomes of our study with more equal weight:

“Computational modeling enables researchers to condense rich behavioral datasets into simple, falsifiable models (e.g., RL) and fitted model parameters (e.g., learning rate, decision temperature) […]. These models and parameters are often interpreted as a reflection of ("window into") cognitive and/or neural processes, with the ability to dissect these processes into specific, unique components, and to measure participants' inherent characteristics along these components.

For example, RL models have been praised for their ability to separate the decision making process into value updating and choice selection stages, allowing for the separate investigation of each dimension. Crucially, many current research practices are firmly based on these (often implicit) assumptions, which give rise to the expectation that parameters have a task- and model-independent interpretation and will seamlessly generalize between studies. However, there is growing---though indirect---evidence that these assumptions might not (or not always) be valid.

The following section lays out existing evidence in favor and in opposition of model generalizability and interpretability. Building on our previous opinion piece, which---based on a review of published studies---argued that there is less evidence for model generalizability and interpretability than expected based on current research practices […], this study seeks to directly address the matter empirically.”

We now also provide more even evidence for both potential outcomes:

“Many current research practices are implicitly based on the interpretability and generalizability of computational model parameters (despite the fact that many researchers explicitly distance themselves from these assumptions). For our purposes, we define a model variable (e.g., fitted parameter, reward-prediction error) as generalizable if it is consistent across uses, such that a person would be characterized with the same values independent of the specific model or task used to estimate the variable. Generalizability is a consequence of the assumption that parameters are intrinsic to participants rather than task dependent (e.g., a high learning rate is a personal characteristic that might reflect an individual's unique brain structure). One example of our implicit assumptions about generalizability is the fact that we often directly compare model parameters between studies---e.g., comparing our findings related to learning-rate parameters to a previous study's findings related to learning-rate parameters. Note that such a comparison is only valid if parameters capture the same underlying constructs across studies, tasks, and model variations, i.e., if parameters generalize. The literature has implicitly equated parameters in this way in review articles […], meta-analyses […], and also most empirical papers, by relating parameter-specific findings across studies. We also implicitly evoke parameter generalizability when we study task-independent empirical parameter priors […], or task-independent parameter relationships (e.g., interplay between different kinds of learning rates […]), because we presuppose that parameter settings are inherent to participants, rather than task specific.

We define a model variable as interpretable if it isolates specific and unique cognitive elements, and/or is implemented in separable and unique neural substrates. Interpretability follows from the assumption that the decomposition of behavior into model parameters "carves cognition at its joints", and provides fundamental, meaningful, and factual components (e.g., separating value updating from decision making).

We implicitly invoke interpretability when we tie model variables to neural substrates in a task-general way (e.g., reward prediction errors to dopamine function […]), or when we use parameters as markers of psychiatric conditions (e.g., working-memory parameter and schizophrenia […]). Interpretability is also required when we relate abstract parameters to aspects of real-world decision making […], and generally, when we assume that model variables are particularly "theoretically meaningful" […].

However, in midst the growing recognition of computational modeling, the focus has also shifted toward inconsistencies and apparent contradictions in the emerging literature, which are becoming apparent in cognitive […], developmental […], clinical […], and neuroscience studies […], and have recently become the focus of targeted investigations […]. For example, some developmental studies have shown that learning rates increased with age […], whereas others have shown that they decrease […]. Yet others have reported U-shaped trajectories with either peaks […] or troughs […] during adolescence, or stability within this age range […] (for a comprehensive review, see […]; for specific examples, see […]). This is just one striking example of inconsistencies in the cognitive modeling literature, and many more exist […]. These inconsistencies could signify that computational modeling is fundamentally flawed or inappropriate to answer our research questions. Alternatively, inconsistencies could signify that the method is valid, but our current implementations are inappropriate […]. However, we hypothesize that inconsistencies can also arise for a third reason: Even if both method and implementation are appropriate, inconsistencies like the ones above are expected---and not a sign of failure---if implicit assumptions of generalizability and interpretability are not always valid. For example, model parameters might be more context-dependent and less person-specific that we often appreciate […].”

In the Results section, we now highlight findings more that are compatible with generalization: “For α+, adding task as a predictor did not improve model fit, suggesting that α+ showed similar age trajectories across tasks (Table 2). Indeed, α+ showed a linear increase that tapered off with age in all tasks (linear increase: task A: β = 0.33, p < 0.001; task B: β = 0.052, p < 0.001; task C: β = 0.28, p < 0.001; quadratic modulation: task A: β = −0.007, p < 0.001; task B: β = −0.001, p < 0.001; task C: β = −0.006, p < 0.001). For noise/exploration and Forgetting parameters, adding task as a predictor also did not improve model fit (Table 2), suggesting similar age trajectories across tasks.”

“For both α+ and noise/exploration parameters, task A predicted tasks B and C, and tasks B and C predicted task A, but tasks B and C did not predict each other (Table 4; Figure 2D), reminiscent of the correlation results that suggested successful generalization (section 2.1.2).”

“Noise/exploration and α+ showed similar age trajectories (Figure 2C) in tasks that were sufficiently similar (Figure 2D).”

And with respect to our simulation analysis (for details, see next section):

“These results show that our method reliably detected parameter generalization in a dataset that exhibited generalization. ”

We also now provide more nuance in our discussion of the findings:

“Both generalizability […] and interpretability (i.e., the inherent "meaningfulness" of parameters) […] have been explicitly stated as advantages of computational modeling, and many implicit research practices (e.g., comparing parameter-specific findings between studies) showcase our conviction in them […]. However, RL model generalizability and interpretability has so far eluded investigation, and growing inconsistencies in the literature potentially cast doubt on these assumptions. It is hence unclear whether, to what degree, and under which circumstances we should assume generalizability and interpretability. Our developmental, within-participant study revealed a nuanced picture: Generalizability and interpretability differed from each other, between parameters, and between tasks.”

“Exploration/noise parameters showed considerable generalizability in the form of correlated variance and age trajectories. Furthermore, the decline in exploration/noise we observed between ages 8-17 was consistent with previous studies [13, 66, 67], revealing consistency across tasks, models, and research groups that supports the generalizability of exploration / noise parameters. However, for 2/3 pairs of tasks, the degree of generalization was significantly below the level of generalization expected for perfect generalization.

Interpretability of exploration / noise parameters was mixed: Despite evidence for specificity in some cases (overlap in parameter variance between tasks), it was missing in others (lack of overlap), and crucially, parameters lacked distinctiveness (substantial overlap in variance with other parameters).”

“Taken together, our study confirms the patterns of generalizable exploration/noise parameters and task-specific learning rate parameters that are emerging from the literature [13].”

[Essential revisions]: “To determine what might be a reasonably expected ceiling on the degree of generalizability, as suggested by the reviewer, the authors could simulate data from agents using the same generative parameters across the three tasks and report the correlations between the resulting parameters estimates.”[Reviewer #3]: “With regard to obtaining a ceiling on correlations that indicate relative generalizability, it might be possible to get an answer to this w/o collecting new data by simulating behavior of the *same N agents* (with uniformly sampled params, or params sampled from prior group distribution used in the hierarchical fit, or params sampled from distributions centered around ideal observer values). If we then re-run model fitting on these simulated datasets, what is the max correlation between LRs estimated from different tasks? And can we bootstrap to obtain some Cis around this ceiling? In general, the authors consider the tasks to be fairly similar, but I'm not sure that is actually the case–- even A and C, which share the most similarities, differ markedly in the reward function (probabilistic vs. deterministic).”

We thank the reviewer for this excellent suggestion, which we think helped answer a central question that our previous analyses had failed to address, and also provided answers to several other concerns raised by both reviewers in other section. We have conducted these additional analyses as suggested, simulating artificial behavioral data for each task, fitting these data using the models used in humans, repeating the analyses performed on humans on the new fitted parameters, and using bootstrapping to statistically compare humans to the hence obtained ceiling of generalization. We have added the following section to our paper, which describes the results in detail:

“Our analyses so far suggest that some parameters did not generalize between tasks, given differences in age trajectories (section 2.1.3) and a lack of mutual prediction (section 2.1.4). However, the lack of correspondence could also arise due to other factors, including behavioral noise, noise in parameter fitting, and parameter trade-offs within tasks. To rule these out, we next established the ceiling of generalizability attainable using our method.

We established the ceiling in the following way: We first created a dataset with perfect generalizability, simulating behavior from agents that use the same parameters across all tasks (suppl. Appendix E). We then fitted this dataset in the same way as the human dataset (e.g., using the same models), and performed the same analyses on the fitted parameters, including an assessment of age trajectories (suppl. Table E.8) and prediction between tasks (suppl. Tables E.9, E.10, and E.11). These results provide the practical ceiling of generalizability. We then compared the human results to this ceiling to ensure that the apparent lack of generalization was valid (significant difference between humans and ceiling), and not in accordance with generalization (lack of difference between humans and ceiling).

Whereas humans had shown divergent trajectories for parameter α- (Figure 2B; Table 1), the simulated agents did not show task differences for α- or any other parameter (suppl. Figure E.8B; suppl. Table E.8), even when controlling for age (suppl. Tables E.9 and E.10), as expected from a dataset of generalizing agents. Furthermore, the same parameters were predictive between tasks in all cases (suppl. Table E.11). These results show that our method reliably detected parameter generalization in a dataset that exhibited generalization.

Lastly, we established whether the degree of generalization in humans was significantly different from agents. To this aim, we calculated the Spearman correlations between each pair of tasks for each parameter, for both humans (section 2.1.2; suppl. Figure H.9) and agents, and compared both using bootstrapped confidence intervals (suppl. Appendix E). Human parameter correlations were significantly below the ceiling for all parameters except α+ (A vs B) and epsilon / 1/β (A vs C; suppl. Figure E.8C). This suggests that humans were within the range of maximally detectable generalization in two cases, but showed less-than-perfect generalization between other task combinations and for parameters Forgetting and α-.”

[Essential revisions]: “Descriptions of the tasks that capture their distinguishing features through the lens of Markov Decision Process theory might be valuable to contextualize the current results within a broader reinforcement learning literature.”[Reviewer #3]: “The current study was not designed for systematically varying different aspects of the MDP to see how this affects inferred parameter generalizability and interpretability, which is okay. But perhaps one step the paper can make in this direction of theoretical consistency is to explicitly name the features of each task from an MDP theory standpoint, or at least additionally include this as a table or additional panel in Figure 1. E.g. size of state and action space (+ mapping to "set size" jargon), reward function, etc. Authors do this at several points in the paper (including the appendix), and this can only help as a practice that allows the RL community to think about tasks in a common way. In the present study, this convention can illuminate why the tasks presented here are actually quite different from one another. Task A: 4 states (1 per trial), 2 actions (visible on screen), stable stochastic reward function; Task B: stateless, 2 actions, reversal in stochastic reward function; Task C: RL-WM task: 2-5 states (1 per trial), 3 actions (not visible on screen), stable deterministic reward function.”

We thank the reviewer for this comment, and will address both points in turn:

1) We agree with the reviewer's sentiment about relative generalizability: If we all interpreted our models exclusively with respect to our specific task design, and never expected our results to generalize to other tasks or models, there would not be a problem. However, the current literature shows a different pattern: Literature reviews, meta-analyses, and Discussion sections of empirical papers regularly compare specific findings between studies. We compare specific parameter values (e.g., empirical parameter priors), parameter trajectories over age, relationships between different parameters (e.g., balance between LR+ and LR-), associations between parameters and clinical symptoms, and between model variables and neural measures on a regular basis. The goal of this paper was really to see if and to what degree this practice is warranted. And the reviewer rightfully alerted us to the fact that our data imply that these assumptions might be valid in some cases, just not in others.

2) With regard to providing task descriptions that relate to the MDP framework, we have included the following sentence in the Discussion section:

“Our results show that discrepancies are expected even with a consistent methodological pipeline, and using up-to-date modeling techniques, because they are an expected consequence of variations in experimental tasks and computational models (together called "context"). Future research needs to investigate these context factors in more detail. For example, which task characteristics determine which parameters will generalize and which will not, and to what extent? Does context impact whether parameters capture overlapping versus distinct variance? A large-scale study could answer these questions by systematically covering the space of possible tasks, and reporting the relationships between parameter generalizability and distance between tasks. To determine the distance between tasks, the MDP framework might be especially useful because it decomposes tasks along theoretically meaningful features of the underlying Markov Decision Process.”

[Reviewer #3]: “Finally, I think it is important to provide a bit more detail upfront about the training and instructions protocol for each task, and, to the extent that it is possible, provide evidence that instruction comprehension is not related to parameter differences. One simple analysis here would be to examine the "lose-stay" distribution in Tasks B and C and look for any evidence of bimodality that can be traced to how instructions were presented. I would also be curious if this distribution varies with age, potentially explaining the striking divergence in age trajectories for LR- in Tasks B and C.”

We are happy to hear the reviewer's general agreement to our interpretation of the results with regard to LR- task differences, and appreciate the reviewer's suggested alternative explanation. Indeed, if participants assumed (wrongly) that feedback was stochastic in task C, elevated values of LR- on this task might reflect a misunderstanding of task instructions, rather than learning from negative feedback, and the resulting difference in what LR- captures across tasks would lead to a decrease in generalizability. In order to address this issue, we will address each of the reviewer's points in turn:

(1) The task order was identical for all participants and is described in the section “Testing Procedure” in the Methods. The order was: task C; break; task A; task B. Because task B was presented after task C, global strategies could not have transferred from task B to task C.

(2) We have now added more specific information about training and task instructions to section “4.4. Task Design” in the Methods section (also see Appendix C). Because the instructions and training differed between tasks and accommodated for the younger population in multiple ways, we cannot explain everything in the current paper, but we refer the interested reader to the original publications:

“Task A ("Butterfly task")

The goal of task A was to collect as many points as possible, by guessing correctly which of two flowers was associated with each of four butterflies. Participants were instructed to guess which flower each butterfly liked more, having been told that butterflies would sometimes also choose the less-liked flower (i.e., act probabilistically). Correct guesses were rewarded with 70% probability, and incorrect guesses with 30%. The task contained 120 trials (30 for each butterfly) that were split into 4 equal-sized blocks, and took between 10-20 minutes to complete. More detailed information about methods and results can be found in […].

Task B ("Stochastic Reversal")

The goal of task B was to collect golden coins, which were hidden in two green boxes. Participants completed a child-friendly tutorial, in which they were instructed to help a leprechaun find his treasure by collecting individual coins from two boxes. Task volatility (i.e., boxes switching sides) and stochasticity (i.e., correct box not rewarded each time) were introduced one-by-one (for details, see […]). The task could be in one of two states: "Left box is correct" or "Right box is correct". In the former, selecting the left box led to reward in 75\% of trials, while selecting the right box never led to a reward (0\%). Several times throughout the task, task contingencies changed unpredictably and without notice (after participants had reached a performance criterion indicating they had learned the current state), and the task switched states. Participants completed 120 trials of this task (2-9 reversals), which took approximately 5-15 minutes. For more information and additional task details, refer to […].

Task C ("Reinforcement Learning-Working Memory")

The goal of task C was to collect as many points as possible by pressing the correct key for each stimulus. Participants were instructed to learn an "alien language" of key presses by associating individual pictures with specific key presses. Pressing the correct key for a specific picture deterministically led to reward, and the correct key for a stimulus never changed. Stimuli appeared in blocks that varied in the number of different stimuli, with set sizes ranging from 2-5. In each block, each stimulus was presented 12-14 times, for a total of 13 * set size trials per block. Three blocks were presented for set sizes 2-3, and 2 blocks were presented for set sizes 4-5, for a total of 10 blocks. The task took between 15-25 minutes to complete. For more details, as well as a full analysis of this dataset, refer to […].”

(1) We examined the “lose-stay” distributions of tasks B and C, but did not find evidence of bimodality:

**Author response image 4. sa2fig4:** 

Figure C.6 in the paper shows the age trajectories of lose-stay behavior over age, and we do not see increased error bars for the younger participants, which we would expect if there was increased bimodality at that age: